# Large-scale genome-wide study reveals climate adaptive variability in a cosmopolitan pest

Yanting Chen[1,2,3,4], Zhaoxia Liu[1,2,3,5], Jacques Régnière [6], Liette Vasseur [1,2,7], Jian Lin[8], Shiguo Huang[8], Fushi Ke[1,2,3,9], Shaoping Chen[1,2,3,4], Jianyu Li[1,2,3,4], Jieling Huang[1,2,3], Geoff M. Gurr [1,2,10✉], Minsheng You [1,2,3✉] & Shijun You [1,2,3✉]

Understanding the genetic basis of climatic adaptation is essential for predicting species' responses to climate change. However, intraspecific variation of these responses arising from local adaptation remains ambiguous for most species. Here, we analyze genomic data from diamondback moth (*Plutella xylostella*) collected from 75 sites spanning six continents to reveal that climate-associated adaptive variation exhibits a roughly latitudinal pattern. By developing an eco-genetic index that combines genetic variation and physiological responses, we predict that most *P. xylostella* populations have high tolerance to projected future climates. Using genome editing, a key gene, *PxCad*, emerged from our analysis as functionally temperature responsive. Our results demonstrate that *P. xylostella* is largely capable of tolerating future climates in most of the world and will remain a global pest beyond 2050. This work improves our understanding of adaptive variation along environmental gradients, and advances pest forecasting by highlighting the genetic basis for local climate adaptation.

[1] State Key Laboratory of Ecological Pest Control for Fujian-Taiwan Crops, Institute of Applied Ecology, Fujian Agriculture and Forestry University, Fuzhou 350002, China. [2] Joint International Research Laboratory of Ecological Pest Control, Ministry of Education, Fuzhou 350002, China. [3] Key Laboratory of Integrated Pest Management for Fujian-Taiwan Crops, Ministry of Agriculture, Fuzhou 350002, China. [4] Institute of Plant Protection, Fujian Academy of Agricultural Sciences, Fuzhou 350013, China. [5] College of Oceanology and Food Science, Quanzhou Normal University, Quanzhou 362000, China. [6] Natural Resources Canada, Canadian Forest Service, Quebec City, QC G1V 4C7, Canada. [7] Department of Biological Sciences, Brock University, St. Catharines, ON L2S 3A1, Canada. [8] College of Computer and Information Sciences, Fujian Agriculture and Forestry University, Fuzhou 350002, China. [9] Key Laboratory of Plant Resources Conservation and Sustainable Utilization, South China Botanical Garden, Chinese Academy of Sciences, Guangzhou 510650, China. [10] Graham Centre, Charles Sturt University, Orange, NSW 2800, Australia. ✉email: GGurr@csu.edu.au; msyou@fafu.edu.cn; sjyou@fafu.edu.cn

Human-induced climate change, especially gradual changes in temperature and precipitation[1], is impacting species' survival and distribution[2]. The ability of pests to successfully adapt to these changes will impact biodiversity, food production, and the economy. Intraspecific variation in tolerance to climate change has been documented for many plant and animal species[3,4]. Populations with high adaptive potential are expected to cope with changes in habitat suitability arising from climate change[5,6], but the mechanisms are not well understood. Studying the genetic mechanisms that underpin the adaptation of species to local climate is therefore important to predict both population- and global-level responses to future environmental change and assist in management efforts[5].

Genetic variation associated with climate variables has been demonstrated in several species[5,7–9]. Insects with high fecundity and short generation time can accumulate adaptive alleles rapidly through new mutations and standing variation, and potentially have high capacity to respond to changing environmental conditions[10–12]. However, little is known about the extent to which adaptive variation is driven by climate in arthropod species, and how populations differ in their capacity to adapt to climate change.

In this study, by combining a newly available genomic resource with climate data, we analyze the relationship between genomic variation and climate variables in the diamondback moth (DBM), *Plutella xylostella* L. (Lepidoptera: Plutellidae). This insect is one of the world's top 10 arthropod pests[13], with a global distribution spanning a remarkably wide range of climates[14]. We define a new eco-genetic index to examine population-level variation in response to climate change by combining the genetic offset (that quantifies the disruption of gene-environment relationships subject to future climates) with the ecoclimatic index (that describes phenology-based habitat suitability for species persistence). Subsequently, informed by a core dataset of identified nuclear SNPs, we functionally test a temperature-related gene to reveal its role in climatic adaptation in DBM. Our results imply that *P. xylostella* is largely capable of tolerating future climates in most regions of the world, and its pest status will be maintained beyond 2050.

## Results

**Climate associated genomic variation.** The fundamental resource for this study is a new dataset of genome-wide single nucleotide polymorphisms (SNPs) sequences of a worldwide sample of 532 DBM individuals collected from 114 locations (sites) in a diverse range of biogeographical regions[15]. To investigate the adaptive genetic variation associated with contemporary climates, we used a subset of samples from regions in which DBM is able to persist year-round with a positive ecoclimatic index (EI > 0)[16], where populations are subject to seasonally uninterrupted local selection by climatic factors. After quality filtering, we generated a dataset of 200,055 SNPs across 357 DBM individuals collected from 75 different sites worldwide (Supplementary Fig. 1, Supplementary Data 1 and 2). Using three complementary models: Samβada v0.5.3[17], latent factor mixed models (LFMM)[18], and Bayenv 2[19], we conducted a genome-wide scan to test climate associations for the quality-filtered SNPs. A total of 3648 putatively adaptive SNPs were identified by one or more of the three models (Fig. 1a, Supplementary Data 3, 4, and 5), showing the association between genetic variation and specific climate variables.

Understanding which climatic variables most strongly drive natural selection can provide insights into the biological mechanisms involved in species distribution and population dynamics[4,5]. We used generalized dissimilarity modelling (GDM)[20] to examine climate-mediated genomic variation among different populations based on 517 SNPs selected among the 3648 SNPs. Of the 12 environmental variables that had a pairwise Pearson's r less than 0.8, the top four variables included one that is precipitation-related (bio18) and three that are temperature-related (bio03, bio09, and bio08) (Fig. 1b and Supplementary Table 1). Environment-associated genetic variation exhibited a roughly latitudinal pattern, regardless of geographical region or continent, suggesting that DBM populations from the same latitude exhibit comparable genomic composition (Fig. 1c).

**Genetic vulnerability and adaptive potential.** We applied a metric of genetic vulnerability called "genetic offset", originally developed by Fitzpatrick and Keller[4], to investigate which DBM populations might be more vulnerable to future climate change. Populations with higher genetic offset are more vulnerable to climate change and require greater adaptive potential (or genetic variation necessary for adaptation) to the changing environment[4]. Under greenhouse gas emission scenarios RCP8.5 for 2050, the genetic offset was low for most populations (Fig. 2a). The comparison of genetic offset under different scenarios (RCP2.6, RCP4.5, RCP6.0, and RCP8.5) showed an increasing trend with rising greenhouse gas emissions (Supplementary Fig. 2). Most DBM populations appeared to experience low disruption of gene-climate associations under future climate. Taken together with our previous findings that high levels of both intrapopulation genetic polymorphism and interpopulation genetic differentiation enable DBM to adapt readily to different environments worldwide[15,21], we thus assume that DBM will likely remain a damaging pest across most of its range. In contrast, high levels of genetic offset, indicating the need for more comprehensive adaptive change to future climate change, were observed in scattered populations of South America and Southeast Asia (Fig. 2a).

We then used a validated bio-climatic model (CLIMEX) that combined climate data with eco-physiological traits[16,22] to predict the habitat suitability for DBM under the RCP8.5 scenario for 2050. Using the current EI value as a benchmark, we calculated the difference in ecoclimatic index (DEI) between current and projected future climate scenarios across year-round persistence regions of DBM worldwide. We observed that the total area of regions with decreasing EI was much larger than that of regions where EI increased under the RCP8.5 scenario (Fig. 2b). DBM populations in regions of decreasing EI may be challenged by habitat suitability decline, suggesting they may need to harness their adaptive potential to cope with the habitat degradation under future climate.

Local adaptation, resulting from environment-driven intraspecific genetic differentiation, is an important feature of species that inhabit spatially heterogeneous habitats[23]. To better reflect the role of population-level genetic variation in DBM's responses to climate change, we developed a new metric, the "eco-genetic index" (EGI), which combines the genetic offset[4] and EI-based prediction of climatic habitat suitability. We assume that regions with increasing EI will remain hospitable to DBM while regions with decreasing EI will be challenged by habitat suitability decline under the RCP8.5 scenario for 2050. Thus, our analysis focused on regions with decreasing EI. Under climate-change scenario RCP8.5, the challenges to most populations (Fig. 2b) should be moderate since they are predicted to experience only minor interruptions of gene-climate associations, except for some populations in South America and Southeast Asia (Fig. 2a). Therefore, most DBM populations will maintain their pest status in the context of future climate beyond 2050, without dramatic

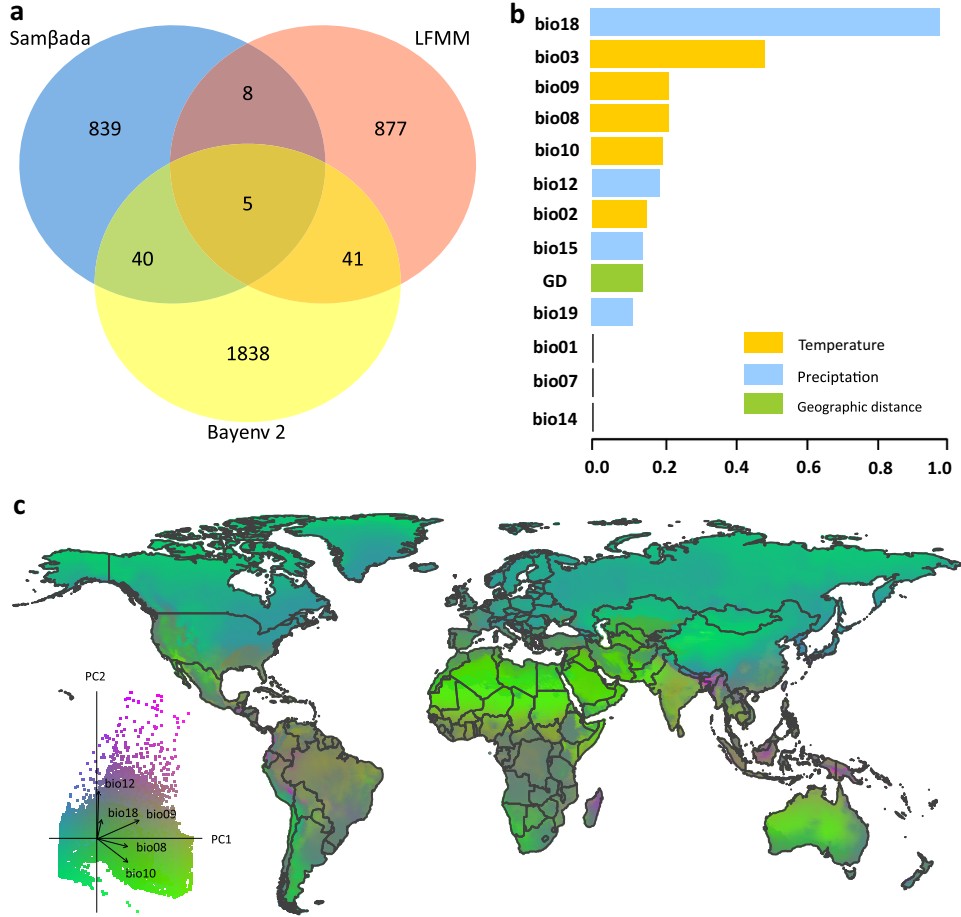

**Fig. 1 Association between genomic variation and climate variables for diamondback moth. a** Venn diagram showing the number of climate-associated SNPs identified by the three different models, Samβada, LFMM, and Bayenv 2. **b** Ranked importance of climatic and geographical variables based on generalized dissimilarity modelling (GDM), showing that the genomic variation can be mainly explained by climate variables. **c** GDM-predicted pattern of climate-associated genomic variation along environmental gradients across the world. Colors are based on the results of the principal components analysis (PCA) of transformed climate variables. The PCA-based biplot indicates the contribution of climate variables to the predicted pattern of genomic variation. Arrows (or vectors) show the loadings of climate variables on PCA.

change to EGI-based habitat suitability in most of the world (Fig. 2c).

**Genetic basis of climate adaption**. To explore the molecular and genetic basis of climate adaptation, we performed a functional analysis of a core dataset of 94 putatively adaptive SNPs that were identified by at least two of three models (Samβada, LFMM, and Bayenv 2; Fig. 1a, Supplementary Table 2). These SNPs are widely distributed across the DBM genome, with 39 SNPs located in the coding sequence (CDS) and intronic regions across 30 genes (Supplementary Table 2). These genes are predicted to cover a wide range of functions, mainly associated with physiological responses and metabolic regulation[21], with some previously documented as having temperature-related functions. For example, the cytochrome P450 gene (*Px007339*) is known for heat and cold stress tolerance in insects[24,25]; whilst folding[26] and gene expression[27] of the nicotinic acetylcholine receptor (*Px017786*) are temperature-sensitive.

Based on the 39 SNPs located in the CDS and intron regions from the core subset (Supplementary Table 2), we identified four SNPs from the coding region of a single gene, *PxCad*. We also found two SNPs from the temperature-related *Px007339* and *Px017786* genes (Supplementary Table 2). Allelic frequencies for

these six SNPs are presented according to the global distribution of their genotype frequency. Strong selection signals were observed in the North American populations, followed by some parts of Southeast Asia, Eurasia, Southern Africa, and Southern Oceania, while no such selection signals were detected for most populations of South America, and Central Africa (Supplementary Fig. 3).

With the highest number of SNPs (4 out of 94) identified in the core dataset (94 putatively adaptive SNPs), *PxCad* was the most likely candidate to be involved in regulating DBM's responses to climate change. Homologs of *PxCad* encode cadherin-like proteins, which are known to be involved in cell adhesion[28], sensory perception of light[29], vocal and locomotory behavior[30], and have been evidenced to be responders for thermal stress[31]. To verify the function of *PxCad* in mediating adaptation to extreme temperatures, we performed RT-qPCR to profile its expression under different temperature regimes using a wild-type strain of DBM, G88. Overall, both in males and females, the expression of *PxCad* gene tended to be more sensitive to low temperatures than high temperatures and exhibited significantly upregulated expression at −14 °C, −17 °C, and −20 °C, suggesting an important role in response to cold extremes (Fig. 3).

Using CRISPR/Cas9 genome-editing, we successfully knocked out *PxCad* in the wild-type G88 strain (Fig. 4). A mixture of Cas9

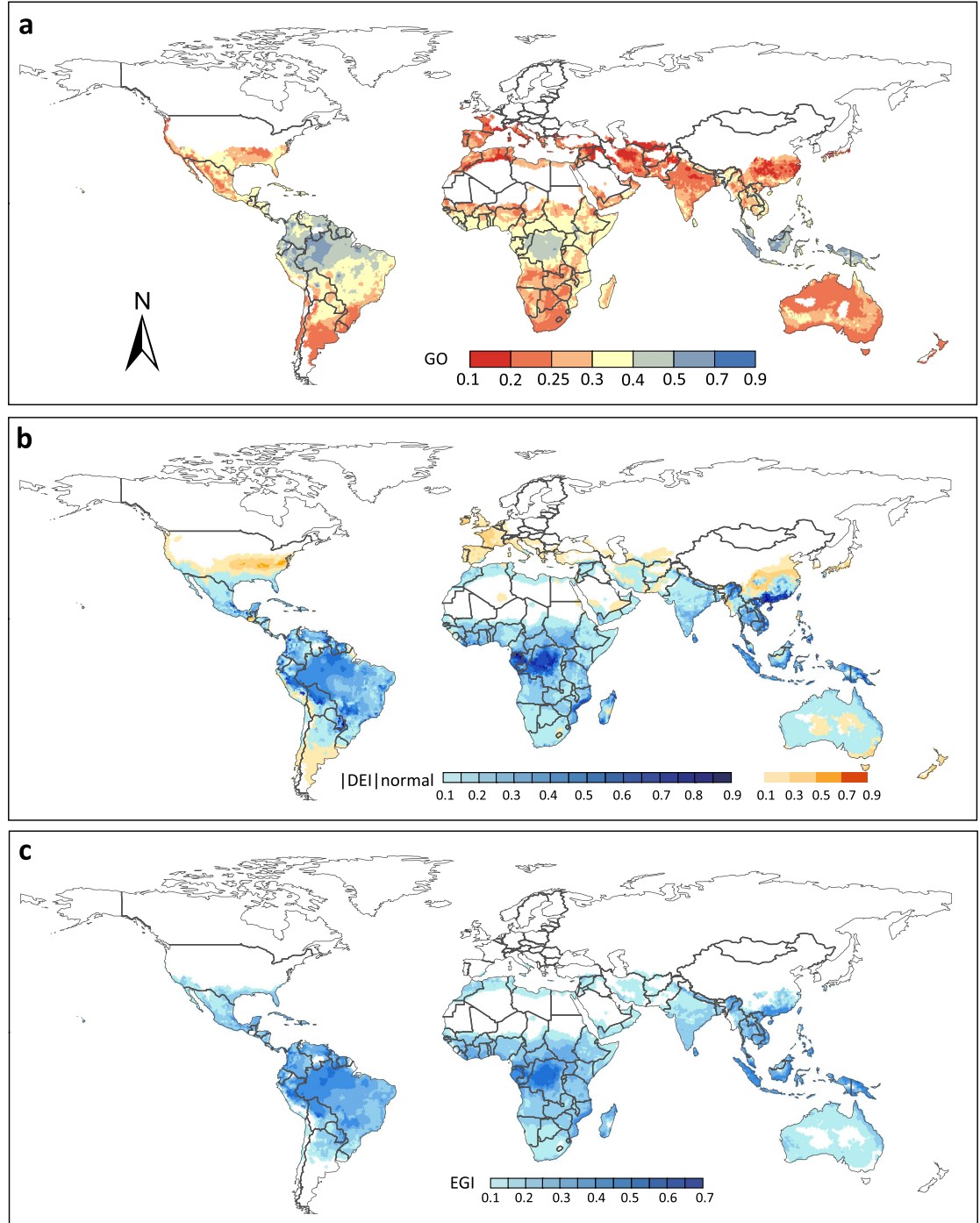

**Fig. 2 Vulnerability of diamondback moth to climate change under greenhouse gas emission scenario RCP8.5 in 2050. a** Projection of genetic offset (GO) based on generalized dissimilarity modelling (GDM). **b** Projection of |DEI|$_{normal}$ (difference in ecoclimatic index) between current and projected future climate scenarios based on CLIMEX model, with warm colors showing the EI-increased regions (DEI > 0) and blue colors showing the EI-decreased regions (DEI < 0). **c** Projection of eco-genetic index (EGI) based on the combined estimation of genetic offset (GO) and eco-climatic index (EI).

mRNA and sgRNAs was injected into 195 eggs of this strain, 102 of them successfully hatched, and 83.3% (85/102) of larvae reached adulthood. After being individually crossed with G88, we confirmed site-specific mutagenesis in 32.8% (21/64) of $G_0$ moths, observing 15 bp, 46 bp, 8 bp, and 18 bp deletions at the target site (Fig. 4a). After screening using single-pair crosses and molecular identification, one homozygous mutant strain (MU, G88-Cad) with a 46-bp deletion in *PxCad* exon 3 was generated (Fig. 4b). Using nano-LC-MS/MS to analyze the gel slices (~120–250 kDa

of the BBMV proteins) separated by SDS-PAGE (Fig. 4b), we identified nineteen tryptic peptides specific to *PxCad* from the G88 strain (Fig. 4c and d). None of these peptides were detected in the G88-Cad strain, which confirmed that PxCad protein was totally disrupted in the *PxCad*-knockout strain.

We then examined the difference in survival rate between the wild-type (WT) and *PxCad*-deficient mutant (MU) strains under favorable temperature (26 °C) and several extreme temperatures (40 °C, 41 °C, 42 °C, 43 °C, −14 °C, −17 °C, −20 °C). Compared

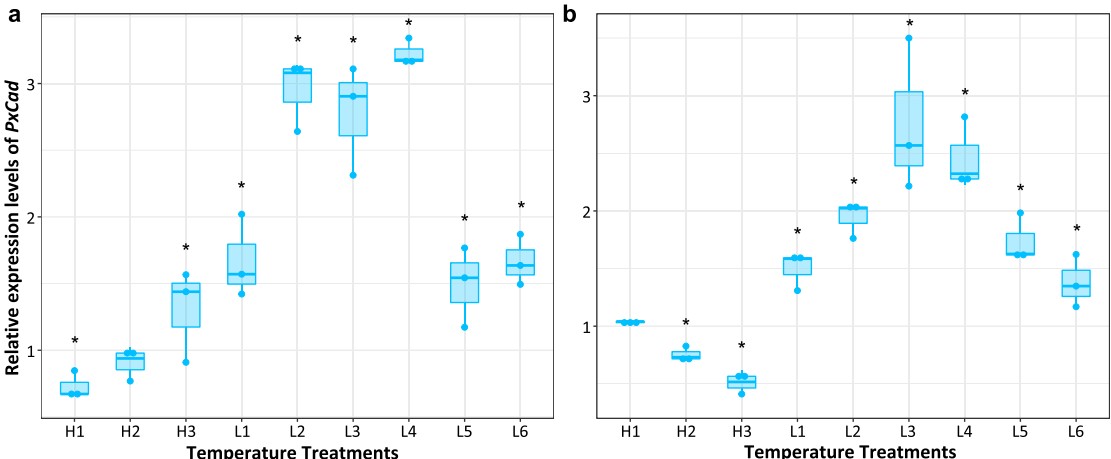

**Fig. 3 Effect of extreme temperatures on expression patterns of the *PxCad* gene in the wild-type (WT) diamondback moth strain (G88). a** Expression level of *PxCad* in males. **b** Expression level of *PxCad* in females. Temperature treatments included three high-temperatures treatments: (1) H1: 40 °C for 30 min, (2) H2: 43 °C for 30 min, (3) H3: 43 °C for 30 min with 24 h of recovery at 26 °C; and six low-temperature treatments: (1) L1: −14 °C for 30 min, (2) L2: −14 °C for 30 min with 24 h of recovery at 26 °C, (3) L3: −17 °C for 30 min, (4) L4: −17 °C for 30 min with 24 h of recovery at 26 °C, (5) L5: −20 °C for 15 min, and (6) L6: −20 °C for 15 min with 24 h recovery at 26 °C. Expression of *PxCad* at 26 °C was set as control with a relative expression value being set as 1. The horizontal line in boxes represents the median value of three replicates, boxes show 25th–75th percentiles, and points represent the original data. Expression of *PxCad* in each treatment was compared with control using independent t-test. *denotes significant difference between control and treatment (*t*-test, α < 0.05).

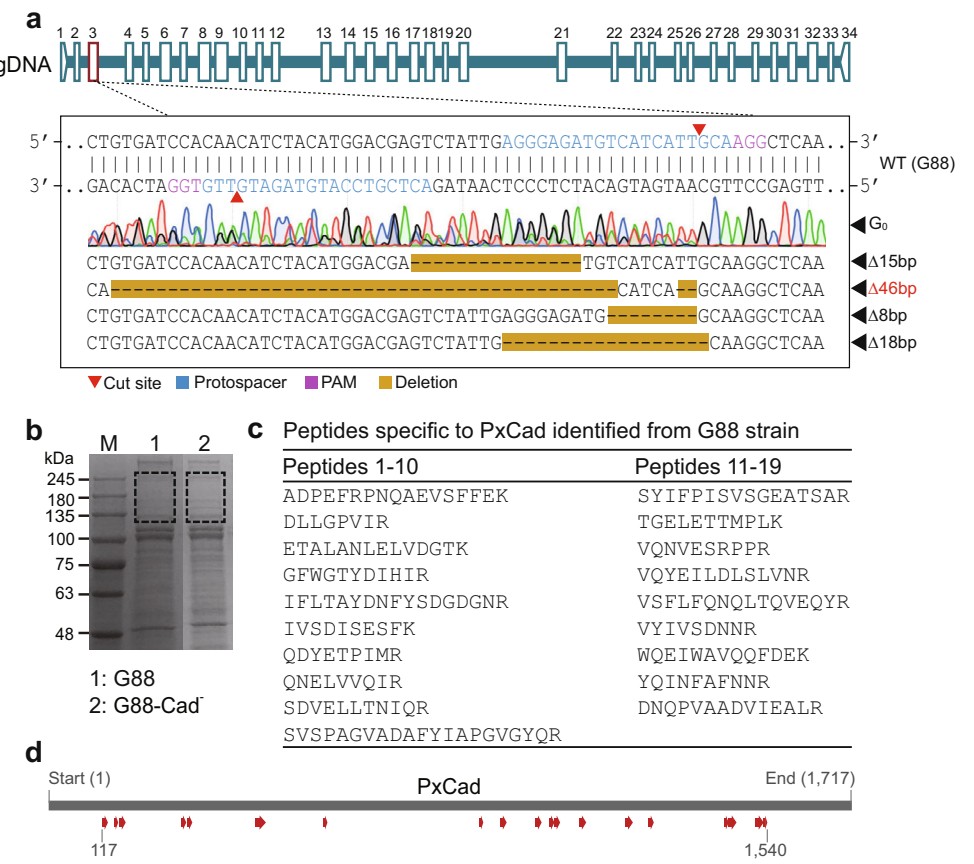

**Fig. 4 Mutagenesis of the gene, *PxCad*, mediated by the CRISPR/Cas9 genome editing system for diamondback moth. a** Representative sequencing trace of the PCR fragment from mutated G0 adults with multi-peaks at the cleavage site and representative sequence of the diverse indel mutations flanking the sgRNA target sites of *PxCad* in the G0 individuals. The Δ46 in red denotes the deletion mutation kept establishing the G88-Cad mutant (MU) strain. **b** SDS-PAGE profile of BBMV protein from the WT (G88) strain and the MU (G88-Cad) strain. **c** Details of the 19 peptides specific to *PxCad* identified from the WT (G88) strain. **d** Map of the full-length PxCad protein showing the position of 19 peptides (red arrows) specific to *PxCad* identified from the WT (G88) strain and absent from the MU (G88-Cad) strain. Numbers indicate the position of amino acid residues.

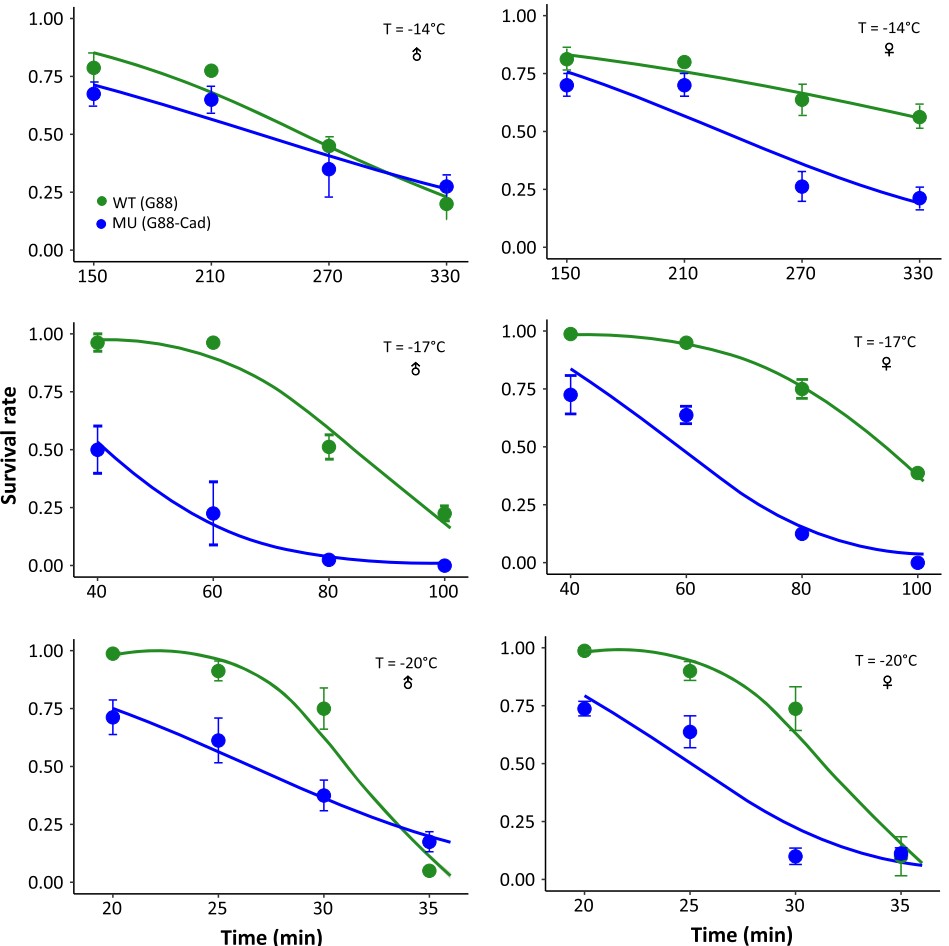

**Fig. 5 Effect of extreme temperatures on survival rates of the wild-type (WT, green) and *PxCad*-deficient (MU, blue) diamondback moth strains.** Survival rates of diamondback moth are presented for males (left) and females (right) after exposure to low temperatures (T = −14 °C, −17 °C, and −20 °C, respectively). Survival rates are represented as mean ± SE. Twenty individuals were used for each replicate, with four replicates in each treatment. The curves were generated from the equations in Supplementary Table 3.

with the WT, survival of the MU was not affected at favorable temperature, but the survival rate declined at extreme temperatures. In the cold-shock experiment (Fig. 5 and Supplementary Table 3), the MU was less resistant to cold stress than the WT, with a significantly lower survival rate at all temperatures tested, although among males at −14 °C, there was no significant difference in survival between strains. The heat shock experiment showed that the MU was also less tolerant to heat stress than the WT, especially at the highest temperature (Supplementary Fig. 4 and Supplementary Table 4). Altogether, this evidence supports a previously unknown function of *PxCad* in regulating DBM response to temperature, both under cold and heat stresses.

## Discussion

In this study, we provide evidence of the genetic basis of climate adaptation in DBM, a worldwide pest important to cruciferous crop production and thus the economy. Climate-associated genetic variation in DBM populations was quantified and visualized. A multi-model analysis of Samβada, LFMM, and Bayenv 2 allowed robust identification of climate-associated adaptive loci, reducing false-positives[32].

Our analyses with the nuclear SNPs from geographically distributed samples go further in defining the effects of both temperature- and precipitation-related variables on the climate-associated genetic variation in DBM populations worldwide. This follows a number of recent studies demonstrating the key role of temperature in mediating environment-associated adaptive variation for other insects: *Phaulacridium vittatum*[33], *Chironomus riparius*[34], and *Ceracris kiangsu*[35]. Further, physiological data can be used to determine the tolerance of DBM to future climate at the species level[30]. To date, however, there are no site-specific data available for investigation on the physiological variation in different populations of DBM. In this study, we found that most DBM populations might experience little interruption of existing gene-environment associations under projected future climates. This genomic association with climate may elicit region-specific responses to climate change and indicates that DBM is capable of persisting year-round as a pest in most regions of the world beyond 2050 under the RCP8.5 climate change scenario. This is of practical use to future pest management because the majority of DBM populations are shown to be resistant to changing environments under a future climate. Not only will pest management in these regions need to be maintained and strengthened, but also cooler areas where DBM seasonally colonizes are likely to become more vulnerable as temperatures increase, rendering the habitats favorable for permanent residence. For a species with high migratory capacity, like DBM, pest status is likely to increase given high levels of gene flow. DBM needs to be monitored at landscape and regional scales (in addition to conventional monitoring at a local scale), considering its genetic adaptive capacity and the spatial dynamics of insecticide resistant strains.

Using RT-qPCR and CRISPR/Cas9 approaches in this species, we verified that a specific gene, *PxCad*, represented a temperature-sensitive responder to climatic change, thus contributing to the genetic basis of adaptive evolution. *PxCad* is annotated as encoding cadherin-like proteins[36,37]. Classical cadherins are a superfamily of transmembrane proteins involved in regulating cell-cell adhesion, signal transduction and tissue morphogenesis[38,39]. In mammals, epithelial cadherin (E-cadherin) is involved in morphogenesis[40,41] whilst in insects, studies of cadherin-like proteins have focused primarily on their involvement in mediating resistance to the biological insecticide, Bt[42–45]. Pigott & Ellar[46] have also demonstrated the roles of cadherin-like proteins in maintaining structural integrity of midgut epithelial organization.

Thermal stress (heat or cold) generally disrupts cellular homeostasis[47,48]. Our results show that *PxCad* expression in female adults was significantly downregulated at high temperature. This is consistent with E-cadherin studies in human lung adenocarcinoma cells[49] and in the purple sea urchin *Strongylocentrotus purpuratus*[50]. In contrast, cold stress significantly upregulated *PxCad* expression in both males and females, and the tolerance to cold stress in DBM declined when *PxCad* was knocked-out. Comparable phenotypes have also been reported in the clam, *Ruditapes philippinarum*, with cadherin genes acting as responders to cold stress[31]. Under heat and cold stress, considerable variation of *Pxcad* expression and survival rates between DBM stains indicates that *PxCad* is involved in regulating DBM's response to thermal stress.

To understand the potential distribution and pest status of DBM, Zalucki & Furlong[16] developed a CLIMEX-based algorithm to predict the ecoclimatic index (EI) for the habitat suitability of DBM using climate data and eco-physiological traits. However, the EI-based prediction of habitat suitability is highly generalized without reflecting population-specific levels of genetic tolerance. In this paper, based on our recently-generated genomic data from a worldwide sample[15], we have developed a new metric, the eco-genetic index (EGI), which combines the genetic offset (GO)[4] and difference in ecoclimatic index (DEI) between current and projected future climate scenarios. This index mechanistically represents the genetic change required for adaptation to a changing environment, and reflects how population-specific genomic data can be incorporated into EI to better predict the habitat suitability of DBM when subjected to climate change, particularly when the population-specific data on physiological tolerance levels are not available. Our results indicate that DEI and GO are functionally complementary. With the EI prediction, we are able to determine the regional distribution of DBM throughout the world, which helps us focus on the regions in which DBM persists year-round with a positive ecoclimatic index (EI > 0)[16]. Using the new metric (EGI), we have identified that some DBM populations in regions of decreasing EI (such as in the central Africa and southern China) under future climate can overcome the challenge of habitat suitability decline given lower levels of genetic vulnerability, suggesting that only minor adaptive evolution will be required in situ to keep-up with climate change (Fig. 2). Looking ahead, if population-specific data on tolerance levels (to match our population-specific genomic data), and data on phenotypic plasticity and epigenetic responses were available, we would be able to make more robust conclusions, potentially supported by the development of a still more sophisticated metric that incorporates relevant information on additional aspects of adaptive capacity.

It is increasingly recognized that acclimatization through non-genetic inheritance (e.g., epigenetic processes) may buffer populations against environmental changes, allowing rapid adaptive responses to climate change[51–53]. Because the mutation rate of epigenetic sites is significantly higher than that of DNA sequences, epigenetic modification provides a complementary mode for species to respond to a changing environment in a rapid and finely regulated process[54,55]. In addition to directly regulating the expression of temperature responsive genes, epigenetic effects can also regulate other traits to indirectly affect the response of insects to temperature fluctuation[56]. However, epigenetics is a recently emerged field in insect studies[57], so further work is needed to understand the role of non-genetic effects in adaptation to future climates including how they interact with genetic adaptive capacity.

## Methods

**Genomic data.** The foundational resource for this study was a dataset of 40,107,925 nuclear SNPs sequenced from a worldwide sample of 532 DBM individuals collected in 114 different sites based on our previous project[15]. DNA was extracted from each of the 532 individuals using DNeasy Blood and Tissue Kit (Qiagen, Hilden, Germany) following the manufacturer's protocol, and eluted from the DNeasy Mini spin column in 200 µl TE buffer. Genomic sequencing was performed with Illumina HiSeq 2000 at BGI, Shenzhen, China, to produce 90 bp paired-end reads for every individual. Using custom scripts, raw reads were processed to filter out poor reads with 10 ambiguous "N" bases, >40% low-quality bases, or identical sequences at the two ends and obtain clean reads. The clean reads were mapped onto the DBM reference genome (v2)[21] with Stampy (v1.0.27) using default parameters. SNP calling was then performed using the GATK HaplotypeCaller with parameters --emitRefConfidence GVCF --variant_index_type LINEAR --variant_index_parameter 128,000. The 40,107,925 nuclear SNPs generated present one variant on average in every six bp of the reference genome, which is the densest variant map for any organism, including the recently released data on human[58] and *Arabidopsis thaliana* genome sequences[59]. The SNP dataset is available at https://www.ebi.ac.uk/ena with the accession code PRJEB24034.

In the present study, to investigate the genetic variation associated with climate, we excluded samples from the regions that are only seasonally suitable for growth of DBM (with the Ecoclimatic Index EI = 0)[16]. This was done because in these regions, populations are unlikely to receive perennially unpunctuated selection by local environmental variables and genetic variation cannot be continuously passed on to future generations over years. Specifically, regions that are only seasonally suitable for DBM growth and development (i.e., with an ecoclimatic index, EI = 0) are too harsh to allow survival in low temperature conditions during the winter. Annual recolonization of those regions from areas where DBM can overwinter (with an ecoclimatic index, EI > 0) has been biologically and genetically confirmed[14,60–63]. No genetic differentiation was found among different geographical populations spanning from overwintering regions to seasonally inhabited regions[15,62,63]. If migration from one habitat overwhelms the other, migration from the source introduces new genetic variation that may prevent local adaptation[64,65]. The retained samples included 372 individuals from 78 sampling sites in the year-round persistence regions of DBM across the world (where EI > 0). These samples were collected from different continents, with 13 samples from Africa, 29 from Asia, 5 from Europe, 13 from North America including Hawaii, 12 from South America, and 6 from Oceania (Supplementary Fig. 1 and Supplementary Data 1).

The retained 372 individuals shared a subset of 34,969,375 SNPs, which accounted for 87.19% of the total SNPs (40,107,925) and represented most of the genomic variation among 532 individuals (Supplementary Data 2)[15]. Using VCFtools v.0.1.6, we excluded the SNPs with minor allele frequency (MAF) < 5% and missing rate >10%. We then sampled data by examining single SNPs in small, 25 bp DNA window to focus on loci independent of linkage disequilibrium. After quality filtering, a total of 200,055 bi-allelic SNPs across 357 DBM individuals collected from 75 different sites worldwide was retained for further analysis (Supplementary Fig. 1, Supplementary Data 1 and 2).

**Climate Data.** Nineteen climate variables related to temperature and precipitation (Supplementary Table 1), which are known to have impacts on physiological and ecological traits of insects, were retrieved with high resolution from WorldClim, a public database (https://www.worldclim.org)[66].

**Climate associated genomic variation**

*Identification of SNPs under climate selection.* Three models, Samβada v0.5.3[17], latent factor mixed model (LFMM) v1.4[18] and Bayenv 2[19], were used to identify putatively adaptive loci associated with climate variables. Samβada identifies associations between specific genetic markers and environmental variables by logistic regression[17]. Simple univariate and multivariate logistic regression models for each climate variable were fitted. A single SNP was considered to be a candidate locus when the log-likelihood ratios (G scores) and/or Wald scores were significant with Bonferroni correction at a 99% confidence level. LFMM is based on population genetics, ecological modelling, and statistical analysis to identify the

candidate loci that are highly correlated with environmental variables[18]. SNPs showing an association with climate variables were identified based on z-scores, which was computed using 10,000 cycles and 5000 sweeps for burn-in. We used the R package *LEA* to estimate the median z-scores of 5 runs and re-adjusting p-values with FDR correction[67]. SNPs with median z-scores above the absolute value of 4 and corresponding to $P$ value $< 10^{-5}$ were considered as significant locus. In Bayenv 2, a covariance matrix based on putatively neutral markers is used as a null model to control for demographic effects when testing relationships between the genetic differentiation and a given environmental variable[19]. We randomly sampled SNPs at 200 SNP intervals from the SNP dataset. A total of 117,887 SNPs with loose linkage disequilibrium were obtained for developing the covariance matrix, which was estimated with 100,000 iterations. We then assessed the correlations between individual SNP and 19 climate variables at 100,000 Markov chain Monte Carlo (MCMC) for Bayes factor analysis. Five independent runs of the Bayenv program were performed with different random seeds[68]. The results were presented as Bayes factors (BFs). An averaged log10(BF) value of five runs >1.5 is considered as high support for a model where environmental parameters have significant effects on allele frequencies[69]. A total of 3648 putative adaptive SNPs were identified by at least one of the three models (Samβada, LFMM, and Bayenv 2; Supplementary Data 3, 4 and 5)

*Prediction of climate-associated genomic variation.* Generalized dissimilarity modelling (GDM)[20], a distance-based method, can account for the nonlinear relationship between genetic variation and environmental/geographical factors, and has been recently used to map ecological adaptation from genomic data under current and future climates[4]. First, we examined a spatially explicit selection process for each of the putative adaptive SNPs using GDM[4], with the R package *gdm*[70]. We subsampled the genetic dataset to include only populations with sample size ≥5 (60 populations) to obtain accurate allele frequencies. To reduce the number of bioclimatic variables, we preferentially discarded those with multiple correlated variables (Supplementary Table 5). We then ran the GDM model using 3,648 SNPs and the variables with Pearson $|r| < 0.8$ (bio02, bio03, bio07, bio08, bio09, bio12, bio15, bio18, and bio19), one of bio01 and bio11 ($|r| = 0.92$), one of bio5 and bio 10 ($r| = 0.94$), as well as one of bio14 and bio17 ($|r| = 1.0$). The subset of 12 bioclimatic variables with highest value of explained deviance (28.17%) in the GDM model were retained, including bio01, bio02, bio03, bio07, bio08, bio09, bio10, bio12, bio14, bio15, bio18, bio19, for further analysis. Pairwise $F_{ST}$ matrix among 60 populations were calculated for each of the 3648 SNPs using the R package *hierfstat*[71], and rescaled between 0 and 1. Geographical distance in the GDM was based on Euclidean distance as the thirteenth variable to test whether genetic variation across environmental gradients was better explained by climate variables than geographical distance, which effectively acts as a screening for SNPs that may respond predominantly to neutral genetic process including isolation by distance[72]. The relative importance of the 12 climate variables and geographical distance was ranked based on the fitted I-Splines in GDM (Fig. 1b). The maximum value of each variable in the fitted I-Splines was rescaled between 0 and 1. Those SNPs with geographical distance ranking as one of the 3 most important variables were excluded in the following GDM analysis. In addition, we randomly sampled 200 SNPs as a "reference group" to test its explainable proportion of the GDM deviance. According to our GDM record, the reference SNP group accounted for 11.2% of the GDM deviance for the entire model, so that those SNPs with a < 11.2% contribution to the GDM deviance were also excluded in further GDM analysis. After additional filtering, 517 of 3,648 SNPs were retained. The 517-SNP-based genetic distance matrix was further integrated with geographical distance and climate variables to be used in the entire GDM model, that explained 42.80% of the deviance for the 517 SNPs. To predict the climate adaptation of DBM, we finally retrieved current climate variables at 61,655 gridded points across the world from WorldClim, using ArcGIS 10.2. The *gdm.transform* function was used to predict and map the pattern of climate-associated genomic variation along environmental gradients across the world (Fig. 1c). The genetic turnover was summarized using a principal component analysis (PCA), with the top three components transformed for visualization in a red-green-blue (RGB) color scale as suggested in the GDM manual[70]. Loadings based on the principal components indicate the direction and magnitude of association with adaptation to different predictors (Fig. 1c). The genetic variation along environmental gradients in DBM across the world was visualized, with similar pattern of genetic composition at climate-adaptive loci illustrated by similar colors (Fig. 1c).

## Genetic vulnerability and adaptive potential
*Prediction of genetic vulnerability.* To predict the population-level variation in "genetic vulnerability" (GV) under future climate scenarios, we used the "Genetic offset" (GO) method of Fitzpatrick and Keller[4]. The GO represents the extent of mismatch between current and expected future genetic variation based on genotype-environment relationships modelled by GDM analysis[73]. The projected future climate variables of 2050 and 2080 for four different greenhouse gas scenarios, Representative Concentration Pathways (RCPs), including RCP2.6, RCP4.5, RCP6.0 and RCP8.5 based on the NorESM1-M Global Climate Model (GCM) across the world were retrieved from WorldClim, using ArcGIS 10.2. Those four RCPs represent different gas emission scenarios, reflecting mild (RCP2.6) to extreme (RCP8.5) conditions[74]. The GO was predicted by *predict.gdm* function in

R package *gdm*. A metric of GO for each of the gridded climate points was obtained, which implied that the populations with greater genetic offset would be more vulnerable or less tolerant to the future climate change. The resulting GOs were rescaled between 0.1 and 0.9[75,76], and then mapped with ArcGIS 10.2 to show the geographical distributions of population-level variation in genetic tolerance to future climate changes (Fig. 2a).

*Prediction of habitat suitability.* The CLIMEX model, which has been shown to be effective for examination of species distribution under future climate scenarios[77], was used to predict the habitat suitability for DBM in 2050 under the RCP8.5 scenario, the only one that is available in CliMond (https://www.climond.org/). The CLIMEX model for DBM in Zalucki and Furlong[16] was developed based on temperature, moisture, and stress indices. Temperature indices include limiting minimum (DV0) and maximum temperature (DV3), lower (DV1), and upper (DV2) optimal temperature. Moisture indices include minimum (SM0) and maximum (SM3) tolerable soil moisture, lower (SM1) and upper (SM2) optimal soil moisture. Stress indices include cold stress, heat stress, dry stress, wet stress, and hot-wet stress. The values of these parameters for prediction of habitat suitability in our study were taken from Zalucki & Furlong[16]. We altered one of the parameters in CLIMEX[16], changing the hot-wet stress temperature threshold from 30 °C to 32 °C based on a study on the relationship between temperature and developmental rate showing that DBM can survive and develop at temperatures <32 °C[78] (Supplementary Table 6). The resulting ecoclimatic index (EI) values based on our CLIMEX simulation were used to calculate the difference in ecoclimatic index (DEI) between current and projected future climate scenarios using the equation: $DEI = EI_F - EI_C$, where $EI_F$ is the ecoclimatic index under the projected future climate scenario and $EI_C$ is the ecoclimatic index under the current climate. Functions of Inverse Distance Weighting (IDW) and Overlay in ArcGIS 10.2 were performed to generate maps showing the predicted distribution of DEI values (with DEI > 0 showing the EI-increased regions and DEI < 0 showing the EI-decreased regions) for each of the gridded climate points across the year-round persistence regions worldwide (Fig. 2b).

*Prediction of eco-genetic adaptation.* The ecoclimatic index (EI) values based on CLIMEX simulations are highly generalized and do not reflect evolutionary adaptation and variation of tolerance levels among populations, because of the absence of information on fitness traits. Based on our recently-generated dataset of genome-wide single nucleotide polymorphisms (SNPs) sequenced from a world-wide DBM sample of different locations (sites) across a diverse range of biogeographical regions[15], we developed a new metric, the "eco-genetic index" (EGI), which combines the predictions based on genetic offset (GO) with the difference in ecoclimatic index between current and projected future climate scenarios (DEI). EGI allows us to incorporate our population-specific genomic data into EI for better predicting the habitat suitability of DBM subject to climate in 2050 under RCP8.5 scenario based on NorESM1-M GCM. Because DEI and GO, are correlated (Pearson's $R$[79] = 0.53, $P < 0.001$; Supplementary Fig. 5), we used a linear normalization transfer function to make values of $dei_i$ and $go_i$ dimensionless, and then used the weighted geometric averaging (WGA) operator and the artificial bee colony (ABC) algorithm to optimize the value of alpha and improve the algorithm of EGI. Here, only regions of decreasing EI (DEI < 0) were considered because in these regions, DBM populations will be challenged by habitat suitability decline under the RCP8.5 scenario for 2050. Each gridded point in ArcGIS was described as a vector: $P_i = \{dei_i, go_i\}$, where i = 1, 2, …, n, and n is the number of gridded points. Let the EGI and GO of each point be $egi_i$ and $go_i$, calculated as follows:

Step 1: Calculate the absolute values of $dei_i$ and $go_i$ and normalize[75,76] them into [0.1, 0.9].

Step 2: Combine the normalized $dei_i$ and $go_i$ with the weighted geometric averaging (WGA) operator[80] as:

$$egi_i = dei_i{}^{\alpha} go_i{}^{1-\alpha} \tag{1}$$

where $\alpha \in [0, 1]$ is the weight of normalized $dei_i$, i = 1, 2, …, n.

Step 3: The optimal value of $\alpha$ can be determined with the following steps:

Step 3.1: Take the natural logarithm of Eq. (1):

$$ln\, egi_i = \alpha * ln\, dei_i + (1 - \alpha) ln\, go_i, i = 1, 2, \cdots, n \tag{2}$$

Obviously, $ln\, dei_i < 0$ and $ln\, go_i < 0$

Step 3.2: Assuming $a, b > 0$, there exists a theorem such that:

$$\frac{a}{b} + \frac{b}{a} - 2 = \frac{(a-b)^2}{ab} \geq 0 \tag{3}$$

The equality in Eq. (3) holds only if $a = b$.

To balance the indices of $dei_i$ and $go_i$, we minimized the total deviation between $egi_i$ and $dei_i$, $go_i$. According to Eq. (3) and Zhou et al.[81], the deviations $d_i$ and $t_i$ are formulated as follows:

$$d_i = \frac{-ln\, egi_i}{-ln\, dei_i} + \frac{-ln\, dei_i}{-ln\, egi_i} - 2 \tag{4}$$

$$t_i = \frac{-ln\, egi_i}{-ln\, go_i} + \frac{-ln\, go_i}{-ln\, egi_i} - 2 \tag{5}$$

Hence, the minimized model $(M - 1)$ can be expressed as:

$$(M - 1) : \min Y = \sum_{i=1}^{n} (d_i + t_i)$$
$$= \sum_{i=1}^{n} \left[ \left( \frac{-\ln egi_i}{-\ln dei_i} + \frac{-\ln dei_i}{-\ln egi_i} - 2 \right) + \left( \frac{-\ln egi_i}{-\ln go_i} + \frac{-\ln go_i}{-\ln egi_i} - 2 \right) \right] \quad (6)$$

Based on Eq. (3), the minimized model $(M - 1)$ can be equivalently written as:

$$(M - 2) : \min Y = \sum_{i=1}^{n} \left[ \frac{\alpha \ln dei_i + (1 - \alpha)\ln go_i}{\ln dei_i} + \frac{\ln dei_i}{\alpha \ln dei_i + (1 - \alpha)\ln go_i} \right.$$
$$\left. + \frac{\alpha \ln dei_i + (1 - \alpha)\ln go_i}{\ln go_i} + \frac{\ln go_i}{\alpha \ln dei_i + (1 - \alpha)\ln go_i} - 4 \right] \quad (7)$$

Step 3.3: The Artificial bee colony (ABC) algorithm[82] is used to solve model $(M - 2)$, which is a fractional programming problem where $Y$ is the objective function and $\alpha$ is the independent variable. The parameters are set as: population size = 20, number of iterations = 30, and *limit* = 5. The convergence curve of the ABC algorithm for solving the above model is plotted in Supplementary Fig. 6. The algorithm converged at the sixth iteration with the optimal estimate $\alpha = 0.5046$ (Supplementary Fig. 7). Thus, we get $egi_i = dei_i^{0.5046} go_i^{0.4954}$. The resulting EGI values were then mapped with ArcGIS 10.2 to show its geographical distribution in the EI-decreased regions under the projected future climate scenario (Fig. 2c). It is noteworthy that this analysis used high-quality genomic data with individuals collected from 75 sampling sites across a wide range of eco-climate regions worldwide. This allowed us to estimate the population-level genetic variation in response to climate change. However, we currently do not have the site-specific physiological DBM data available for calculation of the eco-climate index. Future detailed studies on physiological variation in different populations will further improve our prediction of eco-genetic adaptation to climate change.

## Genetic basis of climate adaption

*Gene expression analysis by RT-qPCR.* The wild-type strain of DBM, Geneva (G88), was used in this assay. This G88 strain was collected from the New York State Agricultural Experiment Station in 1988 and has since been maintained on artificial diet without exposure to insecticides[83]. It was provided by Dr. Antony M. Shelton (Cornell University, USA) to the Institute of Applied Ecology, Fujian Agriculture and Forestry University in 2016. Since then, we maintained this strain on artificial diet without exposure to insecticides at 26 °C that is a favorable temperature to rear and maintain this wild-type strain.

We used nine temperature treatments and one control at 26 °C. A male or female individual was placed in a 1.5 ml plastic vial (4.0 cm height) with a pinhole in the side wall to allow air exchange. Before treatments, all vials with DBM were placed into the incubator at 26 °C. A group of thirty vials (15 vials with DBM females and 15 with males) were frozen in liquid nitrogen and used as controls. Additional groups of thirty vials (15 containing females, 15 containing males) were exposed to each of nine distinct temperature treatments. These treatments were defined from a previous study on lethal temperature limits of DBM[84] and our preliminary experiments: three high-temperature treatments: (1) H1: 40 °C for 30 min, (2) H2: 43 °C for 30 min, (3) H3: 43 °C for 30 min with 24 h of recovery at 26 °C; and six low-temperature treatments: (1) L1: −14 °C for 30 min, (2) L2: −14 °C for 30 min with 24 h of recovery at 26 °C, (3) L3: −17 °C for 30 min, (4) L4: −17 °C for 30 min with 24 h of recovery at 26 °C, (5) L5: −20 °C for 15 min, and (6) L6: −20 °C for 15 min with 24 h recovery at 26 °C (Fig. 3). High and low temperature treatments were conducted in incubators and freezers, respectively. The exposure duration of moths at −20 °C was set for 15 min because moths started to die when exposed to −20 °C for over 20 min. After each of the treatments, moths were immediately frozen in liquid nitrogen. The thirty moths from each treatment were grouped into three replicates of 5 females and 5 males in tubes of 1.5 ml each (in total, six tubes each containing 5 individuals of same sex for each treatment including controls). All tubes with frozen moth samples were stored at −80 °C before RNA extraction.

Total RNA was extracted using Eastep® Super Total RNA Extraction kit (LS1040, Promega, USA), following the manufacturer's protocol. A NanoDrop Spectrophotometer (ND2000, Thermo Scientific, USA) and 2% agarose gel electrophoresis were used to determine the quality and quantity of RNA. cDNA was synthesized with 1 µg of total RNA using GoScript™ Reverse Transcription System (A5001, Promega Corporation, USA). RT-qPCR was carried out using the target gene-specific primer (forward: 5′-AACCCCCCCTTCATCCAAG-3′, reverse: 5′-CTGCTGAGGCTGTAGGTCATG-3′), which was designed by Oligo 7 and the reference gene primer for normalization (forward: 5′-CAATCAGGCCAATTTACC GC-3′, reverse: 5′-CTGGGGTTTACGCCAGTTACG-3′) according to the previous reports[85]. qRT-PCR was conducted with 2 µL cDNA, 0.4 µL of each primer, 0.15 µL CXR Reference Dye, 7.05 µL DEPC water and 10 µL SYBR Green Supermix (A6001, Promega Corporation, USA) in a 20 µL total reaction mixture using QuantStudio™ 6 Flex real-time PCR system (ThermoFisher Scientific, USA). PCR conditions were set as follows: 10 min at 95 °C followed by 40 cycles of 15 s at 95 °C, 30 s at 60 °C and then 15 s at 95 °C, 1 min at 60 °C, 15 s at 95 °C for a melt curve. The relative expression level of *PxCad* in each treatment was normalized to the abundance of samples under control temperature, using the $2^{-\Delta\Delta Ct}$ method[86]. In the present study, we focused on the comparison of difference in gene expression between each of the temperature treatments and the control rather than the comparison between different non-control temperature treatments. Therefore,

we used independent t-tests to perform in R to establish differences in gene expression between each temperature treatment and the control.

*CRISPR/Cas9-based genome editing.* The wild-type strain of G88 was used for functional validation of *PxCad*. We selected two sgRNA target sites, 5′-AGGGAGA TGTCATCATTGCA-3′ and 5′-ACTCGTCCATGTAGATGTTG-3′, in *PxCad* exon 3 according to the principle of 5′-N20NGG−3′ (with the PAM sequence underlined). To obtain the templates for in vitro transcription of the two sgRNAs, we performed PCR with two primer pairs (Supplementary Table 7) using the KOD-Plus-Neo Kit (TOYOBO, Osaka, Japan), and then purified the products using the Gel Extraction Kit (Omega, Morgan Hill, GA, USA). In vitro transcription was then performed to generate two sgRNAs using the HiScribe™ T7 Quick High Yield RNA Synthesis Kit (New England Biolabs, Ipswich, MA, USA) and Cas9 mRNA using the HiScribe™ T7 ARCA mRNA Kit (with tailing) (New England Biolabs, Ipswich, MA, USA) using linearized PTD-T7-Cas9 vector as a template[87]. The synthesized sgRNAs and Cas9 mRNA were further purified using phenol–chloroform extraction and ethanol precipitation and stored at −80 °C until use.

Fresh eggs from G88 moths laid within 15–20 min were injected with a mixture of Cas9 mRNA (380 ng/µl) and sgRNAs (100 ng/µl) using an IM 300 Microinjector (Narishige, Tokyo, Japan). Microinjected eggs were then incubated at 26 °C and allowed to develop to adult (G0). Virgin G0 adults were crossed with the wild-type G88 strain in single pairs to produce the G1 progeny. After oviposition, we sacrificed G0 adults and extracted the genomic DNA using the Tissue DNA Kit (Omega, Morgan Hill, GA, USA). gDNA was then used to amplify the fragment containing the sgRNA target sites using PCR with the Phanta Max Super-Fidelity DNA Polymerase (Vazyme, Nanjing, China) and a primer pair (Supplementary Table 7). The generated PCR products were Sanger sequenced by the Biosune Biotech Company (Fuzhou, China) to examine the mutation in the target sequence region. After mutation identification, we focused on G1 progeny derived from mutated G0 parents. Retained G1 siblings were crossed in single pairs to generate G2 progeny and progeny of heterozygous G1 parents were kept after molecular identification. Similarly, we performed single-pair crosses between G2 siblings and kept only G3 from homozygous mutant G2 parents. Retained G3 individuals were pooled to establish the *PxCad* knockout strain (MU, G88-Cad).

To verify the absence of PxCad protein in the *PxCad* knockout strain, we isolated midgut brush border membrane vesicle (BBMV) proteins from the G88 strain and knockout strain using the differential magnesium precipitation method[88]. Total extracted BBMV proteins (30 µg) were separated on 7.5% SDS-PAGE, and the regions (~120–250 kDa) predicted to contain PxCad proteins were excised from the gel stained with Coomassie blue. These gel slices were subjected to tryptic digestion and nano-LC-MS/MS analysis at Huada Protein Research Center (Shenzhen, China).

*Bioassay: behavioral responses to different temperatures.* DBM strains used in this assay were wild-type (G88 or WT) and mutant (G88-Cad or MU) adults. Female or male individuals emerging within 24 h were placed into a 1.5 ml clear plastic vial (4.0 cm height) with a pinhole in the side wall. Twenty vials with females and twenty with males were then put into a plastic container (16.5 cm length, 11.5 cm width, 3.5 cm height).

Four plastic containers with DBM adults were placed into a climate incubator (DRX-400-DG, Ningbo Saifu Experimental Instrument CO., LTD, China) where DBM adults were exposed to one of the following heat shock treatments: 40 °C, 41 °C, 42 °C, 43 °C, and 44 °C, for 2 h. After each heat shock treatment, the plastic containers were removed from the climate incubator and placed in a temperature-controlled room at 26 °C for 24 h, after which survival was recorded.

A similar procedure was used for cold stress, with four plastic containers of DBM adults being placed into a freezer. Three temperature treatments were used: −14 °C, −17 °C, and −20 °C. When the temperature was set at −14 °C, the adults were exposed for periods of 150, 210, 270, and 330 min. At −17 °C, durations were 40, 60, 80, 100, and 120 min. At −20 °C, durations were 20, 25, 30 and 35 min. After cold treatments, adults were transferred to a temperature-controlled room at 26 °C for 24 h, after which survival was recorded.

## Statistical analysis.
Logistic regression models were fitted to the survival data to test behavioral responses to different temperatures. Results of the cold-exposure experiment were analyzed first using the model: $\log[p/(1-p)] = a_{ijk} + b_{ijk} t$ where $p$ is the probability of survival, $i$ is the temperature index (−14, −17, or −20 °C), $j$ is the strain index (WT or MU), $k$ is sex (male or female), and $t$ is exposure duration (min) used as a continuous predictor. The responses were complex, with significant interactions to the third order. Therefore, to allow clearer interpretation of the results, a separate analysis was performed for each temperature using the simpler model $\log[p/(1 - p)] = a_{jk} + b_{jk} t$.

The results of the heat-shock experiment were analyzed using the model $\log[p/(1 - p)] = a_{jk} + b_{jk} T$ where $j$ is the strain (WT or MU), $k$ is sex (male or female), and $T$ is temperature (°C) used as a continuous predictor.

## Reporting summary.
Further information on research design is available in the Nature Research Reporting Summary linked to this article.

## Data availability
Raw reads of all 372 sequenced individuals used in this study have been deposited in the CNSA (https://db.cngb.org/cnsa/) of CNGBdb with the accession code CNP0000018, and

been synchronously deposited in the EMBL Nucleotide Sequence Database (ENA) (https://www.ebi.ac.uk/ena) with the accession code PRJEB24034.

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

## Acknowledgements

We are very grateful to many researchers and volunteers for their kind help with collection of the DBM specimens worldwide. This work was financially supported by the National Natural Science Foundation of China (No. 31972271 granted to S.Y., No. 31320103922 granted to M.Y., and No. 31230061 granted to M.Y.), Scientific Research Foundation of Graduate School of Fujian Agriculture and Forestry University (No. 324-1122yb058 granted to Y.C.), State Key Laboratory of Ecological Pest Control for Fujian and Taiwan Crops, Joint International Research Laboratory of Ecological Pest Control, Fujian-Taiwan Joint Innovation Centre for Ecological Control of Crop Pests, International science and technology cooperation and exchange program of FAFU (KXb16014A granted to M.Y.), the Thousand Talents Program and the "111" Program in China.

## Author contributions

M.Y., S.Y., and G.M.G. coordinated the project. Y.C., S.Y., and M.Y. conceived, designed, and managed the project. Y.C., Z.L., S.C., Jianyu Li, and J.H. performed the experiments. Y.C., Z.L., J.R., Jian Lin, S.H., and F.K. generated and/or analyzed the data. Y.C., S.Y., and M.Y. conceived and structured the manuscript. Y.C., S.Y., M.Y., J.R., L.V., and G.M.G. interpreted the results, co-wrote and/or revised the manuscript.

## Competing interests

The authors declare no competing interests.
