## [Peer Review File · Nature Communications]

Reviewer comments, initial round - -

Reviewer #1 (Remarks to the Author):

I have reviewed the manuscript, "Large-scale genome wide study reveals climate adaptive variability in a cosmopolitan herbivore" by Chen and coauthors. This paper uses pan global genomic samples of the diamond back moth (DBM), *Plutella xylostella*, to ask questions about global distribution and adaptability to climate change. From a bank of over 200,000 SNPs spanning the DBM genome, the authors eventually narrow down to 97 SNPs that are putatively involved in adaptation to climatic variables, namely temperature. Several of these SNPs are clustered within PxCad, a cadherin-like protein. The authors delved deeper into a potential role for PxCad in climate adaptation by undertaking manipulative approaches (namely CRISPR/cas9 site-directed mutagenesis). The manuscript represents a very broad comprehensive assessment of adaptation to climate change in a species that is both economically valuable (considerable crop pest) and cosmopolitan (wide distribution across all continents excluding Antarctica).

Overall I feel there is some potential with this manuscript. However, I feel that at this point it is far too underdeveloped both from a written, methodological, and analytical perspective, to be considered for publication at this time. This paper needs considerable refinement, revision, and experimentation. I outline my criticisms below.

The section on habitat suitability is particularly interesting. Many researchers that investigate connectivity in landscapes often consider land-use change and its effects on habitat suitability. Connecting adaptive potential, habitat suitability and climate change effects are exciting.

I do not have the analytical expertise to confidently assess the global scale associations between climate variables and genomic variation. The SNP assessment with respect to climate seems to be using the right tools and methodology to address their questions.

However, I am qualified to assess the sections where the authors delve deeper into one particular candidate gene, PxCad. Therefore, I will focus the majority of my criticisms toward this aspect of the manuscript.

Narrowing down the SNPs

Starting on line 233, the authors state "...we excluded samples from regions that are only seasonally suitable for growth of DBM with an ecoclimatic index (EI=0) because populations are seasonally inhibited in these regions and are unlikely to receive perennial selection by local environmental variables." What was the rationale for this decision? You are potentially missing some of the most important regions with respect to climate adaptation. Presumably these regions represent the northernmost edges of DBM's range. Genetic variation in these regions might actually be where the most important adaptations are occurring as the species expands its range and also might represent where temperature changes are most impactful.

Effects of temperature on gene expression.

I cannot understand the authors' methodology for the temperature treatments. The way I read it I interpret as:

- Groups of five caterpillars (combination of males and females) are assigned to one of three treatments: 1) control (26°C), 2) Low temperature, 3) High temperature.
- Each treatment is repeated 5 times (75 total caterpillars, haphazard mix of males and females spread across all treatments).

I have questions about the temperature treatments and the stresses encountered by the caterpillars. What happens to the control caterpillars? Are they just left in the incubator while the others are handled, moved, stressed. Therefore, I find it difficult to disentangle the effects of

temperature from stress. Although temperature is itself imposing a stress, I do not feel the treatments can conclude that temperature is the cause of the changes in gene expression.

It seems that the treatments are quite unbalanced. The High temperature treatment was "40°C for 30 min, 43°C for 30 min, 43°C for 30 min with 24 h of recovery." What is the rationale for these temperatures? The Low temperature treatment is even more confusing: "-14°C for 30 min, -17°C for 30 min, -20°C for 15 min, -14°C for 30 min with 24 h of recovery at 26°C, -17°C for 30 min with 24 h of recovery at 26°C, and -20°C for 15 min with 24 h of recovery at 26°C" ? Why so many different temperatures, different exposure times?

Gene expression by RT-qPCR

"After treatments, moths were frozen in liquid nitrogen and stored at -80°C" Is there a rationale to look at cadherin gene expression after a series of temperature treatments followed by periods of recovery. Gene expression is transient. Caterpillars should have been frozen during the exposure to look at the acute effects. I feel this authors have likely missed important information on gene expression differences in response to extreme temperatures.

The display of the resultant data (Figure 3) is extremely confusing. Males are on the left, females on the right. Why are there different colours. What does the x-axis represent, are these paired comparisons? What is 43 compared to 43R? Where is the data for the wildtype (WT) and the mutant (MU)? What are the t-tests comparing? I'm sorry but I cannot make any rational interpretation of the data displayed in this figure.

The susceptible wildtype G88 strain

What is this strain? How was it collected? Is it inbred, isogenic? Why is it considered susceptible. I'm sorry but I find myself with 100 questions for every answer. This needs to be tight and it needs to be clear.

Site-directed mutagenesis in PxCad via CRISPR/Cas9

On line 190 the authors state : "None of these peptides were detected in the G88-Cad strain, confirming..." However, when I look at Fig 4b I cannot see a difference between 1: G88 and 2: G88-Cad (the mutant) in the SDS Page gel. Presumably there should be differences in the banding patterns to show that the mutant strain is not making the 19 peptides in the 120-250 kda range encoded by PxCad.

PxCad – Cadherins and temperature

At no point in this manuscript is a conceptual link between cadherins and temperature adaptation explored, discussed, or even proposed. Canherin proteins, in my mind, would not have been a candidate gene for climate change adaptation. Therefore, it appears, at least as it is written, that the authors aggressively pursued this gene without thinking about what it does, why it might respond to temperature, and why genetic variation would exist.

What do the SNPs in PxCad do? Aren't they more important than creating a mutant?

The authors zero in on PxCad due to the convergence of significant associations and several SNPs? What do these SNPs do? Several are located the coding region? I wonder why the authors chose to dive straight into genetic manipulation without exploring the important standing genetic variation in PxCad and its role in climactic adaptation. This represents a far more natural scenario than creating a mutant that knocks out cadherin function. Furthermore, this may represent the functional adaptations themselves! Why not asses temperature adaptation (and several other variables) in DBM collected from different temperatures? You already have candidate SNPs within your candidate gene to look.

Is this really temperature?

I seriously question whether the genetic variation detected among the populations in this study could be due to demographic effects. For example, in *Drosophila melanogaster*, there is a strong collinearity between thermal clines and ancestry making it particularly challenging to disentangle demographic signals from true adaptive signals. For example see Excoffier et al. 2009 *Annu Rev Ecol Syst* 40: 481, Duchon et al. 2013 *Genetics* 193: 291, Bergland et al. 2016 *PLoS Genet* 10: e1004775, Flatt 2016 *Mol Ecol* 25: 1023.

Non-genetic inheritance?

I think the authors need to at least discuss the possibility that non genetic inheritance could play an important part in organisms adapting to climate change. Epigenetic effects can respond much quicker to environmental perturbations than genetic inheritance. In return, because such effects erode much quicker than genetic adaptation it is a major challenge to infer how DBM have responded to climate change in the past. The authors use genetic offset to infer the full adaptive potential for this species. They might not be able to address this experimentally, but they have to at least acknowledge that there is potentially an additional major side to how DBMs and other species might respond adaptively to climate change.

The written document

Overall the manuscript is quite poorly written. The introduction is very confusing. Several statements are discussed without proper set up. For example, the topic sentence claims that human impact influences global temperature and levels of precipitation. This statement is not very impactful as it is not backed up, nor is it surprising since it is general knowledge. The second sentence provides a new statement that claims that adapting to changes directly influence survival etc. The next sentence provides a third statement that claims adaptive capacity is important to mitigate environmental perturbations. Adaptive capacity is not explained or defined, and nonuniformity is also not defined/explained. Next authors switch to adaptive plasticity, again without explaining or making it explicit what they mean by adaptive plasticity. The last sentence in the paragraph is actually the most impactful statement and the one they should start with and should provide evidence for.

The authors do not develop the idea for using an insect pest in the context of climate change. They need to do a better job of setting up the connection between climate change adaptation and pest/invasive species.

The discussion is far too short, and completely ineffective. It is one very short paragraph that fails to place their study in the context of climate change adaptation, fails to discuss why PxCad, and completely fails to convince the reader (if they aren't already unconvinced).

Minor comments:

Lines 24-25: "Efforts to investigate these responses have largely neglected intraspecific variation arising from local adaptation"

I suggest avoiding pointing out what hasn't been done or what has been neglected. Instead point out what you are adding and why that is important.

Lines 54-55: "These attributes make it an excellent..."

Which attributes? Why does this make it an excellent system? Why does the fact that it is a pest make it a good system for climate change?

Line 59: Define 'genetic offset' and 'ecoclimatic index' here since the definitions come far too late in the manuscript

Line 100: "gene-environment relationships"

Although I acknowledge these are important with respect to genetic offset, I think you need to

define these a bit more since it may not be intuitive to the reader.

Line 136: "genetically adaptive potential"

What does this mean?

Line 173: "locomotory behaviour"

This gene was not shown to affect behaviour, rather mutations in this gene affect hearing and deaf animals show differences in behaviour. These differences are due to a loss of hearing not due to a loss of cadherins.

Line 175-176: "...a susceptible strain of DBM, G88..."

What makes this strain susceptible? How was this determined? Did you determine this? If so then where are the methods, if not then where is the reference?

Line 178: "-14oC, -17oC and -20oC"

Why were these temperatures chosen along with the high temperatures. What is the rationale?

Line 378: "with a pinhole in the side wall"

Why is this done? What is the purpose? How are the caterpillars raised in the lab? Is the pin hole to allow air exchange? This needs more explanation.

Line 490: publication year is 2015 not 2014

Figure 2: The light blue is very difficult to see, especially since many spots are very small. This will be even hard to see in the published version. Suggest considering alternatives.

Figure 4: "Twenty individuals were used for each of the treatments with four replicates."

This means 4 replicates of 20. Do you mean to say 4 replicates of 5 (N = 20)?

Very minor comments:

Line 46: "...adaptation at the local scale is therefore important ..."

I suggest adding 'of cosmopolitan species' after scale

Line 83-84: "We used the gradient forest, a machine-learning regression approach, to examine climate-mediated..."

Suggest rewording: 'We used a machine-learning regression approach, gradient forest, to examine climate-mediated...'

Line 107: "...tolerant to future climate change thus likely to remain damaging..."

Suggested edit: '...tolerant to future climate change making them likely to remain damaging...'

Line 131: "...the "eco-genetic index" (EGI). This combines the genetic..."

Suggested edit: '...the "eco-genetic index" (EGI) which combines the genetic...'

Line 148: "...coding sequence (CDS)..."

Why the acronym? You only use it once in the entire manuscript. Suggest removing.

Line 149-150: "These genes cover a wide range of functions..."

Suggest: 'These genes are predicted to cover a wide range of functions...'

Is this genome annotated. Has gene function been confirmed?

Line 172: "PxCad, a cadherin-like protein, is known to be involved..."

Suggested edit: 'Homologs of PxCad encode cadherin-like proteins known to be involved ...'

Lines 215-216: "...we verify that a specific gene representing a temperature-sensitive responder to climate change..."

Awkward wording

Line 231: *Arabidopsis thaliana* should be italicized

Line 285: "Gradient forest (GF) is originally a community-level model..."

Suggested edit: 'Gradient forest (GF) was originally created as a community-level model...'

Line 337-338: "...the hot-wet stress temperature threshold (TTHW)..."

Another unnecessary acronym. Moreover, I think it should be HWTT not TTHW.

Line 341: calculated should be changed to calculate

Line 355: "EGI = DEI x GON"

Why the subscript N? N is not defined. Does this simply mean it is indexed to sample size?

Line 435: "...in the PxCad knockout strain"

PxCad should be italicized

Reviewer #2 (Remarks to the Author):

This work combines approaches from genomics, climate variability and change, and genome editing to explore adaptive variation in *Plutella xylostella*. I really like the holistic way in which this has been done. The result is a clear demonstration of genomic variation associated with climate plus a key gene which likely plays a role in cold adaptation in the species (and also maybe heat response).

My major concern has to do with some of the main conclusions in the paper. For example, that the "results demonstrate that *P. xylostella* is genetically capable to tolerate future climate change and will likely maintain its global pest status past 2050 and even 2080". This doesn't seem very novel to me. Physiological study of DBM has already suggested the species will tolerate future warming, for example:

Nguyen C, Bahar MH, Baker G, Andrew NR (2014) Thermal tolerance limits of diamondback moth in ramping and plunging assays. *PLoS One*, 9, e87535.

The results here then confirm these expectations and results. What is it specifically about the genomic associations that add to this conclusion?

Also, in the abstract (and elsewhere in the manuscript) there is a suggestion that these results could guide management of the species. How could this be implemented?

Another major concern I have is in the use of different scenarios in the climate change modelling (see details below). It's hard to interpret the results given the outdated emissions scenarios used. I'm not sure the result would change qualitatively if other scenarios were changed for the reason outlined above on physiology of the study. Still, as the spatial patterns are key, it's important to use the most up to date climatic data to both demonstrate the utility of the approach and present the most accurate results.

Title – would “cosmopolitan pest” or “cosmopolitan insect pest” be more descriptive?

Lines 27-28 – this is somewhat circular logic, or anyway isn't really a specific result, more of a description of the approach

Lines 70-71, 234-236 – why subset the sampling in this way? I would guess that such sites would either yield interesting results with respect to selection or would be undifferentiated from other sites.

Lines 105-106, 319-323 – why use A1B and A2? These are very old and outdated scenarios

Line 216 – specifically what is verified is that a specific gene responds to climatic variation spatially, not specifically to climate change. This may have climate change implications but need to be careful about wording.

Line 253 – need to cite worldclim

Fick, S.E. and R.J. Hijmans, 2017. WorldClim 2: new 1km spatial resolution climate surfaces for global land areas. *International Journal of Climatology* 37: 4302-4315.

Line 336 – why use CSIRO GCM?

Line 328 – “tolerant to future climate change”

Fig 2 – what does the white in the figures represent? I'm also having difficulty with the scales. In the text it is noted that “relatively high levels of EGI” are need for habitat suitability. Relative to what? What's a high level of EGI?

Reviewer #3 (Remarks to the Author):

I have reviewed the manuscript entitled: “Large-scale genome-wide study reveals climate adaptive variability in a cosmopolitan herbivore”. The authors used a multi-pronged approach to study adaptive variation in the diamondback moth. They have a SNP dataset from 357 individuals in 78 sampling sites distributed across the globe. They then asked the questions whether there are significant SNP-environment associations and how local allele frequencies of significantly associated SNPs should change under future climate conditions. To answer these questions, the authors employed a combination of landscape genomics, predictions of habitat suitability, and gene-editing techniques to confirm the function of genes that SNPs were located in. The results suggest that there are > 3000 SNPs significantly associated to temperature variables, and that the genetic vulnerability of many populations is likely low enough for the species to thrive and maintain its status as a pest species.

The manuscript is generally well written. Although I can't comment on the genome editing work, I was particularly pleased to see the confirmation that indeed there is a temperature-sensitive response. However, I do have some reservations with the landscape genomics work.

First, some of the bioclimatic variables are highly correlated, and I was surprised that the authors did not make any effort to downselect a subset out of the 19 variables. There are several ways to go about this, based on for instance variance inflation factors or just cross-correlation coefficients. In addition, surely climate variables are not the only factors influencing adaptive variation in these moths. Some of the factors will be hard to capture, but others may be readily available as GIS layers. The focus on future climate change dictates that only climate variables can be included in a model under current climate conditions, but the conclusion that temperature is the most important factor in determining climate-associated genomic variation in this moth (line 207-208) seems rather overstated. Many other drivers have not been taken into account, and of those that were, the majority is temperature-related.

Second, the future climate scenarios A1B, A2 etc are very outdated. These were used in the CMIP3 / IPCC AR4 scenarios published in 2007. In the meantime, Representative Concentration Pathways have substituted the A-scenarios, and as a matter of fact even the RCP are by now obsolete and replaced by CMIP6 Shared Socioeconomic Pathways for IPCC's AR6. Although I don't think the overall conclusions will change, the SSPs generally predict larger temperature changes (compared to the RCPs), which could change the maps at least to some extent. The bioclimatic variables for a variety of SSP scenarios are available for download from WorldClim. I would suggest updating the maps for a moderate and a more severe SSP scenario.

Third, I'm unsure about the conceptual approach taken by combining predicted changes in habitat suitability with those in allele frequencies. Both predictions of habitat suitability and of the genomic makeup of populations in the future assume that current relationship between environment and allele frequencies or between environment and species occurrence remain constant over time. If,

for instance, temperature changes in a given area are predicted to be severe, the allele frequencies of temperature-sensitive genes are also predicted to change dramatically. At the same time, habitat suitability may change drastically, but suitability and local adaptations are highly related. If habitat becomes less suitable, considerable allele frequency changes are also expected to be required. Thus, simply multiplying the two responses to come up with a new 'eco-genetic' index appears to be cutting a lot of corners, and does not improve our understanding of the vulnerability or persistence of the species in the future. Identifying areas where conditions in the future are not mirrored in any of the sites where the species was sampled may provide a more realistic idea of where populations are at risk.

Fourth, the authors used gradient forests (GF) to create maps of genetic variation under current and future climate conditions, and to compute a map representing 'genetic vulnerability'. A major caveat of GF in landscape genomic approaches (i.e. analyses of population structure in a spatial context) is that it cannot account for the effects of isolation by distance or isolation by resistance. Yet, teasing apart the effects of neutral processes from those of selection is a crucial component in landscape genomics, and key to the current paper. By focusing on adaptive genetic variation and genetic vulnerability, the authors are specifically interested in that part of genetic variation that cannot be attributed to neutral divergence between populations. To this end, generalized dissimilarity modeling is a more appropriate method, because it can directly consider the effects of distance and variation in habitat permeability (see also Box 1 in Fitzpatrick and Keller, 2015). I realize that the authors only included the SNPs identified in Bayenv/Sambada/LFMM, thus focusing on those SNPs that are putatively under selection, at least partly mitigating the problem. Yet, GF starts all over again with these SNPs, not accounting for neutral processes that very likely also contribute to population divergence.

Minor comments

l. 40 - 'increased' -> the effects could go both ways, so why focus on an increase only?

l. 44 - 'adaptive plasticity' -> although the level of plasticity can be adaptive by itself, I think that's not what the authors mean to say here. Rather 'adaptive potential'.

l. 229 - Drop one of the occurrences of 'on average'

l. 231 - '*Arabidopsis thaliana*' in italics

l. 270 - What's the reason for the low threshold for p after FDR correction? Or does the mentioned p-value correspond to a value before correction?

l. 315 - 'disruption of gene-environment relationships' -> As a matter of fact, the major assumption of the approach being used is that gene-environment relationships will remain the same in the future. Rather, the spatial patterns will change because of climate change. If gene-environment relationships would be disrupted, there is no way that we can make any predictions of the spatial distribution of alleles/genetic variation.

l. 334 - 'distribution' -> 'distributions'

Supplements page 11 - add captions for Tables 6, 8, 9, 10

Reviewer #4 (Remarks to the Author):

Chen et al present a manuscript investigating climate adaptation and the potential to adapt to future environments in a worldwide dataset of the herbivore *Plutella xylostella* (diamondback moth). This study represents a statistical re-analysis of a genomic dataset published in Nature Communications by the same team earlier this year. The authors go beyond a simple scan for adaptive variation in two ways: they use the results of the adaptation analysis to predict the potential of dealing with future climatic conditions and they verify one of the top genes using genome editing. This is an interesting approach and distinguishes this study from many other studies of local adaptation. The main conclusion of this study is that the diamondback moth will maintain its global pest status at least until 2050.

I am not an expert on pests, conservation or *Plutella xylostella* specifically so my comments are from a general population genomics perspective. I am convinced that this article will ultimately become a valuable addition to the record but I do not feel like I have the overview over the specific area to make a statement about the potential impact.

The main part of the manuscript is relatively short and it quickly becomes clear that this is mainly a re-analysis of data that the authors presented in another paper. For a lot of the details about the

underlying dataset, the reader will have to refer to another publication. It is generally positive to re-use published data but some descriptive information would be useful to the readers of this article. For example, the geographic distribution of populations from Figure S1 could easily be included in the main figures. How many locations and individuals were in the final dataset etc. Sometimes the authors could add more information in the Results section, e.g. when they are introducing the different indices and metrics used, otherwise the reviewer has to jump between different sections of the manuscript.

What is crucial to the conclusions is the solid identification of candidate loci. The authors are using three different methods to identify these loci but they include all candidates in their downstream analysis. In my impression, the authors apply different stringency criteria on the different approaches. Using hard thresholds on each of these three very different statistics is difficult to interpret and, consequently, the number of candidates identified per method differ by a factor of almost two. Approaches like LFMM and Bayenv2 which are using a user-defined number of iterations can lead to somewhat noisy results in local optima, especially in situations of small sample sizes (like the <5 individuals per population on average in this case which makes frequency estimates quite noisy). For LFMM, the authors use the median of five independent runs which should account for some of the noise. For Bayenv2, however, they only use a single run with a comparatively low number of iterations (I have usually seen at least an order of magnitude more iterations in the literature).

I am not sure if fully understand the motivation for restricting the GF analysis to the candidate SNPs, but I also have not read the original publication on that. The candidates were identified by association between allele frequency and environmental variables. The GF analysis now again identifies between the allele frequencies at these SNPs and environmental variables which feels circular. How would the results differ if all SNPs were tested? Would the same environmental variables be the driving forces behind neutral population structure?

I do not feel qualified to comment on the methodology of the CRISPR/Cas9 genome editing approach. I am only wondering why we do not see any control populations (comparing WT and mutant without heat or cold stress) in the figures (Fig 5 and S8). According to the methods section, there was such a control group.

Minor comments:

L61: I think you are only functionally testing a single gene not "genes".

L151-154: If there is more literature showing temperature related functions for these genes, please add the references to the text or Table S7.

L164: Does "signals of adaptive variation" mean "presence of the selected allele"? Wouldn't that be a bit circular since the exact same information was included in the scan for adaptive variants so maybe the absence of the variant actually contributes to the worldwide signal of adaptation.

L177/178: Change "the PxCad gene tended..." to "the expression of the PxCad gene..."?

Fig S8: Please display standard errors in this figure

Tables S3-S5: Please add the values of the scores underlying these candidates to the tables

Responses to the Comments from Reviewers

Reviewer #1 (Remarks to the Author):

I have reviewed the manuscript, “Large-scale genome wide study reveals climate adaptive variability in a cosmopolitan herbivore” by Chen and coauthors. This paper uses pan global genomic samples of the diamond back moth (DBM), *Plutella xylostella*, to ask questions about global distribution and adaptability to climate change. From a bank of over 200,000 SNPs spanning the DBM genome, the authors eventually narrow down to 97 SNPs that are putatively involved in adaptation to climatic variables, namely temperature. Several of these SNPs are clustered within PxCad, a cadherin-like protein. The authors delved deeper into a potential role for PxCad in climate adaptation by undertaking manipulative approaches (namely CRISPR/cas9 site-directed mutagenesis). The manuscript represents a very broad comprehensive assessment of adaptation to climate change in a species that is both economically valuable (considerable crop pest) and cosmopolitan (wide distribution across all continents excluding Antarctica).

Overall I feel there is some potential with this manuscript. However, I feel that at this point it is far too underdeveloped both from a written, methodological, and analytical perspective, to be considered for publication at this time. This paper needs considerable refinement, revision, and experimentation. I outline my criticisms below.

The section on habitat suitability is particularly interesting. Many researchers that investigate connectivity in landscapes often consider land-use change and its effects on habitat suitability. Connecting adaptive potential, habitat suitability and climate change effects are exciting.

I do not have the analytical expertise to confidently assess the global scale associations between climate variables and genomic variation. The SNP assessment with respect to climate seems to be use the right tools and methodology to address their questions.

However, I am qualified to assess the sections where the authors delve deeper into one particular candidate gene, PxCad. Therefore, I will focus the majority of my criticisms toward this aspect of the manuscript.

Narrowing down the SNPs

Starting on line 233, the authors state “...we excluded samples from regions that are only seasonally suitable for growth of DBM with an ecoclimatic index (EI=0) because populations are seasonally inhibited in these regions and are unlikely to receive

perennial selection by local environmental variables.” What was the rationale for this decision? You are potentially missing some of the most important regions with respect to climate adaptation. Presumably these regions represent the northernmost edges of DBM’s range. Genetic variation in these regions might actually be where the most important adaptations are occurring as the species expands its range and also might represent where temperature changes are most impactful.

Adaptive evolution is a biological process that describes how organisms change in response to perennial selection pressure over successive generations (Sejian et al., 2015). As mentioned in the revised Methods, populations from regions that are seasonally suitable for growth of DBM do not receive perennially unpunctuated selection by local environmental variables, so genetic variation in these regions cannot be continuously passed on to future generations over years. In other words, for DBM, regions that are only seasonally suitable for its growth and development (i.e., with an ecoclimatic index, $EI = 0$) are too harsh to allow survival in low temperature conditions during the winter. Annual recolonization of those regions from regions where DBM can overwinter (with an ecoclimatic index, $EI > 0$) has been biologically and genetically confirmed (Talekar and Shelton, 1993; Chapman et al. 2002; Furlong et al., 2013; Wei et al., 2013; Ke et al., 2019). No genetic differentiation was found among different geographical populations spanning from overwintering regions to seasonally inhabiting regions (Wei et al., 2013; Ke et al., 2019; You et al., 2020). If migration from one habitat overwhelms the other, migration from the source introduces new genetic variation that may prevent local adaptation (Kawecki et al., 2008; Olson-Manning et al., 2012). Therefore, we believe that samples from regions that are only seasonally suitable for DBM (with $EI=0$ and the populations being subject to seasonally punctuated local selection by climatic factors) should be excluded in the analysis of correlation between genetic variation and climatic variables.

Effects of temperature on gene expression.

I cannot understand the authors’ methodology for the temperature treatments. The way I read it I interpret as:

- Groups of five caterpillars (combination of males and females) are assigned to one of three treatments: 1) control (26°C), 2) Low temperature, 3) High temperature.
- Each treatment is repeated 5 times (75 total caterpillars, haphazard mix of males and females spread across all treatments).

In total, we set nine temperature treatments and one control at 26°C that is a favorable temperature for DBM to grow and develop. Each treatment had three replicates: a total of fifteen vials with DBM females and fifteen vials with males were used in each treatment (so 5 vials with females and 5 vials with males repeated three times for each treatment). After treatment and freezing in liquid nitrogen, every five females/males

was placed into one 1.5ml tube for RNA extraction.

To deal with this query, the methodology for heat- and chilling-stress treatments has been revised as follows: (Line 396-413) “In total we used nine temperature treatments and one control at 26°C. A male or female individual was placed in a 1.5ml plastic vial (4.0 cm height) with a pinhole in the side wall to allow air exchange. Before treatments, all vials with DBM were placed into the incubator at 26°C. A group of thirty vials (15 vials with DBM females and 15 with males) was frozen in liquid nitrogen and used as control, and other groups of vials were exposed to each of the following nine distinct temperature treatments according to a previous study on lethal temperature limits of DBM⁶⁹ and our pre-experiment trials: three high-temperature treatments: (1) H1: 40°C for 30 min, (2) H2: 43°C for 30 min, (3) H3: 43°C for 30 min with 24 h of recovery at 26°C; and six low-temperature treatments: (1) L1: -14°C for 30 min, (2) L2: -14°C for 30 min with 24 h of recovery at 26°C, (3) L3: -17°C for 30 min, (4) L4: -17°C for 30 min with 24 h of recovery at 26°C, (5) L5: -20°C for 15 min, and (6) L6: -20°C for 15 min with 24 h recovery at 26°C (Fig. 3). High and low temperature treatments were conducted in incubators and freezers, respectively. The exposure duration of moths at -20°C was set for 15 min because moths started to die when exposed to -20°C for over 20min. After each of the treatments, moths were immediately frozen in liquid nitrogen. Each group of vials was divided into three replicates: both 15 vials with females and 15 with males were evenly put into three 1.5ml tubes, respectively (or both 5 vials with females and 5 with males were repeated three times, respectively). All tubes with frozen moth samples were stored at -80°C before RNA extraction.”

I have questions about the temperature treatments and the stresses encountered by the caterpillars. What happens to the control caterpillars? Are they just left in the incubator while the others are handled, moved, stressed. Therefore, I find it difficult to disentangle the effects of temperature from stress. Although temperature is itself imposing a stress, I do not feel the treatments can conclude that temperature is the cause of the changes in gene expression.

In the present study, we focus on the comparison of difference between each of the temperature treatments and the control rather than the comparison between different non-control temperature treatments. Before treatments, all vials with DBM were placed into the incubator at 26°C. According to a previous study (Dumas et al., 2019), a group of thirty vials (15 vials with DBM females and 15 with males) was frozen in liquid nitrogen and used as control, and then other groups of vials were removed from the initial incubator at 26°C and handled manually in a room (with the room temperature at 26°C) to allocate them randomly to the assigned temperature treatments. After each of the treatments, moths were immediately frozen in liquid nitrogen so that we could compare the difference in gene expression between each of the temperature treatments and the control.

We have provided more information in the Methods as follows: (Line 429-433)

“In the present study, we focused on the comparison of difference in gene expression between each of the temperature treatments and the control rather than the comparison between different non-control temperature treatments. Therefore, we used independent t-tests to perform in R to establish differences in gene expression between each temperature treatment and the control.”

It seems that the treatments are quite unbalanced. The High temperature treatment was “40°C for 30 min, 43°C for 30 min, 43°C for 30 min with 24 h of recovery.” What is the rationale for these temperatures? The Low temperature treatment is even more confusing: “-14°C for 30 min, -17°C for 30 min, -20°C for 15 min, -14°C for 30 min with 24 h of recovery at 26°C, -17°C for 30 min with 24 h of recovery at 26°C, and -20°C for 15 min with 24 h of recovery at 26°C” ? Why so many different temperatures, different exposure times?

Different temperatures and exposure times were set in our experiment according to a previous study on lethal temperature limits (temperatures where there is no survival) of DBM in ramping and plunging assays (Nguyen et al., 2014) and our pre-experiment trials.

In that study, the upper lethal temperature limit (ULT₀) for DBM was recorded as 42.6°C and the lower lethal temperature limit (LLT₀) was recorded as -16.5°C, respectively, which were significantly impacted by, and could be varied with, the duration of exposure to temperature. In their heat shock experiment, when DBM were exposed for 10 or 30 minutes at 42.5°C, 100% of moths survived, whilst at 60 minutes survival was 60%, and at 120 minutes survival was 20%. In the cold shock experiment, the lowest survival occurred at -20°C. When DBM were exposed for 10 minutes at different low temperatures, there was 100% survival until -15°C, with 40% survival at -20°C; survival substantially decreased when exposed to low temperatures for >1 hour.

In our study, we need to ensure the treated moths were alive under different temperatures for RNA extraction. According to our pre-experiment trials, we therefore set nine different temperature treatments: three high-temperature treatments: (1) H1: 40°C for 30 min, (2) H2: 43°C for 30 min, (3) H3: 43°C for 30 min with 24 h of recovery at 26°C; and six low-temperature treatments: (1) L1: -14°C for 30 min, (2) L2: -14°C for 30 min with 24 h of recovery at 26°C, (3) L3: -17°C for 30 min, (4) L4: -17°C for 30 min with 24 h of recovery at 26°C, (5) L5: -20°C for 15 min, and (6) L6: -20°C for 15 min with 24 h recovery at 26°C (Fig. 3). The exposure duration of moths below -20°C was set for 15 min because moths started to die when exposed to -20°C for over 20 min.

In the section of Methods, we have changed the relevant statement to addressed why we set so many different temperatures and different exposure times (as shown in the

previous response).

Gene expression by RT-qPCR

“After treatments, moths were frozen in liquid nitrogen and stored at -80°C” Is there a rationale to look at cadherin gene expression after a series of temperature treatments followed by periods of recovery. Gene expression is transient. Caterpillars should have been frozen during the exposure to look at the acute effects. I feel this authors have likely missed important information on gene expression differences in response to extreme temperatures.

In the present study, we designed our experiment to look at cadherin gene expression after a series of temperature treatments followed by periods of recovery by referring to a previous study (Zhang and Denlinger, 2010). In that study, the authors examined the expression patterns of heat shock protein transcripts, *hsp90*, *hsp70*, *hsc70*, isolated from the corn earworm during thermal stress and pupal diapause in the heat (40°C) and chilling (0°C, 4°C, 8°C) conditions, with a recovery at 25°C for each of the temperature treatments.

In our experiment, a recovery at 26°C was arranged for each of the heat/chilling stress treatments (43°C, -14°C, -17°C and -20°C) in order to examine whether or not gene expression could return to the original level when moths were transferred to a favorable temperature at 26°C after heat/chilling stress treatments. We did not arrange a recovery for the treatment at 40°C because that is a tolerable temperature for DBM to live within a duration of 2 hours (Nguyen et al., 2014). In total, therefore, we set nine temperature treatments and one control. The nine temperature treatments included five different temperature treatments (43°C, 40°C, -14°C, -17°C and -20°C), and four temperature treatments (43°C, -14°C, -17°C and -20°C) followed separately by a recovery for 24 hours. After each treatment, moths were immediately frozen in liquid nitrogen so that we could compare the difference in gene expression between each of the temperature treatments and the control.

The display of the resultant data (Figure 3) is extremely confusing. Males are on the left, females on the right. Why are there different colours. What does the x-axis represent, are these paired comparisons? What is 43 compared to 43R? Where is the data for the wildtype (WT) and the mutant (MU)? What are the t-tests comparing? I'm sorry but I cannot make any rational interpretation of the data displayed in this figure.

We apologize for the confusion. In this experiment, we only examined the gene expression of *PxCad* in the wild-type (WT) strain. There was no such work with the mutant (MU). Therefore, we have removed the erroneous “and *PxCad*-deficient (MU)” in the figure caption.

Different colors have been omitted in the figure (Figure 3). The different treatments

are shown on the X-axis.

We compared the expression of *PxCad* for each treatment against the control using independent t-tests. The relative expression level of PxCad in each treatment was normalized to the abundance of samples under control temperature, using the $2^{-\Delta\Delta C_t}$ method⁴². Expression of *PxCad* at 26°C was set as control with a relative expression value being set as 1. We have updated the legend of Figure 3 accordingly.”

The susceptible wildtype G88 strain

What is this strain? How was it collected? Is it inbred, isogenic? Why is it considered susceptible. I’m sorry but I find myself with 100 questions for every answer. This needs to be tight and it needs to be clear.

The insecticide susceptible strain, Geneva 88 (G88), was collected from the New York State Agricultural Experiment Station in 1988 and maintained on artificial diet without exposure to insecticides (Shelton et al., 1991). It was provided by Dr. Antony M. Shelton (Cornell University, USA) to the Institute of Applied Ecology, Fujian Agriculture and Forestry University, in 2016. Since 2016, we have been maintaining this strain on artificial diet without exposure to insecticides. The strain was tested by Shelton’s team and proved sensitive to several insecticides including Bt-products (Selton et al., 1993). We used the word “susceptible” based on some of the previous relevant studies that considered G88 as susceptible (e.g., Shelton et al., 1993; Liu et al., 2020). We have revised the relevant statements in the Methods, and replace “susceptible” with “wild-type” to avoid possible confusion.

Site-directed mutagenesis in PxCad via CRISPR/Cas9

On line 190 the authors state : “None of these peptides were detected in the G88-Cad strain, confirming...” However, when I look at Fig 4b I cannot see a difference between 1: G88 and 2: G88-Cad (the mutant) in the SDS Page gel. Presumably there should be differences in the banding patterns to show that the mutant strain is not making the 19 peptides in the 120-250 kDa range encoded by PxCad.

The amount of PxCad proteins in the G88 strain was too small to be visible to the naked eye in the SDS PAGE gel. However, we excised the 120-250 kDa range (that was predicted to contain PxCad proteins) from the gel lanes and used nano-LC-MS/MS to analyze these gel segments (~120-250 kDa of the BBMV proteins). This identified nineteen tryptic peptides specific to PxCad from the G88 strain (Figs. 4c and d). None of these peptides were detected in the G88-Cad strain, which confirmed that PxCad protein was totally disrupted in the PxCad-knockout strain (G88-Cad).

We have changed the relevant section of the text to “After screening using single-pair crosses and molecular identification, one homozygous mutant strain (G88-Cad) with a 46-bp deletion in *PxCad* exon 3 was generated (Fig. 4b). Using nano-LC-MS/MS to analyze the gel slices (~120-250 kDa of the BBMV proteins) separated by SDS-PAGE (Fig. 4b), we identified nineteen tryptic peptides specific to *PxCad* from the G88 strain (Figs. 4c and d). None of these peptides were detected in the G88-Cad strain, which confirmed that *PxCad* protein was totally disrupted in the *PxCad*-knockout strain.” (Line 161-167)

PxCad – Cadherins and temperature

At no point in this manuscript is a conceptual link between cadherins and temperature adaptation explored, discussed, or even proposed. Cadherin proteins, in my mind, would not have been a candidate gene for climate change adaptation. Therefore, it appears, at least as it is written, that the authors aggressively pursued this gene without thinking about what it does, why it might respond to temperature, and why genetic variation would exist.

We have added the following two paragraphs in the text (Discussion) to present the potential contribution of cadherins to climate adaptation and justify their selection for study.

Line 207-222

“*PxCad* is annotated as a cadherin-like protein^{39,40}. Classical cadherins are a superfamily of transmembrane proteins involved in regulating cell-cell adhesion, signal transduction and tissue morphogenesis^{41,42}. In mammals, epithelial cadherin (E-cadherin) is involved in morphogenesis^{43,44} whilst in insects, studies of cadherin-like proteins primarily have focused on their involvement in mediating resistance to the biological insecticide, Bt⁴⁵⁻⁴⁸. Pigott & Ellar⁴⁹ have also demonstrated the roles of cadherin-like proteins in maintaining structural integrity of midgut epithelial organization.

Thermal stress (heat or cold) generally disrupts cellular homeostasis^{50,51}. Our results show that *PxCad* expression in female adults was significantly down-regulated at high temperature. This is consistent with E-cadherin studies in human lung adenocarcinoma cells⁵² and in the purple sea urchin *Strongylocentrotus purpuratus*⁵³. In contrast, cold stress significantly up regulated *PxCad* expression in both males and females, and the tolerance to cold stress in DBM declined when *PxCad* was knocked-out. Comparable phenotypes have also been reported in *Ruditapes philippinarum*, with cadherin genes acting as responders to cold stress⁵⁴. Under heat and cold stress, considerable variation of *Pxcad* expression and survival rates between DBM stains indicates that *PxCad* is involved in regulating DBM’s response to thermal stress.”.

What do the SNPs in *PxCad* do? Aren’t they more important than creating a mutant?

The authors zero in on *PxCad* due to the convergence of significant associations and several SNPs? What do these SNPs do? Several are located the coding region? I wonder why the authors chose to dive straight into genetic manipulation without exploring the important standing genetic variation in *PxCad* and its role in climactic adaptation. This represents a far more natural scenario than creating a mutant that knocks out cadherin function. Furthermore, this may represent the functional adaptations themselves! Why not asses temperature adaptation (and several other variables) in DBM collected from different temperatures? You already have candidate SNPs within your candidate gene to look.

We zeroed in on *PxCad* because: 1) it had the highest number of SNPs (9 SNPs, 7 of which are located in coding region), identified to be putatively associated with temperature adaptation according to the prediction based on three models of Samþáda v0.5.3, latent factor mixed models (LFMM), and Bayenv 2 (Table S7); and 2). *PxCad* is annotated as a cadherin-like protein (Guo et al., 2015; Park et al., 2015). Classical cadherins are a superfamily of transmembrane proteins involved in regulating cell-cell adhesion, signal transduction and tissue morphogenesis (Bulgakova, Klapholz, & Brown, 2012; Wu & Maniatis, 1999). For example, in mammals, epithelial cadherin (E-cadherin) is involved in morphogenesis (Batlle et al., 2000; Behrens, Löwrick, Klein-Hitpass, & Birchmeier, 1991). Cadherin genes have also been reported as responders for cold stress in *Ruditapes philippinarum* (Menike et al., 2014). Pigott & Ellar (2007) have also demonstrated the roles of cadherin-like proteins in maintaining structural integrity of midgut epithelial organization. All of the above support the potential of *PxCad* as a temperature-sensitive responder to climate change.

Once having a target gene of interest, it is common to functionally verify the gene identity using reverse genetics approaches (Zhan et al., 2014, Huang et al., 2018). In this study, therefore, the RT-qPCR and CRISPR-Cas9-mediated gene knockout were employed to examine the role of *PxCad* in regulating DBM's responses to temperature, i.e. we investigated the phenotypes of DBM resulting from the alternation of this known gene, *PxCad* (Huang et al., 2018). Although the only one nonsynonymous mutation (E101G) resulted from those identified 7 SNPs in the coding region does not lead to a significant conformational change, according to the homology modeling (the following figure, Fig. 1), the exact roles of the identified SNPs of *PxCad* indeed warrants further research, which we are currently conducting.

We are also using the forward genetics approach to explore mechanisms underlying temperature adaptation of the DBM strains reared at different temperatures (lab conditions). The relevant findings are presented in a separate manuscript that will be submitted soon.

Fig. 1 Homology models with the structural superpositions of wild type and mutant proteins of *PxCad* (E: glutamate; G: glycine).

Is this really temperature?

I seriously question whether the genetic variation detected among the populations in this student could be due to demographic effects. For example, in *Drosophila melanogaster*, there is a strong collinearity between thermal clines and ancestry making it particularly challenging to disentangle demographic signals from true adaptive signals. For example see Excoffier et al. 2009 *Annu Rev Ecol Evol Syst* 40: 481, Duchon et al. 2013 *Genetics* 193: 291, Bergland et al. 2016 *PLoS Genet* 10: e1004775, Flatt 2016 *Mol Ecol* 25: 1023.

We acknowledge the contribution of demographic effects to genetic variation, especially for such a study covering the global scale. The first step in this study therefore was to exclude SNPs that were not identified in Bayenv/Sambada/LFMM, three robust models that provide statistical framework for controlling the confounding effects of neutral genetic structure (Rellstab et al., 2015), and focused on those putative SNPs under selection. By doing so, we could at least partly tease apart the effects of neutral processes from those of selection. To further disentangle demographic signals from true adaptive signals, we have replaced the gradient forests (GF) with the generalized dissimilarity modeling (GDM), a more appropriate method that can directly identify the effects of distance and variation in habitat permeability (Fitzpatrick and Keller, 2015). The entire process for identifying climate-associated genomic variation is described in the revised Methods as follows:

Line 273-334

“Climate associated genomic variation

Identification of SNPs under climate selection

Three models, Samβada¹⁷, latent factor mixed model (LFMM)¹⁸ and Bayenv 2¹⁹, were used to identify putatively adaptive loci associated with climate variables. Samβada applies logistic regression models to identify associations between specific genetic

markers and environmental variables¹⁷. Simple univariate and multivariate logistic regression models for each climate variable were performed. A single SNP was considered to be a candidate locus when the log-likelihood ratios (G scores) and/or Wald scores were significant with Bonferroni correction at a 99% confidence level. LFMM is developed based on population genetics, ecological modeling, and statistical analysis to identify the candidate loci that are highly correlated with environmental variables¹⁸. SNPs showing an association with climate variables were identified based on z-scores, which was computed using 10,000 cycles and 5,000 sweeps for burn-in. We used R package 3.6.2 to run the median z-scores of 5 runs and re-adjusting p-values with FDR correction. SNP with median z-scores above the absolute value of 4 and corresponding to P value $< 10^{-5}$ were considered as significant locus. In Bayenv 2, a covariance matrix based on putatively neutral markers is used as a null model to control demographic effects when testing relationships between the genetic differentiation and a given environmental variable¹⁹. We randomly sampled SNPs at 200 SNP intervals from the SNP dataset. A total of 117,887 SNPs with loose linkage disequilibrium were obtained for developing the covariance matrix, which was estimated with 100,000 iterations. We then assessed the correlations between individual SNPs and 19 climate variables at 100,000 Markov chain Monte Carlo (MCMC) for Bayes factor analysis. The results were presented as Bayes factors (BFs). A $\log_{10}(\text{BF})$ value > 1.5 is usually considered as high support for a model where environmental parameters have significant effects on allele frequencies⁶¹. A total of 3,307 putative adaptive SNPs were identified by at least one of the three models (Samβada, LFMM and Bayenv 2; Supplementary Tables 3, 4 and 5).

Prediction of climate-associated genomic variation based on generalized dissimilarity modelling

Generalized dissimilarity modelling (GDM)²⁰, a distance-based model, can account for the nonlinear relationship between genetic variation and environmental / geographical factors, and has been recently used to map ecological adaptation from genomic data under current and future climates⁴. Firstly, we examined spatially explicit selection process for each of the putative adaptive SNPs using GDM⁴, with the R package *gdm*⁶². We subsampled the genetic dataset to include only populations

with sample size ≥ 5 to obtain accurate allele frequencies. An uncorrelated subset of 12 climate variables (bio01, bio02, bio03, bio05, bio07, bio08, bio09, bio12, bio14, bio15, bio18, bio19 in Supplementary Table 6) with a pairwise Pearson correlation coefficient less than 0.8 in 60 sample sites was used in GDM. Pairwise F_{ST} matrix among 60 populations were calculated for each of the 3,307 SNPs using the R package *hierfstat*⁶³, and rescaled between 0 and 1. Geographical distance in the GDM was based on Euclidean distance as the thirteenth variable to test whether genetic variation across environmental gradients was better explained by climate variables than geographical distance, which effectively acts as a screening for SNPs that may respond predominantly to neutral genetic process including isolation by distance⁶⁴. The relative importance of the 12 climate variables and geographical distance was ranked based on the fitted I-Splines in GDM (Fig. 1b). The maximum value of each

variable in the fitted I-Splines was rescaled between 0 and 1. Those SNPs with geographical distance ranking as one of the top 3 important variables were excluded in the following GDM analysis. In addition, we randomly sampled 200 SNPs as a “reference group” to test its explainable proportion of the GDM deviance. According to our GDM record, the reference SNP group accounted for 11.2% of the GDM deviance for the entire model, so that those SNPs with a < 11.2% contribution to the GDM deviance were also excluded in further GDM analysis. After additional filtering, 419 of 3307 SNPs were retained. The 419-SNP-based genetic distance matrix was further integrated with geographical distance and climate variables to be used in the entire GDM model. The entire GDM model explained 41.2% of the deviance for the 419 SNPs. To predict the climate adaptation of DBM, we finally retrieved current climate variables at 61,655 gridded points across the world from WorldClim, using ArcGIS 10.2. The *gdm.transform* function was used to predict and map the pattern of climate-associated genomic variation along environmental gradients across the world (Fig. 1c). The genetic turnover was summarized using a principal component analysis (PCA), with the top three components transformed for visualization in a red-green-blue (RGB) colour scale as suggested in the GDM manual⁶². Loadings based on the principal components indicate the direction and magnitude of association with adaptation to different predictors (Fig. 1c). The climate-associated genomic variation along environmental gradients in DBM across the world was visualized, with similar pattern of genetic composition at climate-adaptive loci illustrated by similar colours (Fig. 1c).”

The corresponding results based on redefined SNP dataset (i.e. 419 SNPs) are presented in the updated Fig. 1.

Non-genetic inheritance?

I think the authors need to at least discuss the possibility that non genetic inheritance could play an important part in organisms adapting to climate change. Epigenetic effects can respond much quicker to environmental perturbations than genetic inheritance. In return, because such effects erode much quicker than genetic adaptation it is a major challenge to infer how DBM have responded to climate change in the past. The authors use genetic offset to infer the full adaptive potential for this species. They might not be able to address this experimentally, but they have to at least acknowledge that there is potentially an additional major side to how DBMs and other species might respond adaptively to climate change.

We agree that non genetic inheritance may play an important role in facilitating the adaptation of organisms to climate. As suggested, we have acknowledged the potential roles of epigenetic effects in buffering populations against rapid environmental change, and added relevant statements in the last paragraph of Discussion. It now reads: “The present study illustrates how the integration of genomic data with climate variables can be used to improve our understanding of

population-level variation in species. We also show that this approach is useful for the identification of climate-associated responder genes linked to genetically adaptive evolution. It is increasingly recognized that acclimatization through non-genetic inheritance (e.g., phenotypic plasticity or epigenetic processes) may buffer populations against rapid environmental change, allowing adaptive responses to climate change⁵⁵⁻⁵⁷. Further research is needed to explore the role of non-genetic effects in adaptation of DBM to future climate, and to elucidate the underlying mechanisms and their relative importance compared with genetic adaptive capacity.” (Line 223-231)

The written document

Overall the manuscript is quite poorly written. The introduction is very confusing. Several statements are discussed without proper set up. For example, the topic sentence claims that human impact influences global temperature and levels of precipitation. This statement is not very impactful as it is not backed up, nor is it surprising since it is general knowledge. The second sentence provides a new statement that claims that adapting to changes directly influence survival etc. The next sentence provides a third statement that claims adaptive capacity is important to mitigate environmental perturbations. Adaptive capacity is not explained or defined, and nonuniformity is also not defined/explained. Next authors switch to adaptive plasticity, again without explaining or making it explicit what they mean by adaptive plasticity. The last sentence in the paragraph is actually the most impactful statement and the one they should start with and should provide evidence for.

The authors do not develop the idea for using an insect pest in the context of climate change. They need to do a better job of setting up the connection between climate change adaptation and pest/invasive species.

We have substantially rewritten the manuscript. The revised Introduction, for example, now includes the following opening:

Line 34-59

“Human-induced climate change, especially gradual changes in temperature and precipitation¹, is impacting species’ survival and distribution². The ability of pests to successfully adapt to these changes will impact biodiversity, food production and the economy. Intraspecific variation in tolerance to climate change has been documented for many plant and animal species^{3,4}. Populations with high adaptive potential are expected to better cope with changes in habitat suitability arising from climate change^{5,6}. However, the broader phenomenon of adaptive capacity of species is not well-understood. Studying the genetic mechanisms that underpin adaptation of cosmopolitan species at the local scale is therefore important to predict both population- and global-level responses to future environmental change, and assist in management efforts⁵.

Genetic variation associated with climate variables has been demonstrated in

several species^{5,7-9}. Insects with high fecundity and short generation time can rapidly accumulate adaptive alleles and potentially have great capacity to respond to changing environmental conditions¹⁰⁻¹². However, little is known about the extent to which genetic variation is shaped by climate variables in arthropod species, and how populations within species differ in their capacity to adapt to climate change. Here, by combining a newly available genomic resource with climate data, we analyze the relationship between genomic variation and climate variables to identify the important factors determining climate-associated genetic adaptation in the diamondback moth (DBM), *Plutella xylostella*. This insect is one of the world's top 10 arthropod pests¹³, with a global distribution spanning a remarkably extensive range of climates¹⁴. We define a new eco-genetic index to examine population-level variation in response to climate change by combining the genetic offset (that quantifies the disruption of gene-environment relationships subject to future climates) with the ecoclimatic index (that describes habitat suitability for species persistence). Subsequently, based on a core dataset of identified nuclear SNPs, we functionally test a temperature-related gene to reveal its role in climatic adaptation in DBM.”

The discussion is far too short, and completely ineffective. It is one very short paragraph that fails to place their study in the context of climate change adaptation, fails to discuss why PxCad, and completely fails to convince the reader (if they aren't already unconvinced).

We have substantially rewritten the discussion as follows:

Line 179-231

“In this study, we provide evidence of the genetic basis of climate adaptation in DBM, a worldwide pest important to food safety and economy. Climate-associated genetic variation in DBM populations was quantified and visualized. A multi-model analysis of Samβada, LFMM and Bayenv 2 allows robust identification of climate-associated adaptive loci, reducing false-positives³⁵.

Our analyses with the nuclear SNPs from geographically distributed samples goes further in indicating the effects of both temperature- and precipitation-related variables on the climate-associated genetic variation in DBM populations worldwide. This follows a number of recent studies demonstrating the key role of temperature in mediating environment-associated adaptive variation for the other insects, *Phaulacridium vittatum*³⁶, *Chironomus riparius*³⁷, and *Ceracris kiangsu*³⁸. Further, physiological data can be used to determine the tolerance of DBM to future climate at the species level (e.g.,³⁴). To date, however, there are no site-specific data available for investigation on the physiological variation in different populations of DBM. In this study, we find that genetic variation is tied to physiological effects that vary with local adaptation in response to climate change, and that most populations show high genetic tolerance to projected future climate. This result not only shows that DBM will maintain its pest status in most regions of year-round persistence past 2050 under the RCP8.5 climate change scenario, but also present region-specific genomic responses of DBM to the changing climate. This is of practical use to future pest

management because those populations are more resistant to changing environments under a future climate. Not only will pest management in these regions need to be maintained and strengthened as new technologies are developed, areas that are seasonally colonized from the year-round persistence regions are likely to remain vulnerable as recipient habitats. For a species with notable migratory capacity, like DBM, pest status is likely to increase given high levels of gene flow. DBM needs to be monitored at landscape and regional scales (in addition to conventional monitoring at a local scale), considering its genetic background with respect to environmental adaptation and the spatial dynamics of insecticide resistant strains.

For the first time, using RT-qPCR and CRISPR/Cas 9 approaches in this species, we verified that a specific gene, *PxCad*, represented a temperature-sensitive responder to climatic change, thus contributing to the genetic basis of adaptive evolution. *PxCad* is annotated as a cadherin-like protein^{39,40}. Classical cadherins are a superfamily of transmembrane proteins involved in regulating cell-cell adhesion, signal transduction and tissue morphogenesis^{41,42}. In mammals, epithelial cadherin (E-cadherin) is involved in morphogenesis^{43,44} whilst in insects, studies of cadherin-like proteins primarily have focused on their involvement in mediating resistance to the biological insecticide, Bt⁴⁵⁻⁴⁸. Pigott & Ellar⁴⁹ have also demonstrated the roles of cadherin-like proteins in maintaining structural integrity of midgut epithelial organization.

Thermal stress (heat or cold) generally disrupts cellular homeostasis^{50,51}. Our results show that *PxCad* expression in female adults was significantly down-regulated at high temperature. This is consistent with E-cadherin studies in human lung adenocarcinoma cells⁵² and in the purple sea urchin *Strongylocentrotus purpuratus*⁵³. In contrast, cold stress significantly up regulated *PxCad* expression in both males and females, and the tolerance to cold stress in DBM declined when *PxCad* was knocked-out. Comparable phenotypes have also been reported in *Ruditapes philippinarum*, with cadherin genes acting as responders to cold stress⁵⁴. Under heat and cold stress, considerable variation of *Pxcad* expression and survival rates between DBM stains indicates that *PxCad* is involved in regulating DBM's response to thermal stress.

The present study illustrates how the integration of genomic data with climate variables can be used to improve our understanding of population-level variation in species. We also show that this approach is useful for the identification of climate-associated responder genes linked to genetically adaptive evolution. It is increasingly recognized that acclimatization through non-genetic inheritance (e.g., phenotypic plasticity or epigenetic processes) may buffer populations against rapid environmental change, allowing adaptive responses to climate change⁵⁵⁻⁵⁷. Further research is needed to explore the role of non-genetic effects in adaptation of DBM to future climate, and to elucidate the underlying mechanisms and their relative importance compared with genetic adaptive capacity.”

Minor comments:

Lines 24-25: “Efforts to investigate these responses have largely neglected

intraspecific variation arising from local adaptation”

I suggest avoiding pointing out what hasn't been done or what has been neglected. Instead point out what you are adding and why that is important.

We have revised this sentence to: “However, intraspecific variation of these responses arising from local adaptation remains ambiguous for most species.”(Line 21-22)

Lines 54-55: “These attributes make it an excellent...”

Which attributes? Why does this make it an excellent system? Why does the fact that it is a pest make it a good system for climate change?

The second paragraph in the Introduction has been rewritten as: (Line 44-53) “Genetic variation associated with climate variables has been demonstrated in several species^{5,7-9}. Insects with high fecundity and short generation time can rapidly accumulate adaptive alleles and potentially have great capacity to respond to changing environmental conditions¹⁰⁻¹². However, little is known about the extent to which genetic variation is shaped by climate variables in arthropod species, and how populations within species differ in their capacity to adapt to climate change. Here, by combining a newly available genomic resource with climate data, we analyze the relationship between genomic variation and climate variables to identify the important factors determining climate-associated genetic adaptation in the diamondback moth (DBM), *Plutella xylostella*. This insect is one of the world's top 10 arthropod pests¹³, with a global distribution spanning a remarkably extensive range of climates¹⁴.”

Line 59: Define ‘genetic offset’ and ‘ecoclimactic index’ here since the definitions come far too late in the manuscript

We now provide a definition of ‘genetic offset’ and ‘ecoclimactic index’ in the second paragraph of Introduction: “We define a new eco-genetic index to examine population-level variation in response to climate change by combining the genetic offset (that quantifies the disruption of gene-environment relationships subject to future climates) with the ecoclimactic index (that describes habitat suitability for species persistence).” (Line 53-57)

Line 100: “gene-environment relationships”

Although I acknowledge these are important with respect to genetic offset, I think you need to define these a bit more since it may not be intuitive to the reader.

The statement has been changed to “To investigate which DBM populations might be more vulnerable to future climate change, we defined a new metric of genetic vulnerability that we called “genetic offset”. It represents the mismatch between

current and expected future genomic variation based on genotype-environment relationships modelled by GDM analysis²¹ across contemporary populations.” (Line 89-93)

Line 136: “genetically adaptive potential”

What does this mean?

It means the genetic potential for climate adaptation. We took this term directly from Fitzpatrick and Edelsparre (2018).

Line 173: “locomotory behaviour”

This gene was not shown to affect behaviour, rather mutations in this gene affect hearing and deaf animals show differences in behaviour. These differences are due to a loss of hearing not due to a loss of cadherins.

Apologies. We mis-cited a reference and have corrected this in the revised text: Nakagawa R, Matsunaga E, Okanoya K. 2012. Defects in ultrasonic vocalization of cadherin-6 knockout mice. PLoS ONE 7(11): e49233. <https://doi.org/10.1371/journal.pone.0049233>

Line 175-176: “...a susceptible strain of DBM, G88...”

What makes this strain susceptible? How was this determined? Did you determine this? If so then where are the methods, if not then where is the reference?

As mentioned above, the G88 strain was tested by Shelton’s team and proved sensitive to several insecticides including Bt-products (Selton et al., 1993). We used the word “susceptible” based on some of the previous relevant studies that considered G88 as susceptible (e.g., Shelton et al., 1993; Liu et al., 2020). To avoid confusion, we have replaced “susceptible” with “wild-type”, and changed the statements in the section of Methods to “The wild-type strain of DBM, Geneva (G88), was used in this assay. This G88 strain was collected from the New York State Agricultural Experiment Station in 1988 and has since been maintained on artificial diet without exposure to insecticides⁶⁸. It was provided by Dr. Antony M. Shelton (Cornell University, USA) to the Institute of Applied Ecology, Fujian Agriculture and Forestry University in 2016. Since then, we maintained this strain on artificial diet without exposure to insecticides at 26°C that is a favorable temperature to rear and maintain the wild-type strain of DBM (G88).” (Line 389-395)

Line 178: “-14oC, -17oC and -20oC”

Why were these temperatures chosen along with the high temperatures. What is the

rationale?

As mentioned above, different temperatures were set in our experiment according to a previous study on lethal temperature limits of DBM (Nguyen et al., 2014) and our pre-experiment trials. In our pre-experiment trials, we first verified that the wild-type (WT) strain and the mutant strain (*PxCad*-deficit) showed different responses at -17°C, and then two other temperature gradients were set to further confirm the roles of *PxCad* in mediating responses of DBM to different temperatures.

Line 378: “with a pinhole in the side wall”

Why is this done? What is the purpose? How are the caterpillars raised in the lab? Is the pin hole to allow air exchange? This needs more explanation.

Yes, the pinhole in the side wall allows air exchange. We have added this information in the text as follow: “A male or female individual was placed in a 1.5ml plastic vial (4.0 cm height) with a pinhole in the side wall to allow air exchange.” (Line 396-398)

The caterpillars were maintained on artificial diet under 26°C in the lab. We have described the rearing conditions in the Methods as follow: “The wild-type strain of DBM, Geneva (G88), was used in this assay. This G88 strain was collected from the New York State Agricultural Experiment Station in 1988 and has since been maintained on artificial diet without exposure to insecticides⁶⁸. It was provided by Dr. Antony M. Shelton (Cornell University, USA) to the Institute of Applied Ecology, Fujian Agriculture and Forestry University in 2016. Since then, we maintained this strain on artificial diet without exposure to insecticides at 26°C that is a favorable temperature to rear and maintain the wild-type strain of DBM (G88).” (Line 389-395)

Line 490: publication year is 2015 not 2014

Publication year has been revised accordingly.

Figure 2: The light blue is very difficult to see, especially since many spots are very small. This will be even hard to see in the published version. Suggest considering alternatives.

The Fig. 2 has been revised as suggested.

Figure 4: “Twenty individuals were used for each of the treatments with four replicates.”

This means 4 replicates of 20. Do you mean to say 4 replicates of 5 (N = 20)?

It means 4 replicates for each treatment, and 20 individuals for each replicate. This

statement has been changed to “Twenty individuals were used for each replicate, with four replicates in each treatment” in the legend of Figure.

Very minor comments:

Line 46: “...adaptation at the local scale is therefore important ...”

I suggest adding ‘of cosmopolitan species’ after scale

“of cosmopolitan species” has been added as suggested. (Line 41)

Line 83-84: “We used the gradient forest, a machine-learning regression approach, to examine climate-mediated...”

Suggest rewording: ‘We used a machine-learning regression approach, gradient forest, to examine climate-mediated...’

We have replaced the gradient forests (GF) with the generalized dissimilarity modeling (GDM). (Line 78)

Line 107: “...tolerant to future climate change thus likely to remain damaging...”

Suggested edit: ‘...tolerant to future climate change making them likely to remain damaging...’

The sentence has been revised as suggested. (Line 97-98)

Line 131: “...the “eco-genetic index” (EGI). This combines the genetic...”

Suggested edit: ‘...the “eco-genetic index” (EGI) which combines the genetic...’

The sentence has been revised as suggested. (Line 117)

Line 148: “...coding sequence (CDS)...”

Why the acronym? You only use it once in the entire manuscript. Suggest removing.

We defined the acronym because we later used it in the first sentence of next paragraph: “Based on the 40 SNPs located in the CDS and intron regions from the core subset (Supplementary Table 7), we identified five SNPs from the coding region of a single gene, *PxCad*.” (Line 141-143)

Line 149-150: “These genes cover a wide range of functions...”

Suggest: ‘These genes are predicted to cover a wide range of functions...’
Is this genome annotated. Has gene function been confirmed?

As suggested, we have changed “These genes cover a wide range of functions...” to “These genes are predicted to cover a wide range of functions...” (Line 130-131)

Yes, the genome is annotated, which is accessible at: <http://iae.fafu.edu.cn/DBM/>, with some of the genes have been functionally confirmed.

Line 172: “PxCad, a cadherin-like protein, is known to be involved...”

Suggested edit: ‘Homologs of PxCad encode cadherin-like proteins known to be involved ...’

The sentence has been revised as suggested. (Line 147)

Lines 215-216: “...we verify that a specific gene representing a temperature-sensitive responder to climate change...”

Awkward wording

The sentence has been revised to: “we verified that a specific gene, *PxCad*, represented a temperature-sensitive responder to climatic change”. (Line 205-206)

Line 231: *Arabidopsis thaliana* should be italicized

Arabidopsis thaliana now italicized. (Line 248)

Line 285: “Gradient forest (GF) is originally a community-level model...”

Suggested edit: ‘Gradient forest (GF) was originally created as a community-level model...’

We have replaced the gradient forests (GF) with the generalized dissimilarity modeling (GDM). (Line 300)

Line 337-338: “...the hot-wet stress temperature threshold (TTHW)...”

Another unnecessary acronym. Moreover, I think it should be HWTT not TTHW.

The acronym has been removed as suggested. (Line 358)

Line 341: calculated should be changed to calculate

As suggested, calculated has been changed to calculate. (Line 361)

Line 355: “ $EGI = DEI \times GON$ ”

Why the subscript N? N is not defined. Does this simply mean it is indexed to sample size?

We define N in the following sentence that “ GO_N is the normalization of $GO...$ ”. (Line 375-376)

Line 435: “...in the PxCad knockout strain”

PxCad should be italicized

PxCad has been italicized. (Line 462)

Reviewer #2 (Remarks to the Author):

This work combines approaches from genomics, climate variability and change, and genome editing to explore adaptive variation in *Plutella xylostella*. I really like the holistic way in which this has been done. The result is a clear demonstration of genomic variation associated with climate plus a key gene which likely plays a role in cold adaptation in the species (and also maybe heat response).

My major concern has to do with some of the main conclusions in the paper. For example, that the “results demonstrate that *P. xylostella* is genetically capable to tolerate future climate change and will likely maintain its global pest status past 2050 and even 2080”. This doesn’t seem very novel to me. Physiological study of DBM has already suggested the species will tolerate future warming, for example:

Nguyen C, Bahar MH, Baker G, Andrew NR (2014) Thermal tolerance limits of diamondback moth in ramping and plunging assays. *PLoS One*, 9, e87535.

The results here then confirm these expectations and results. What is it specifically about the genomic associations that add to this conclusion?

Yes, physiological data can indicate the tolerance of DBM to climate change at the species level (e.g., Nguyen et al., 2014). To date, however, there are no site-specific physiological data available for investigation on the physiological variation in different populations of DBM. In this study, we used the site-specific genomic data to investigate and understand the genetic adaptive variation in different populations. We have shown that the genetic vulnerability to future climate is varied with different populations, and such genetic adaptive variation at the population level can be linked

with physiological effects at the species level to examine the population-level variation in local adaptation of DBM in response to climate change. This finding not only further confirms that *P. xylostella* is genetically capable to tolerate future climate change and will likely retain its global pest status beyond 2050 in many regions of the world, but also present population-level responses of *P. xylostella* to the changing climate, i.e. populations with higher genetic offset are more vulnerable to changing environments and require greater variability to adapt.

We have revised a relevant paragraph in the Discussion to show what it is specifically about the genomic associations that add to this conclusion. It now reads: “Further, physiological data can be used to determine the tolerance of DBM to future climate at the species level (e.g., ³⁴). To date, however, there are no site-specific data available for investigation on the physiological variation in different populations of DBM. In this study, we find that genetic variation is tied to physiological effects that vary with local adaptation in response to climate change, and that most populations show high genetic tolerance to projected future climate. This result not only shows that DBM will maintain its pest status in most regions of year-round persistence past 2050 under the RCP8.5 climate change scenario, but also present region-specific genomic responses of DBM to the changing climate.” (Line 189-196)

Also, in the abstract (and elsewhere in the manuscript) there is a suggestion that these results could guide management of the species. How could this be implemented?

We have changed the suggestion in the abstract “This work advances our understanding of adaptive genomic variation along environmental gradients, providing the genetic basis of local climate adaptation and a foundation for guiding effective management efforts.” to “This work advances our understanding of adaptive genomic variation along environmental gradients, and highlights the genetic basis to local climate adaptation.” (Line 30-31)

To address the potential application of our results to practical management of DBM, we have added some statements in the revised Discussion: (Line 191-204) “In this study, we find that genetic variation is tied to physiological effects that vary with local adaptation in response to climate change, and that most populations show high genetic tolerance to projected future climate. This result not only shows that DBM will maintain its pest status in most regions of year-round persistence past 2050 under the RCP8.5 climate change scenario, but also present region-specific genomic responses of DBM to the changing climate. This is of practical use to future pest management because those populations are more resistant to changing environments under a future climate. Not only will pest management in these regions need to be maintained and strengthened as new technologies are developed, areas that are seasonally colonized from the year-round persistence regions are likely to remain vulnerable as recipient habitats. For a species with notable migratory capacity, like DBM, pest status is likely to increase given high levels of gene flow. DBM needs to be monitored at landscape

and regional scales (in addition to conventional monitoring at a local scale), considering its genetic background with respect to environmental adaptation and the spatial dynamics of insecticide resistant strains.”

Another major concern I have is in the use of different scenarios in the climate change modelling (see details below). It’s hard to interpret the results given the outdated emissions scenarios used. I’m not sure the result would change qualitatively if other scenarios were changed for the reason outlined above on physiology of the study. Still, as the spatial patterns are key, it’s important to use the most up to date climatic data to both demonstrate the utility of the approach and present the most accurate results.

As suggested, the updated RCP emission scenarios have been used for our predictions. The estimation of DEI and EGI was based on RCP 8.5 exclusively because it is the only emission scenario available in CliMond (<https://www.climond.org/>). The estimation of genetic offset was based on RCP 2.6, RCP 4.5, RCP 6.0, and RCP 8.5 from WorldClim with all of the results being presented in Supplementary Fig. 2.

Title – would “cosmopolitan pest” or “cosmopolitan insect pest” be more descriptive?

The title has been changed to: Large-scale genome-wide study reveals climate adaptive variability in a cosmopolitan pest

Lines 27-28 – this is somewhat circular logic, or anyway isn’t really a specific result, more of a description of the approach

The sentence has been revised to: “Here, we analyze genomic data from *Plutella xylostella* collected from 78 geographical sites spanning six continents to reveal that climate-associated adaptive variation exhibits a roughly latitudinal pattern.” (Line 21-24)

Lines 70-71, 234-236 – why subset the sampling in this way? I would guess that such sites would either yield interesting results with respect to selection or would be undifferentiated from other sites.

Adaptive evolution is a biological process that describes how organisms change in response to perennial selection pressure over successive generations (Sejian et al., 2015). As mentioned in the Methods, populations from regions that are seasonally suitable do not receive perennial selection by local environmental variables, and genetic variation in these regions cannot be continuously passed on to future generations over years. In other words, for DBM, regions that are only seasonally suitable for its growth and development (i.e., with an ecoclimatic index, $EI=0$) are too harsh to survive in low temperature conditions during the winter. Annual recolonization of those regions from regions where DBM can overwinter (with an ecoclimatic index, $EI>0$) has been biologically and genetically confirmed (Talekar

and Shelton, 1993; Chapman et al. 2002; Furlong et al., 2013; Wei et al., 2013; Ke et al., 2019). No genetic differentiation was found among different geographical populations spanning from overwintering regions to seasonally inhabiting regions (Wei et al., 2013; Ke et al., 2019; You et al., 2020). If migration from one habitat overwhelms the other, migration from the source introduces new genetic variation that may prevent local adaptation (Kawecki et al., 2008; Olson-Manning et al., 2012). Therefore, we believe that samples from regions that are only seasonally suitable for DBM (with EI=0 and the populations being subject to seasonally punctuated local selection by climatic factors) should be excluded in the analysis of correlation between genetic variation and climatic variables.

Lines 105-106, 319-323 – why use A1B and A2? These are very old and outdated scenarios

We have replaced A1B and A2 with the updated scenarios, RCP2.6, RCP4.5, RCP6.0 and RCP8.5. (Line 95)

Line 216 – specifically what is verified is that a specific gene responds to climatic variation spatially, not specifically to climate change. This may have climate change implications but need to be careful about wording.

The sentence has been revised to: “...we verified that a specific gene, *PxCad*, represented a temperature-sensitive responder to climatic change,...”. (Line 207)

Line 253 – need to cite worldclim

Fick, S.E. and R.J. Hijmans, 2017. WorldClim 2: new 1km spatial resolution climate surfaces for global land areas. *International Journal of Climatology* 37: 4302-4315.

The citation has been included as suggested. (Line 271)

Line 336 – why use CSIRO GCM?

We have replaced A1B and A2 with the updated scenario RCP8.5, so that the CSIRO GCM has been replaced with NorESM1-M Global Climate Model (GCM), which is compatible with the CLIMEX model and includes RCP2.6, RCP4.5, RCP6.0 and RCP8.5 data from the Worldclim.

Fig 2 – what does the white in the figures represent? I’m also having difficulty with the scales. In the text it is noted that “relatively high levels of EGI” are need for habitat suitability. Relative to what? What’s a high level of EGI?

The white in the figures represents regions that are seasonally suitable for DBM to grow and develop. The description of Fig. 2 in the text has been revised to: “Under climate-change scenario RCP8.5, the challenges to most populations (Fig. 2b) can be

moderated by their genetically adaptive potential (Fig. 2a). Therefore, most DBM populations will maintain their pest status in the context of future climate beyond 2050, without dramatic change to EGI-based habitat suitability in most of the world (Fig. 2c).” (Line 119-124)

Reviewer #3 (Remarks to the Author):

I have reviewed the manuscript entitled: “Large-scale genome-wide study reveals climate adaptive variability in a cosmopolitan herbivore”. The authors used a multi-pronged approach to study adaptive variation in the diamondback moth. They have a SNP dataset from 357 individuals in 78 sampling sites distributed across the globe. They then asked the questions whether there are significant SNP-environment associations and how local allele frequencies of significantly associated SNPs should change under future climate conditions. To answer these questions, the authors employed a combination of landscape genomics, predictions of habitat suitability, and gene-editing techniques to confirm the function of genes that SNPs were located in. The results suggest that there are > 3000 SNPs significantly associated to temperature variables, and that the genetic vulnerability of many populations is likely low enough for the species to thrive and maintain its status as a pest species.

The manuscript is generally well written. Although I can’t comment on the genome editing work, I was particularly pleased to see the confirmation that indeed there is a temperature-sensitive response. However, I do have some reservations with the landscape genomics work.

First, some of the bioclimatic variables are highly correlated, and I was surprised that the authors did not make any effort to downselect a subset out of the 19 variables. There are several ways to go about this, based on for instance variance inflation factors or just cross-correlation coefficients. In addition, surely climate variables are not the only factors influencing adaptive variation in these moths. Some of the factors will be hard to capture, but others may be readily available as GIS layers. The focus on future climate change dictates that only climate variables can be included in a model under current climate conditions, but the conclusion that temperature is the most important factor in determining climate-associated genomic variation in this moth (line 207-208) seems rather overstated. Many other drivers have not been taken into account, and of those that were, the majority is temperature-related.

Following this comment, a subset of 12 variables has now been selected out of the 19 variables, and used for the analysis of genetic vulnerability and adaptive potential based on generalized dissimilarity modeling (GDM). The updated results are presented in Figs.1 and 2, and Supplementary Fig. 2. For this study, we focus on climate-induced adaptive variation based on 19 climate variables from WorldClim exclusively, given that these 19 variables are the only set of predictable factors that

can be applied to study genetic vulnerability and adaptive potential for our target species.

As suggested, the conclusion in the text has been revised as follows: “Our analyses with the nuclear SNPs from geographically distributed samples goes further in indicating the effects of both temperature- and precipitation-related variables on the climate-associated genetic variation in DBM populations worldwide.” (Line 184-186)

Second, the future climate scenarios A1B, A2 etc are very outdated. These were used in the CMIP3 / IPCC AR4 scenarios published in 2007. In the meantime, Representative Concentration Pathways have substituted the A-scenarios, and as a matter of fact even the RCP are by now obsolete and replaced by CMIP6 Shared Socioeconomic Pathways for IPCC’s AR6. Although I don’t think the overall conclusions will change, the SSPs generally predict larger temperature changes (compared to the RCPs), which could change the maps at least to some extent. The bioclimatic variables for a variety of SSP scenarios are available for download from WorldClim. I would suggest updating the maps for a moderate and a more severe SSP scenario.

As suggested, the more updated RCP emission scenarios have been used for our predictions. The estimation of DEI and EGI was based on RCP 8.5 exclusively because it is the only emission scenario available in CliMond (<https://www.climond.org/>). The estimation of genetic offset was based on RCP 2.6, RCP 4.5, RCP 6.0, and RCP 8.5 from WorldClim with all of the results being presented in Supplementary Fig. 2. SSP scenarios are not applicable for our case, since they are not available in CliMond (<https://www.climond.org/>) for incorporating into the CLIMEX model.

Third, I’m unsure about the conceptual approach taken by combining predicted changes in habitat suitability with those in allele frequencies. Both predictions of habitat suitability and of the genomic makeup of populations in the future assume that current relationship between environment and allele frequencies or between environment and species occurrence remain constant over time. If, for instance, temperature changes in a given area are predicted to be severe, the allele frequencies of temperature-sensitive genes are also predicted to change dramatically. At the same time, habitat suitability may change drastically, but suitability and local adaptations are highly related. If habitat becomes less suitable, considerable allele frequency changes are also expected to be required. Thus, simply multiplying the two responses to come up with a new ‘eco-genetic’ index appears to be cutting a lot of corners, and does not improve our understanding of the vulnerability or persistence of the species in the future. Identifying areas where conditions in the future are not mirrored in any of the sites where the species was sampled may provide a more realistic idea of where populations are at risk.

The EI-based habitat suitability of DBM proposed by Zalucki and Furlong (2011) is

based on DBM's physiological traits and represents the population growth potential. Without taking inter-population genetic variation into account, there is no clue on how populations differ in their capacity to adapt to climate change. In this study, we develop a new habitat suitability index (EGI), serving as a calibration of the EI-based habitat suitability based on the integration of species' physiological traits and inter-population adaptive variation. We firstly used the genetic offset (GO) method to examine spatial regions where populations are challenged by the disruption of gene-environment relationships between current and projected future climates. We then followed the idea presented in Zalucki and Furlong (2011), where habitat suitability is represented by multiplying population growth index and stress index, to develop a new habitat suitability index (EGI) by taking population-specific genetic vulnerability (GO) into account as a "stress index". The combination of such a "stress index" (GO) and eco-climate index (i.e., DEI in this study) therefore represents intraspecific variation of responses to future climate, by integrating both genetic and physiological factors.

Fourth, the authors used gradient forests (GF) to create maps of genetic variation under current and future climate conditions, and to compute a map representing 'genetic vulnerability'. A major caveat of GF in landscape genomic approaches (i.e. analyses of population structure in a spatial context) is that it cannot account for the effects of isolation by distance or isolation by resistance. Yet, teasing apart the effects of neutral processes from those of selection is a crucial component in landscape genomics, and key to the current paper. By focusing on adaptive genetic variation and genetic vulnerability, the authors are specifically interested in that part of genetic variation that cannot be attributed to neutral divergence between populations. To this end, generalized dissimilarity modeling is a more appropriate method, because it can directly consider the effects of distance and variation in habitat permeability (see also Box 1 in Fitzpatrick and Keller, 2015). I realize that the authors only included the SNPs identified in Bayenv/Sambada/LFMM, thus focusing on those SNPs that are putatively under selection, at least partly mitigating the problem. Yet, GF starts all over again with these SNPs, not accounting for neutral processes that very likely also contribute to population divergence.

As suggested, we have replaced the gradient forests (GF) with the generalized dissimilarity modeling (GDM) to identify SNPs that are responsible for climate-associated genomic variation. The complete process has been presented in the revised Methods as follows:

Line 298-334

“Prediction of climate-associated genomic variation based on generalized dissimilarity modelling

Generalized dissimilarity modelling (GDM)²⁰, a distance-based model, can account for the nonlinear relationship between genetic variation and environmental / geographical factors, and has been recently used to map ecological adaptation from genomic data under current and future climates⁴. Firstly, we examined spatially

explicit selection process for each of the putative adaptive SNPs using GDM⁴, with the R package *gdm*⁶². We subsampled the genetic dataset to include only populations with sample size ≥ 5 to obtain accurate allele frequencies. An uncorrelated subset of 12 climate variables (bio01, bio02, bio03, bio05, bio07, bio08, bio09, bio12, bio14, bio15, bio18, bio19 in Supplementary Table 6) with a pairwise Pearson correlation coefficient less than 0.8 in 60 sample sites was used in GDM. Pairwise F_{ST} matrix among 60 populations were calculated for each of the 3,307 SNPs using the R package *hierfstat*⁶³, and rescaled between 0 and 1. Geographical distance in the GDM was based on Euclidean distance as the thirteenth variable to test whether genetic variation across environmental gradients was better explained by climate variables than geographical distance, which effectively acts as a screening for SNPs that may respond predominantly to neutral genetic process including isolation by distance⁶⁴. The relative importance of the 12 climate variables and geographical distance was ranked based on the fitted I-Splines in GDM (Fig. 1b). The maximum value of each variable in the fitted I-Splines was rescaled between 0 and 1. Those SNPs with geographical distance ranking as one of the top 3 important variables were excluded in the following GDM analysis. In addition, we randomly sampled 200 SNPs as a “reference group” to test its explainable proportion of the GDM deviance. According to our GDM record, the reference SNP group accounted for 11.2% of the GDM deviance for the entire model, so that those SNPs with a $< 11.2\%$ contribution to the GDM deviance were also excluded in further GDM analysis. After additional filtering, 419 of 3307 SNPs were retained. The 419-SNP-based genetic distance matrix was further integrated with geographical distance and climate variables to be used in the entire GDM model. The entire GDM model explained 41.2% of the deviance for the 419 SNPs. To predict the climate adaptation of DBM, we finally retrieved current climate variables at 61,655 gridded points across the world from WorldClim, using ArcGIS 10.2. The *gdm.transform* function was used to predict and map the pattern of climate-associated genomic variation along environmental gradients across the world (Fig. 1c). The genetic turnover was summarized using a principal component analysis (PCA), with the top three components transformed for visualization in a red-green-blue (RGB) colour scale as suggested in the GDM manual⁶². Loadings based on the principal components indicate the direction and magnitude of association with adaptation to different predictors (Fig. 1c). The climate-associated genomic variation along environmental gradients in DBM across the world was visualized, with similar pattern of genetic composition at climate-adaptive loci illustrated by similar colours (Fig. 1c).”

Also, the results based on the redefined SNP dataset have been presented in the updated Fig. 1c, Fig. 2, and Supplementary Fig. 2.

Minor comments

1. 40 – ‘increased’ -> the effects could go both ways, so why focus on an increase only?

The sentence has been revised to: “Human-induced climate change, especially gradual changes in temperature and precipitation¹, is impacting species’ survival and distribution².” (Line 34-35)

l. 44 – ‘adaptive plasticity’ -> although the level of plasticity can be adaptive by itself, I think that’s not what the authors mean to say here. Rather ‘adaptive potential’.

We have replaced the ‘adaptive plasticity’ with ‘adaptive potential’. (Line 38)

l. 229 – Drop one of the occurrences of ‘on average’

We have dropped one of the occurrences of ‘on average’. (Line 247)

l. 231 – ‘*Arabidopsis thaliana*’ in italics

Arabidopsis thaliana has been italicized. (Line 248)

l. 270 – What’s the reason for the low threshold for p after FDR correction? Or does the mentioned p-value correspond to a value before correction?

We chose a relatively strict z-score and p value based on the methodology from previous publications, such as Abebe et al. (2015) and De Kort et al. (2014).

l. 315 – ‘disruption of gene-environment relationships’ -> As a matter of fact, the major assumption of the approach being used is that gene-environment relationships will remain the same in the future. Rather, the spatial patterns will change because of climate change. If gene-environment relationships would be disrupted, there is no way that we can make any predictions of the spatial distribution of alleles/genetic variation.

The sentence has been rewritten as: “we used the method developed by Fitzpatrick and Keller⁴ to measure the “genetic offset”, which represents the disruption of gene-environment relationships subject to future climates.” (Line 339-341)

l. 334 – ‘distribution’ -> ‘distributions’

‘distribution’ has been revised to ‘distributions’. (Line 351)

Supplements page 11 – add captions for Tables 6, 8, 9, 10

Captions for Tables 6, 8, 9, 10 have been added.

Reviewer #4 (Remarks to the Author):

Chen et al present a manuscript investigating climate adaptation and the potential to adapt to future environments in a worldwide dataset of the herbivore *Plutella xylostella* (diamondback moth). This study represents a statistical re-analysis of a genomic dataset published in Nature Communications by the same team earlier this year. The authors go beyond a simple scan for adaptive variation in two ways: they use the results of the adaptation analysis to predict the potential of dealing with future climatic conditions and they verify one of the top genes using genome editing. This is an interesting approach and distinguishes this study from many other studies of local adaptation. The main conclusion of this study is that the diamondback moth will maintain its global pest status at least until 2050.

I am not an expert on pests, conservation or *Plutella xylostella* specifically so my comments are from a general population genomics perspective. I am convinced that this article will ultimately become a valuable addition to the record but I do not feel like I have the overview over the specific area to make a statement about the potential impact.

The main part of the manuscript is relatively short and it quickly becomes clear that this is mainly a re-analysis of data that the authors presented in another paper. For a lot of the details about the underlying dataset, the reader will have to refer to another publication. It is generally positive to re-use published data but some descriptive information would be useful to the readers of this article. For example, the geographic distribution of populations from Figure S1 could easily be included in the main figures. How many locations and individuals were in the final dataset etc. Sometimes the authors could add more information in the Results section, e.g. when they are introducing the different indices and metrics used, otherwise the reviewer has to jump between different sections of the manuscript.

We have substantially rewritten the manuscript by adding more information, especially in the sections of Introduction, Discussion, and Methods. For example, we have added some information about sampling and generation of genomic data in the Methods (as shown in the following two paragraphs) as well as in the legend of Figure S1 so that the reader will not need to refer to another publication.

Line 234-259

“Genomic data

The foundational resource for this study is a dataset of 40,107,925 nuclear SNPs sequenced from a worldwide sample of 532 DBM individuals collected in 114 different sites based on our previous project¹⁵. DNA was extracted from each of the 532 individuals using DNeasy Blood and Tissue Kit (Qiagen, Hilden, Germany) following the manufacturer’s protocol, and eluted from the DNeasy Mini spin column in 200 µl TE buffer. Genomic sequencing was performed with Illumina HiSeq 2000 at BGI, Shenzhen, China, to produce 90 bp paired-end reads for every individual. Using custom scripts, raw reads were processed to filter out poor reads with 10 ambiguous

“N” bases, >40% low-quality bases, or identical sequences at the two ends and obtain clean reads. Stampy (v1.0.27) was employed to map the clean reads onto the DBM reference genome (v2)²² using default parameters. SNP calling was then performed using the GATK HaplotypeCaller with parameters `--emitRefConfidence GVCF --variant_index_type LINEAR--variant_index_parameter 128,000`. The generated 40,107,925 nuclear SNPs present one variant on average in every six bp of the reference genome, which is the densest variant map for any organism, including the recently released data on human⁵⁸ and *Arabidopsis thaliana* genome sequences⁵⁹. The SNP dataset is available at <https://www.ebi.ac.uk/ena> with the accession code PRJEB24034.

In the present study, to investigate the genetic variation associated with climate change, we excluded samples from the regions that are only seasonally suitable for growth of DBM with an Ecoclimatic Index (EI = 0)¹⁶ because in these regions populations are unlikely to receive perennially unpunctuated selection by local environmental variables and genetic variation cannot be continuously passed on to future generations over years. The retained samples included 372 individuals from 78 sampling sites in the year-round persistence regions of DBM across the world. These samples were collected from different continents, with 13 samples from Africa, 29 from Asia, 5 from Europe, 13 from North America including Hawaii, 12 from South America, and 6 from Oceania (Supplementary Fig. 1 and Supplementary Table 1).”.

What is crucial to the conclusions is the solid identification of candidate loci. The authors are using three different methods to identify these loci but they include all candidates in their downstream analysis. In my impression, the authors apply different stringency criteria on the different approaches. Using hard thresholds on each of these three very different statistics is difficult to interpret and, consequently, the number of candidates identified per method differ by a factor of almost two. Approaches like LFMM and Bayenv2 which are using a user-defined number of iterations can lead to somewhat noisy results in local optima, especially in situations of small sample sizes (like the <5 individuals per population on average in this case which makes frequency estimates quite noisy). For LFMM, the authors use the median of five independent runs which should account for some of the noise. For Bayenv2, however, they only use a single run with a comparatively low number of iterations (I have usually seen at least an order of magnitude more iterations in the literature).

Thank you for this comment, it led us to detect a numerical error. Checking the setting of our Bayenv2 operation we found that the number of iterations used was 100,000 rather than 10,000 as previously stated. We set our number of iterations based on the methodology from previous publications, such as Lotterhos and Whitlock (2015), and Stuck et al. (2017). We have corrected the number in the Methods accordingly.

I am not sure if fully understand the motivation for restricting the GF analysis to the candidate SNPs, but I also have not read the original publication on that. The candidates were identified by association between allele frequency and environmental

variables. The GF analysis now again identifies between the allele frequencies at these SNPs and environmental variables which feels circular. How would the results differ if all SNPs were tested? Would the same environmental variables be the driving forces behind neutral population structure?

Disentangling the effects of neutral processes from those of selection is a crucial component in landscape genomics, and key to our study. By focusing on adaptive genetic variation and genetic vulnerability, we are specifically interested in the genetic variation that cannot be attributed to inter-population neutral divergence. Therefore, the first step in this study was to exclude SNPs that were not identified by Bayenv, Sambada, and/or LFMM, three robust models that provide statistical framework for controlling the confounding effects of neutral genetic structure (Rellstab et al., 2015). We focus on those SNPs that are putatively under selection, which at least partly tease apart the effects of neutral processes from those of selection. To further disentangle demographic signals from true adaptive signals, we have replaced the gradient forests (GF) with the generalized dissimilarity modeling (GDM), a more appropriate method that can directly consider the effects of distance and variation in habitat permeability (Fitzpatrick and Keller, 2015). The entire process for identifying climate-associated genomic variation is described in the revised Methods as follows:

Line 273-334

“Climate associated genomic variation

Identification of SNPs under climate selection

Three models, Sambada¹⁷, latent factor mixed model (LFMM)¹⁸ and Bayenv 2¹⁹, were used to identify putatively adaptive loci associated with climate variables. Sambada applies logistic regression models to identify associations between specific genetic markers and environmental variables¹⁷. Simple univariate and multivariate logistic regression models for each climate variable were performed. A single SNP was considered to be a candidate locus when the log-likelihood ratios (G scores) and/or Wald scores were significant with Bonferroni correction at a 99% confidence level. LFMM is developed based on population genetics, ecological modeling, and statistical analysis to identify the candidate loci that are highly correlated with environmental variables¹⁸. SNPs showing an association with climate variables were identified based on z-scores, which was computed using 10,000 cycles and 5,000 sweeps for burn-in. We used R package 3.6.2 to run the median z-scores of 5 runs and re-adjusting p-values with FDR correction. SNP with median z-scores above the absolute value of 4 and corresponding to P value $< 10^{-5}$ were considered as significant locus. In Bayenv 2, a covariance matrix based on putatively neutral markers is used as a null model to control demographic effects when testing relationships between the genetic differentiation and a given environmental variable¹⁹. We randomly sampled SNPs at 200 SNP intervals from the SNP dataset. A total of 117,887 SNPs with loose linkage disequilibrium were obtained for developing the covariance matrix, which was estimated with 100,000 iterations. We then assessed the correlations between individual SNPs and 19 climate variables at 100,000 Markov chain Monte Carlo (MCMC) for Bayes factor analysis. The results were presented as Bayes factors (BFs).

A $\log_{10}(\text{BF})$ value > 1.5 is usually considered as high support for a model where environmental parameters have significant effects on allele frequencies⁶¹. A total of 3,307 putative adaptive SNPs were identified by at least one of the three models (Samβada, LFMM and Bayenv 2; Supplementary Tables 3, 4 and 5).

Prediction of climate-associated genomic variation based on generalized dissimilarity modelling

Generalized dissimilarity modelling (GDM)²⁰, a distance-based model, can account for the nonlinear relationship between genetic variation and environmental / geographical factors, and has been recently used to map ecological adaptation from genomic data under current and future climates⁴. Firstly, we examined spatially explicit selection process for each of the putative adaptive SNPs using GDM⁴, with the R package *gdm*⁶². We subsampled the genetic dataset to include only populations with sample size ≥ 5 to obtain accurate allele frequencies. An uncorrelated subset of 12 climate variables (bio01, bio02, bio03, bio05, bio07, bio08, bio09, bio12, bio14, bio15, bio18, bio19 in Supplementary Table 6) with a pairwise Pearson correlation coefficient less than 0.8 in 60 sample sites was used in GDM. Pairwise F_{ST} matrix among 60 populations were calculated for each of the 3,307 SNPs using the R package *hierfstat*⁶³, and rescaled between 0 and 1. Geographical distance in the GDM was based on Euclidean distance as the thirteenth variable to test whether genetic variation across environmental gradients was better explained by climate variables than geographical distance, which effectively acts as a screening for SNPs that may respond predominantly to neutral genetic process including isolation by distance⁶⁴. The relative importance of the 12 climate variables and geographical distance was ranked based on the fitted I-Splines in GDM (Fig. 1b). The maximum value of each variable in the fitted I-Splines was rescaled between 0 and 1. Those SNPs with geographical distance ranking as one of the top 3 important variables were excluded in the following GDM analysis. In addition, we randomly sampled 200 SNPs as a “reference group” to test its explainable proportion of the GDM deviance. According to our GDM record, the reference SNP group accounted for 11.2% of the GDM deviance for the entire model, so that those SNPs with a $< 11.2\%$ contribution to the GDM deviance were also excluded in further GDM analysis. After additional filtering, 419 of 3307 SNPs were retained. The 419-SNP-based genetic distance matrix was further integrated with geographical distance and climate variables to be used in the entire GDM model. The entire GDM model explained 41.2% of the deviance for the 419 SNPs. To predict the climate adaptation of DBM, we finally retrieved current climate variables at 61,655 gridded points across the world from WorldClim, using ArcGIS 10.2. The *gdm.transform* function was used to predict and map the pattern of climate-associated genomic variation along environmental gradients across the world (Fig. 1c). The genetic turnover was summarized using a principal component analysis (PCA), with the top three components transformed for visualization in a red-green-blue (RGB) colour scale as suggested in the GDM manual⁶². Loadings based on the principal components indicate the direction and magnitude of association with adaptation to different predictors (Fig. 1c). The climate-associated genomic

variation along environmental gradients in DBM across the world was visualized, with similar pattern of genetic composition at climate-adaptive loci illustrated by similar colours (Fig. 1c).

Corresponding results based on the redefined SNP dataset (i.e., 419 SNPs) are presented in the updated Fig. 1.

I do not feel qualified to comment on the methodology of the CRISPR/Cas9 genome editing approach. I am only wondering why we do not see any control populations (comparing WT and mutant without heat or cold stress) in the figures (Fig 5 and S8). According to the methods section, there was such a control group.

In Fig. 5 and Fig. S8, we compared the response of DBM to extreme temperature conditions between WT (G88), which serves as a control strain, and mutant strain (G88-Cad, PxCad-deficient).

Minor comments:

L61: I think you are only functionally testing a single gene not “genes”.

We have revised the sentence: “we functionally test a temperature-related gene to reveal its role in climatic adaptation in DBM.”. (Line 58-59)

L151-154: If there is more literature showing temperature related functions for these genes, please add the references to the text or Table S7.

For those identified genes that have been previously documented as having temperature-related functions, we have provided some references (24-31) in the text, and have added those references to Table S7.

L164: Does “signals of adaptive variation” mean “presence of the selected allele”? Wouldn’t that be a bit circular since the exact same information was included in the scan for adaptive variants so maybe the absence of the variant actually contributes to the worldwide signal of adaptation.

“Signals of adaptive variation” has been revised to: “Allelic frequency”. (Line 144)

L177/178: Change “the PxCad gene tended...” to “the expression of the PxCad gene...”?

“the PxCad gene tended...” has been changed to “the expression of the *PxCad* gene...” (Line 154)

Fig S8: Please display standard errors in this figure

We presented the observed survival of 20 individuals with four replicates for each of the treatments, rather than the mean survival. So, we did not display the standard errors in Fig. S8 (Fig. S5 in the revised version).

Tables S3-S5: Please add the values of the scores underlying these candidates to the tables

We have added the values of the scores in the Table S3-S5 as suggested.

References

- Abebe TD, Naz AA, Léon J. Landscape genomics reveal signatures of local adaptation in barley (*Hordeum vulgare* L.). *Frontiers in Plant Science*, 2015, 6: 813.
- Battle, E. et al. The transcription factor snail is a repressor of E-cadherin gene expression in epithelial tumour cells. *Nat Cell Biol.* 2, 84-89 (2000).
- Behrens, J., Löwrick, O., Klein-Hitpass, L. & Birchmeier, W. The E-cadherin promoter: functional analysis of a GC-rich region and an epithelial cell-specific palindromic regulatory element. *Proc Natl Acad Sci U S A.* 88, 11495-11499 (1991).
- Bulgakova, N. A., Klapholz, B. & Brown, N. H. Cell adhesion in *Drosophila*: versatility of cadherin and integrin complexes during development. *Curr Opin Cell Biol.* 24, 702-712 (2012).
- Chapman JW, Reynolds DR, Smith AD, et al. High-altitude migration of the diamondback moth *Plutella xylostella* to the UK: a study using radar, aerial netting, and ground trapping. *Ecological Entomology*, 2002, 27(6): 641-650.
- De Kort H, Vandepitte K, Bruun HH, et al. Landscape genomics and a common garden trial reveal adaptive differentiation to temperature across Europe in the tree species *Alnus glutinosa*. *Molecular ecology*, 2014, 23(19): 4709-4721.
- Dumas P, Morin MD, Boquel S, Moffat CE, Morin PJ. Expression status of heat shock proteins in response to cold, heat, or insecticide exposure in the Colorado potato beetle *Leptinotarsa decemlineata*. *Cell Stress Chaperones.* 2019, 24(3):539-547.
- Fitzpatrick M C, Keller S R. Ecological genomics meets community-level modelling of biodiversity: Mapping the genomic landscape of current and future environmental adaptation. *Ecology letters*, 2015, 18(1): 1-16.
- Fitzpatrick MJ, Edelsparre AH. The genomics of climate change. *Science*, 2018, 359(6371): 29-30.
- Furlong MJ, Wright DJ, Dossall LM. Diamondback moth ecology and management: problems, progress, and prospects. *Annual review of entomology*, 2013, 58: 517-541.
- Guo, Z. et al. The midgut cadherin-like gene is not associated with resistance to *Bacillus thuringiensis* toxin Cry1Ac in *Plutella xylostella* (L.). *J Invertebr Pathol.* 126, 21-30 (2015).
- Huang J, Li J, Zhou J, Wang L, Yang S, Hurst LD, Li WH, Tian D. Identifying a large number of high-yield genes in rice by pedigree analysis, whole-genome sequencing, and CRISPR-Cas9 gene knockout. *Proc Natl Acad Sci U S A.* 2018,115(32):E7559-E7567.
- Kawecki TJ. Adaptation to marginal habitats. *Annual Review of Ecology, Evolution, and Systematics*, 2008, 39: 321-342.
- Ke F, You S, Huang S, et al. Herbivore range expansion triggers adaptation in a subsequently-associated third trophic level species and shared microbial symbionts. *Scientific reports*, 2019, 9(1): 1-10.
- Liu Z, Fu S, Ma X, et al. Resistance to *Bacillus thuringiensis* Cry1Ac toxin requires

- mutations in two *Plutella xylostella* ATP-binding cassette transporter paralogs. *PLoS pathogens*, 2020, 16(8): e1008697.
- Lotterhos KE, Whitlock MC. The relative power of genome scans to detect local adaptation depends on sampling design and statistical method. *Molecular Ecology*, 2015, 24(5): 1031-1046.
- Nguyen C, Bahar MH, Baker G, Andrew NR (2014) Thermal tolerance limits of diamondback moth in ramping and plunging assays. *PLoS One*, 9, e87535.
- Olson-Manning CF, Wagner MR, Mitchell-Olds T. Adaptive evolution: evaluating empirical support for theoretical predictions. *Nature Reviews Genetics*, 2012, 13(12): 867-877.
- Park, Y., Herrero, S. & Kim, Y. A single type of cadherin is involved in *Bacillus thuringiensis* toxicity in *Plutella xylostella*. *Insect Mol Biol.* 24, 624-633 (2015).
- Pigott, C. R. & Ellar, D. J. Role of receptors in *Bacillus thuringiensis* crystal toxin activity. *Microbiol. Mol. Biol. Rev.* 71, 255-281 (2007).
- Rellstab C, Gugerli F, Eckert A J, et al. A practical guide to environmental association analysis in landscape genomics. *Molecular Ecology*, 2015, 24(17): 4348-4370.
- Sejian V, Gaughan J, Baumgard L, Prasad C, eds. *Climate Change Impact on Livestock: Adaptation and Mitigation*. Springer, 2015, p.515.
- Shelton AM, Robertson JL, Tang JD, et al. Resistance of diamondback moth (*Lepidoptera: Plutellidae*) to *Bacillus thuringiensis* subspecies in the field. *Journal of Economic Entomology*, 1993, 86(3): 697-705.
- Shelton, A., Cooley, R., Kroening, M., Wilsey, W. & Eigenbrode, S. Comparative analysis of two rearing procedures for diamondback moth (*Lepidoptera: Plutellidae*). *J. Entomol. Sci.* 26, 17-26 (1991)
- Stuck S. et al. High performance computation of landscape genomic models including local indicators of spatial association. *Molecular Ecology Resources*, 2017, 17: 1072-1089.
- Talekar NS, Shelton AM. Biology, ecology, and management of the diamondback moth. *Annual review of entomology*, 1993, 38(1): 275-301.
- Wei SJ, Shi BC, Gong YJ, et al. Genetic structure and demographic history reveal migration of the diamondback moth *Plutella xylostella* (*Lepidoptera: Plutellidae*) from the southern to northern regions of China. *PloS one*, 2013, 8(4): e59654.
- Wu, Q. & Maniatis, T. A striking organization of a large family of human neural cadherin-like cell adhesion genes. *Cell* 97, 779-790 (1999).
- You M, Ke F, You S, et al. Variation among 532 genomes unveils the origin and evolutionary history of a global insect herbivore. *Nature communications*, 2020, 11(1): 1-8.
- Zalucki M. & Furlong M. in *International Workshop on Management of the Diamondback Moth and Other Crucifer Insect Pests (6th) Predicting outbreaks of a migratory pest: an analysis of DBM distribution and abundance 8-14 (AVRDC: The World Vegetable Centre, 2011)*.
- Zhan S, Zhang W, Niitepõld K, Hsu J, Haeger JF, Zalucki MP, Altizer S, de Roode JC, Reppert SM, Kronforst MR. The genetics of monarch butterfly migration and warning coloration. *Nature*. 2014,514(7522):317-321.
- Zhang Q, Denlinger DL. Molecular characterization of heat shock protein 90, 70 and

70 cognate cDNAs and their expression patterns during thermal stress and pupal diapause in the corn earworm. *Journal of Insect Physiology*, 2010, 56(2): 138-150.

Reviewer comments, further round - -

Reviewer #2 (Remarks to the Author):

The authors have done a reasonably good job of addressing my concerns and those of other reviewers (in my view). There are however a few remaining issues/edits that require attention:

Line 24 – “78 sites”

Line 27 – genomic variation?

Line 32 – this final sentence of the abstract seems like a very broad conclusion, is there any way to specify? Maybe relate to pest management or arthropods? In my original review I suggested specifically explaining how the results pertain to management. Avoiding this in the abstract (as done in the revision) is fine, but then a clearer and stronger implication is needed.

Lines 66-68/ 252-254 – it strikes me that two reviewers highlighted this as not clearly justified in the manuscript. While the reviewer response better articulates the reasoning behind this, why not include some version of this in the manuscript?

Line 195 – strictly speaking this isn't what the results show, you've shown genomic associations with climate that suggest warming may elicit region-specific responses... but you haven't presented “region-specific genomic responses of DBM to the changing climate”

Line 284 – which r package?

Reviewer #3 (Remarks to the Author):

I have reviewed a revised version of the manuscript “Large-scale genome-wide study reveals climate adaptive variability in a cosmopolitan pest”. Whereas I think the manuscript has improved considerably, I still think that it is not ready for publication.

I. My main concern revolves around an issue I described previously. For clarity, I have pasted my previous review and the authors' response below.

Third, I'm unsure about the conceptual approach taken by combining predicted changes in habitat suitability with those in allele frequencies. Both predictions of habitat suitability and of the genomic makeup of populations in the future assume that current relationship between environment and allele frequencies or between environment and species occurrence remain constant over time. If, for instance, temperature changes in a given area are predicted to be severe, the allele frequencies of temperature-sensitive genes are also predicted to change dramatically. At the same time, habitat suitability may change drastically, but suitability and local adaptations are highly related. If habitat becomes less suitable, considerable allele frequency changes are also expected to be required. Thus, simply multiplying the two responses to come up with a new 'eco-genetic' index appears to be cutting a lot of corners, and does not improve our understanding of the vulnerability or persistence of the species in the future. Identifying areas where conditions in the future are not mirrored in any of the sites where the species was sampled may provide a more realistic idea of where populations are at risk.

Authors' response: The EI-based habitat suitability of DBM proposed by Zalucki and Furlong (2011) is based on DBM's physiological traits and represents the population growth potential. Without taking inter-population genetic variation into account, there is no clue on how populations differ in their capacity to adapt to climate change. In this study, we develop a new habitat suitability index (EGI), serving as a calibration of the EI-based habitat suitability based on the integration of species' physiological traits and inter-population adaptive variation. We firstly used the genetic offset (GO) method to examine spatial regions where populations are challenged by the disruption of gene-environment relationships between current and projected future climates. We then followed the idea presented in Zalucki and Furlong (2011), where habitat suitability is represented by multiplying population growth index and stress index, to develop a new habitat suitability index (EGI) by taking population-specific genetic vulnerability (GO) into account as a “stress index”. The combination of such a “stress index” (GO) and eco-climate index (i.e., DEI in this study) therefore represents intraspecific variation of responses to future climate, by

integrating both genetic and physiological factors.

I think there is a misconception of the EI and how it is calculated in the paper by Zalucki and Furlong (2011) - or I don't quite get what the authors mean by saying that they are taking GO as a stress index... EI is described as the growth index multiplied by a stress index SI, where SI is defined as heat/drought stress. Hence, SI is *already* included in EI. By first including temperature and precipitation as stress factors in EI, then also using them in calculating the GO, and subsequently multiplying EI by GO, the climate factors are included twice in the EGI. From a biological mechanistic point of view, habitat is suitable because individuals are (sufficiently) well adapted to its associated conditions. Thus, the frequencies of relevant adaptive alleles dictate whether a population will be able to persist, and this is what the authors have already modeled by computing GO. To get an even better sense of vulnerability, it would be more useful to map current and future allele frequencies, and assess vulnerability as a function of the presence in area A of alleles predicted to be necessary for adaptations under future conditions in conjunction with estimates of functional connectivity with regions B/C/D, where current conditions are more similar to future conditions in area A. In addition, are there any regions where future conditions do not match current conditions observed anywhere within the current distribution of DBM? These are areas where our confidence in the predictions are very low - we basically don't know how DBM will respond to completely novel environmental conditions.

II. Another, smaller, concern is that there is much emphasis on the role of climate in shaping genetic variation. However, these are the only variables tested for associations. This is fine, but the only thing that can be concluded is that environment (and not specifically climate) is likely driving adaptive genetic variation in DBM. Indeed, almost 60% of the variation was unaccounted for in GDM models.

III. Much more information is needed on how the CLIMEX models were generated. What were the settings? What input data did you use?

Minor comments:

I. 46 "accumulate" - what do you mean? Are you referring to new mutations, or based on standing variation?

I. 52 "important factors..." - not sure what you mean. What "factors"? Don't you mean to say that you want to identify the environmental variables that are associated with SNP variation (or rather, the other way around)?

I. 69 "357 individuals ... from 75 different sites" - what are the correct numbers? On line 254 you mention 372 individuals from 78 sites.

I. 80 "(none cross-correlated)" - this is incorrect; any variables with cross-correlations < 0.8 were retained.

I. 90 "new metric" - please drop 'new', because it isn't. This has been done many times, including by Fitzpatrick and Keller, the paper you are referring to.

I. 250 "with climate change" - drop 'change', since you are looking at associations with climate variables, not with change.

I. 267-270 Please explain that you downselected variables using Pearson cross-correlation coefficients. Please also justify why you retained one variable over another in a pair of correlated ones.

Reviewer #4 (Remarks to the Author):

I thank the authors for their revised version of the manuscript. The updated manuscript contains a large proportion of additional information which resolves a lot of aspects that were unclear in the previous version.

Unfortunately, the authors seem to have decided to not include additional data or repeat analyses to address my previous comments (but also comments by reviewer 1). In some cases they simply refer to other articles but simply because previous studies have used certain approaches does not mean they chose the most reliable option.

In my previous review, I noted that the three different methods used to identify candidate loci were used with different stringency when selecting the outliers. Especially bayenv2 was only run once and with a relatively low number of MCMC iterations. The authors have now corrected the number of iterations but they still only use a single run (as opposed to five for LFMM). It is well known that the results of such methods can be very noisy in single runs and therefore multiple runs are suggested to obtain reliable results (see e.g. Blair et al 2014: On the stability of the Bayenv method in assessing human SNP-environment associations).

In another comment, I asked whether there are differences in the survival rates between the different strains even without temperature treatment or stress. The authors response was only that the WT was used as control. I may have missed this but it should be made clear that the mutant shows reduced survival rates only under extreme temperatures.

Reviewer 1 correctly highlighted that it would be more valuable to test gene expression directly after treatment while it was measured after recovery in this case. That way, we cannot be sure that the effects seen in this case are a direct consequence of the treatment.

I second reviewer 1 in their impression that the manuscript seems to be relatively narrow and quickly zeroes in on PxCad as candidate gene. There is not much discussion on alternative explanations for the results or potentially confounding factors.

Finally, I would like to ask the authors to include a lot of the additional information and references from the response letter into the main manuscript.

REVIEWER COMMENTS

Reviewer #2 (Remarks to the Author):

The authors have done a reasonably good job of addressing my concerns and those of other reviewers (in my view). There are however a few remaining issues/edits that require attention:

Line 24 – “78 sites”

As suggested, we have changed “78 geographical sites” to “78 sites”. (Line 29)

Line 27 – genomic variation?

Yes. We have changed “variation” to “genomic variation”. (Line 32)

Line 32 – this final sentence of the abstract seems like a very broad conclusion, is there any way to specify? Maybe relate to pest management or arthropods? In my original review I suggested specifically explaining how the results pertain to management. Avoiding this in the abstract (as done in the revision) is fine, but then a clearer and stronger implication is needed.

As suggested, the sentence has been changed to: “This work improves our understanding of adaptive genomic variation along environmental gradients, and advances pest status forecasting by highlighting the genetic basis for local climate adaptation for this important agricultural pest.” (Lines 35-38)

Lines 66-68/ 252-254 – it strikes me that two reviewers highlighted this as not clearly justified in the manuscript. While the reviewer response better articulates the reasoning behind this, why not include some version of this in the manuscript?

As suggested, the relevant paragraph in the Methods has been changed to: “In the present study, to investigate the genetic variation associated with climate, we excluded samples from the regions that are only seasonally suitable for growth of DBM with an Ecoclimatic Index (EI = 0)¹⁶. This was done because in these regions populations are unlikely to receive perennially unpunctuated selection by local environmental variables and genetic variation cannot be continuously passed on to future generations over years. Specifically, regions that are only seasonally suitable for DBM growth and development (i.e., with an ecoclimatic index, EI = 0) are too harsh to allow survival in low temperature conditions during the winter. Annual recolonization of those regions from regions where DBM can overwinter (with an ecoclimatic index, EI > 0) has been biologically and genetically confirmed^{14,56-59}. No genetic differentiation was found among different geographical populations spanning from overwintering regions to seasonally inhabiting regions^{15,58,59}. If migration from one habitat overwhelms the other, migration from the source introduces new genetic variation that may prevent local adaptation^{60,61}. Accordingly, samples from regions that are only seasonally suitable for DBM (with EI=0 and the populations being subject to seasonally punctuated local selection by climatic factors) were excluded

from the analysis of correlation between genetic variation and climatic variables. The retained samples included 372 individuals from 78 sampling sites in the year-round persistence regions of DBM across the world. These samples were collected from different continents, with 13 samples from Africa, 29 from Asia, 5 from Europe, 13 from North America including Hawaii, 12 from South America, and 6 from Oceania (Supplementary Fig. 1 and Supplementary Table 1).” (Lines 252-271)

Line 195 – strictly speaking this isn’t what the results show, you’ve shown genomic associations with climate that suggest warming may elicit region-specific responses... but you haven’t presented “region-specific genomic responses of DBM to the changing climate”

As suggested, the sentence has been changed to: “This result shows genomic associations with climate that suggest climate change may elicit region-specific responses and indicates that DBM is capable of maintaining its pest status in most regions of year-round persistence past 2050 under the RCP8.5 climate change scenario.” (Lines 195-198)

Line 284 – which r package?

We used LEA package. We have added this information and reference “We used R package *LEA* to run the median z-scores of 5 runs and re-adjusting p-values with FDR correction⁶³.”. (Lines 297-299)

Reviewer #3 (Remarks to the Author):

I have reviewed a revised version of the manuscript “Large-scale genome-wide study reveals climate adaptive variability in a cosmopolitan pest”. Whereas I think the manuscript has improved considerably, I still think that it is not ready for publication.

I. My main concern revolves around an issue I described previously. For clarity, I have pasted my previous review and the authors’ response below.

Third, I’m unsure about the conceptual approach taken by combining predicted changes in habitat suitability with those in allele frequencies. Both predictions of habitat suitability and of the genomic makeup of populations in the future assume that current relationship between environment and allele frequencies or between environment and species occurrence remain constant over time. If, for instance, temperature changes in a given area are predicted to be severe, the allele frequencies of temperature-sensitive genes are also predicted to change dramatically. At the same time, habitat suitability may change drastically, but suitability and local adaptations are highly related. If habitat becomes less suitable, considerable allele frequency changes are also expected to be required. Thus, simply multiplying the two responses to come up with a new ‘eco-genetic’ index appears to be cutting a lot of corners, and does not improve our understanding of the vulnerability or

persistence of the species in the future. Identifying areas where conditions in the future are not mirrored in any of the sites where the species was sampled may provide a more realistic idea of where populations are at risk.

Authors' response: The EI-based habitat suitability of DBM proposed by Zalucki and Furlong (2011) is based on DBM's physiological traits and represents the population growth potential. Without taking inter-population genetic variation into account, there is no clue on how populations differ in their capacity to adapt to climate change. In this study, we develop a new habitat suitability index (EGI), serving as a calibration of the EI-based habitat suitability based on the integration of species' physiological traits and inter-population adaptive variation. We firstly used the genetic offset (GO) method to examine spatial regions where populations are challenged by the disruption of gene-environment relationships between current and projected future climates. We then followed the idea presented in Zalucki and Furlong (2011), where habitat suitability is represented by multiplying population growth index and stress index, to develop a new habitat suitability index (EGI) by taking population-specific genetic vulnerability (GO) into account as a "stress index". The combination of such a "stress index" (GO) and eco-climate index (i.e., DEI in this study) therefore represents intraspecific variation of responses to future climate, by integrating both genetic and physiological factors.

I think there is a misconception of the EI and how it is calculated in the paper by Zalucki and Furlong (2011) - or I don't quite get what the authors mean by saying that they are taking GO as a stress index... EI is described as the growth index multiplied by a stress index SI, where SI is defined as heat/drought stress. Hence, SI is *already* included in EI. By first including temperature and precipitation as stress factors in EI, then also using them in calculating the GO, and subsequently multiplying EI by GO, the climate factors are included twice in the EGI. From a biological mechanistic point of view, habitat is suitable because individuals are (sufficiently) well adapted to its associated conditions. Thus, the frequencies of relevant adaptive alleles dictate whether a population will be able to persist, and this is what the authors have already modeled by computing GO. To get an even better sense of vulnerability, it would be more useful to map current and future allele frequencies, and assess vulnerability as a function of the presence in area A of alleles predicted to be necessary for adaptations under future conditions in conjunction with estimates of functional connectivity with regions B/C/D, where current conditions are more similar to future conditions in area A. In addition, are there any regions where future conditions do not match current conditions observed anywhere within the current distribution of DBM? These are areas where our confidence in the predictions are very low – we basically don't know how DBM will respond to completely novel environmental conditions.

Thank you for your insightful comments. To address your concern that "By first including temperature and precipitation as stress factors in EI, then also using them in

calculating the GO, and subsequently multiplying EI by GO, the climate factors are included twice in the EGI”, we invited two mathematicians with expertise in ecological modelling (Jian Lin and Shiguo Huang) from the College of Computer and Information Sciences at our university to join our team as coauthors of the manuscript and improve the algorithm of EGI. The revised algorithm for calculating EGI has been provided in the Methods as follows:

(Lines 396-434)

“To examine intraspecific variation in genetic and physiological responses to future climate in 2050 under RCP8.5 scenario based on NorESM1-M GCM, we developed a new metric, the “eco-genetic index” (EGI), which combines the predictions based on genetic offset (GO) with difference in ecoclimatic index (DEI). Here, regions of decreasing EI (DEI < 0) were considered exclusively. Each gridded point in ArcGIS was described as a vector: $P_i = \{dei_i, go_i\}$, where $i = 1, 2, \dots, n$, and n is the number of the gridded points. Let the EGI and GO of each point be egi_i and go_i , and they are calculated as follows:

Step 1: Calculate the absolute values of dei_i and go_i and normalize^{70,71} them into [0.1, 0.9].

Step 2: Combine the normalized dei_i and go_i with the weighted geometric averaging (WGA) operator⁷⁴ as:

$$egi_i = dei_i^\alpha * go_i^{1-\alpha} \quad (1)$$

where $\alpha \in [0,1]$ is the weight of normalized dei_i , $i = 1, 2, \dots, n$.

Step 3: The optimal value of α can be determined with the following steps:

Step 3.1: Take the natural logarithm of Equation (1):

$$\ln egi_i = \alpha * \ln dei_i + (1 - \alpha) \ln go_i, \quad i = 1, 2, \dots, n \quad (2)$$

Obviously, $\ln dei_i < 0$ and $\ln go_i < 0$.

Step 3.2: Assuming $a, b > 0$, there exists a theorem such that:

$$\frac{a}{b} + \frac{b}{a} - 2 = \frac{(a-b)^2}{ab} \geq 0 \quad (3)$$

The equality in Equation (3) holds if and only if $a = b$.

To balance the indices of dei_i and go_i , we minimized the total deviation between egi_i and dei_i, go_i . According to Equation (3) and Zhou et al.⁷⁵, the deviations d_i and t_i are formulated as follows:

$$d_i = \frac{-\ln egi_i}{-\ln dei_i} + \frac{-\ln dei_i}{-\ln egi_i} - 2 \quad (4)$$

$$t_i = \frac{-\ln egi_i}{-\ln go_i} + \frac{-\ln go_i}{-\ln egi_i} - 2 \quad (5)$$

Hence, the minimized model (M - 1) can be expressed as:

$$\begin{aligned} (M - 1): \min Y &= \sum_{i=1}^n (d_i + t_i) \\ &= \sum_{i=1}^n \left[\left(\frac{-\ln egi_i}{-\ln dei_i} + \frac{-\ln dei_i}{-\ln egi_i} - 2 \right) + \left(\frac{-\ln egi_i}{-\ln go_i} + \frac{-\ln go_i}{-\ln egi_i} - 2 \right) \right] \end{aligned} \quad (6)$$

Based on Equation (3), the minimized model (M - 1) can be equivalently written as:

$$(M - 2): \min Y = \sum_{i=1}^n \left[\frac{\alpha * \ln dei_i + (1-\alpha) * \ln go_i}{\ln dei_i} + \frac{\ln dei_i}{\alpha * \ln dei_i + (1-\alpha) * \ln go_i} \right]$$

$$+ \frac{\alpha * \ln dei_i + (1-\alpha) * \ln go_i}{\ln go_i} + \frac{\ln go_i}{\alpha * \ln dei_i + (1-\alpha) * \ln go_i} - 4] \quad (7)$$

Step 3.3: The Artificial bee colony (ABC) algorithm⁷⁶ is utilized to solve model (M - 2), which is a fractional programming problem. Y is the objective function, α is the independent variable of the function. The parameters are set as: population size = 20, number of iterations = 30, and $limit = 5$. The convergence curve of the ABC algorithm for solving the above model is plotted in Supplementary Figure 5. The algorithm converged at the fifth iteration. The converged, optimal α equals 0.6049 (Supplementary Figure 6). Accordingly, the eco-genetic index can be calculated by $egi_i = dei_i^{0.6049} * go_i^{0.3951}$. The resulting EGI values were then mapped with ArcGIS 10.2 to show the geographical distribution of EGI in the EI-decreased regions under the projected future climate scenario (Fig. 2c).”

We appreciate your kind suggestion on mapping current and future allele frequencies, and assessing vulnerability as a function of the presence in area A of alleles predicted to be necessary for adaptations under future conditions in conjunction with estimates of functional connectivity with regions B/C/D, but in this paper, we attempted to directly and quantitatively estimate both physiological and genetic variations (rather than just about genetic effects) necessary for climate adaptation and predict the habitat suitability of DBM.

II. Another, smaller, concern is that there is much emphasis on the role of climate in shaping genetic variation. However, these are the only variables tested for associations. This is fine, but the only thing that can be concluded is that environment (and not specifically climate) is likely driving adaptive genetic variation in DBM. Indeed, almost 60% of the variation was unaccounted for in GDM models.

We totally agree that environment is likely driving adaptive genetic variation, but we do not have enough evidence to identify the exact role of climate in shaping genetic variation. In this manuscript, therefore, what we have done is to visualize the climate-associated genetic variation and analyze the relationship between genomic variation and climate variables in DBM.

As suggested, a few sentences in the text have been revised as follows:

1) in Abstract:

“Here, we analyze genomic data from *Plutella xylostella* collected from 78 sites spanning six continents to reveal that climate-associated adaptive variation exhibits a roughly latitudinal pattern.” (Lines 28-30)

“This work advances our understanding of adaptive genomic variation along environmental gradients, and advances pest status forecasting by highlighting the genetic basis associated with local climate adaptation for this important agricultural pest.” (Lines 35-38)

2) in the 2nd paragraph of the main text:

“However, little is known about the extent to which adaptive genetic variation is driven by climate in arthropod species, and how populations within species differ in their capacity to adapt to climate change. Here, by combining a newly available

genomic resource with climate data, we analyze the relationship between genomic variation and climate variables in the diamondback moth (DBM), *Plutella xylostella*.” (Lines 52-56)

3) in the 2nd paragraph of the Results:

“We used generalized dissimilarity modelling (GDM)²⁰ to examine climate-mediated genomic variation among different populations based on 517 SNPs selected among the 3,648 SNPs. Of the 12 environmental variables that had a pairwise Pearson’s *r* less than 0.8, the top four included one precipitation-related (bio18) and three temperature-related (bio03, bio09 and bio08) variables (Fig. 1b and Supplementary Table 6). Environment-associated genetic variation exhibited a roughly latitudinal pattern, regardless of geographical region or continent, suggesting that DBM populations from the same latitude exhibit comparable genomic variation (Fig. 1c).” (Lines 82-88)

III. Much more information is needed on how the CLIMEX models were generated. What were the settings? What input data did you use?

We have provided information about CLIMEX models in the Method as follows:

“The CLIMEX model, which has been shown to be effective for examination of species distribution under future climate scenarios⁷², was used to predict the habitat suitability for DBM in 2050 under RCP8.5 scenario, the only one that is available in CliMond (<https://www.climond.org/>). The CLIMEX model for DBM in Zalucki and Furlong¹⁶ was developed based on temperature, moisture and stress index. Temperature index includes limiting minimum (DV0) and maximum temperature (DV3), lower (DV1) and upper (DV2) optimal temperature. Moisture index includes minimum (SM0) and maximum (SM3) tolerable soil moisture, lower (SM1) and upper (SM2) optimal soil moisture. Stress index includes cold stress, heat stress, dry stress, wet stress and hot-wet stress. The setting of these parameters for prediction of habitat suitability in our study were from the CLIMEX model for DBM in Zalucki and Furlong¹⁶. We altered one of the parameters in CLIMEX¹⁶, changing the hot-wet stress temperature threshold from 30°C to 32°C based on a study on the relationship between temperature and developmental rate showing that DBM can survive and develop at temperatures < 32°C⁷³ (Supplementary Table 9).” (Lines 373-386)

Minor comments:

l. 46 “accumulate” – what do you mean? Are you referring to new mutations, or based on standing variation?

We are referring to both new mutations and standing variation, and now make this clear in the revised manuscript: “Insects with high fecundity and short generation time can rapidly accumulate adaptive alleles through new mutations and standing variation.....” (Lines 50-51).

l. 52 “important factors...” – not sure what you mean. What “factors”? Don’t you

mean to say that you want to identify the environmental variables that are associated with SNP variation (or rather, the other way around)?

We have revised the sentence accordingly. It now reads: “Here, by combining a newly available genomic resource with climate data, we analyze the relationship between genomic variation and climate variables in the diamondback moth (DBM), *Plutella xylostella*.” (Lines 54-56)

l. 69 “357 individuals ... from 75 different sites” – what are the correct numbers? On line 254 you mention 372 individuals from 78 sites.

Once we excluded samples from the regions that are only seasonally suitable for growth of DBM, the retained samples included 372 individuals from 78 sampling sites. After SNP quality filtering, 357 individuals from 75 sites were eventually retained for further analysis.

We have revised the paragraph in the Methods to make our statement clearer. It now reads: “The retained 372 individuals shared a subset of 34,969,375 SNPs, which accounted for 87.19% of the total SNPs (40,107,925) and represented most of the genomic variation among 532 individuals (Supplementary Table 2)¹⁵. Using VCFtools v.0.1.6, we excluded the SNPs with minor allele frequency (MAF) < 5% and missing rate > 10%. We then sampled data by examining single SNPs in small, 25 bp DNA window to focus on loci independent of linkage disequilibrium. After quality filtering, a total of 200,055 bi-allelic SNPs across 357 DBM individuals collected from 75 different sites worldwide were eventually retained for further analysis (Supplementary Fig. 1, Supplementary Tables 1 and 2).” (Lines 272-279)

l. 80 “(none cross-correlated)” – this is incorrect; any variables with cross-correlations < 0.8 were retained.

We have changed this sentence to “Of the 12 environmental variables that had a pairwise Pearson’s *r* less than 0.8,....” (Line 84)

l. 90 “new metric” – please drop ‘new’, because it isn’t. This has been done many times, including by Fitzpatrick and Keller, the paper you are referring to.

As suggested, the sentence has been changed to “...., we defined a metric of genetic vulnerability that we called “genetic offset”.” (Line 91)

l. 250 “with climate change” – drop ‘change’, since you are looking at associations with climate variables, not with change.

As suggested, the sentence has been changed to “In the present study, to investigate the genetic variation associated with climate,....” (Line 252)

l. 267-270 Please explain that you downselected variables using Pearson cross-correlation coefficients. Please also justify why you retained one variable over another in a pair of correlated ones.

Pearson correlation coefficients (Pearson’s *r*) are usually used to indicate how collinear two variables are (eg. Bay et al., 2018; Jia et al., 2020). High absolute correlation coefficients usually indicated high linear relatedness (Dormann et al.,

2013).

We have introduced the procedure for variable screening in the Methods as follows: “To down-select the bio variables, we preferentially discarded the variables with multiple related variables (Supplementary Table 8). We then ran the GDM model using 3,648 SNPs and the variables with $| \text{Pearson } r |$ less than 0.8 (bio02, bio03, bio07, bio08, bio09, bio12, bio15, bio18 and bio19), one of bio01 and bio11 ($| r | = 0.92$), one of bio5 and bio 10 ($| r | = 0.94$), as well as one of bio14 and bio17 ($| r | = 1.0$). The subset of 12 climate variables with highest value of explained deviance (28.17%) in the GDM model were retained, including bio01, bio02, bio03, bio07, bio08, bio09, bio10, bio12, bio14, bio15, bio18, bio19, for further analysis.” (Lines 321-327)

Reviewer #4 (Remarks to the Author):

I thank the authors for their revised version of the manuscript. The updated manuscript contains a large proportion of additional information which resolves a lot of aspects that were unclear in the previous version.

Unfortunately, the authors seem to have decided to not include additional data or repeat analyses to address my previous comments (but also comments by reviewer 1). In some cases they simply refer to other articles but simply because previous studies have used certain approaches does not mean they chose the most reliable option.

In my previous review, I noted that the three different methods used to identify candidate loci were used with different stringency when selecting the outliers. Especially bayenv2 was only run once and with a relatively low number of MCMC iterations. The authors have now corrected the number of iterations but they still only use a single run (as opposed to five for LFMM). It is well known that the results of such methods can be very noisy in single runs and therefore multiple runs are suggested to obtain reliable results (see e.g. Blair et al 2014: On the stability of the Bayenv method in assessing human SNP-environment associations).

As suggested, we have reanalyzed the data using Bayenv2 with five independent runs. Each run used a different random seed but the same input data set and 100,000 MCMC iterations. Figure 1, Figure 2, Supplementary Figure 2, Supplementary Figure 3, Supplementary Table 5, and Supplementary Table 7 have been updated accordingly.

The following information has also been added to the Methods: “Five independent runs of the Bayenv program were performed with different random seeds⁶⁴. The results were presented as Bayes factors (BFs). A averaged $\log_{10}(\text{BF})$ value of five runs > 1.5 is considered as high support for a model where environmental parameters have significant effects on allele frequencies⁶⁵.” (Lines 307-310)

In another comment, I asked whether there are differences in the survival rates between the different strains even without temperature treatment or stress. The authors response was only that the WT was used as control. I may have missed this but it should be made clear that the mutant shows reduced survival rates only under extreme temperatures.

We have provided additional information about the survival rate between different strains (WT vs. MU) even without temperature treatment/stress. In the Results, it now reads: We then examined the difference in survival rate between the wild-type (WT) and *PxCad*-deficient mutant (MU) strains under both favorable temperature (26°C) and several extreme temperatures. Comparing to the WT, the survival of the MU was not affected under the favorable temperature and the survival rate only declines when exposing to extreme temperatures.

Reviewer 1 correctly highlighted that it would be more valuable to test gene expression directly after treatment while it was measured after recovery in this case. That way, we cannot be sure that the effects seen in this case are a direct consequence of the treatment.

We did perform two experimental settings to test gene expression, i.e. gene expression directly after treatment vs. gene expression after recovery. We have provided the rationale with respect to such an experimental design in our previous response letter as follows (responses to reviewer 1):

“In the present study, we designed our experiment to examine cadherin gene expression after a series of temperature treatments followed by periods of recovery by referring to a previous study (Zhang and Denlinger, 2010). In that study, the authors examined the expression patterns of heat shock protein transcripts, *hsp90*, *hsp70*, *hsc70*, isolated from the corn earworm during thermal stress and pupal diapause in the heat (40°C) and chilling (0°C, 4°C, 8°C) conditions, with a recovery at 25°C for each of the temperature treatments.

In our experiment, a recovery at 26°C was arranged for each of the heat/chilling stress treatments (43°C, -14°C, -17°C and -20°C) in order to examine whether or not gene expression could return to the original level when moths were transferred to a favorable temperature at 26°C after heat/chilling stress treatments. We did not arrange a recovery for the treatment at 40°C because that is a tolerable temperature for DBM to live within a duration of 2 hours (Nguyen et al., 2014). In total, therefore, we set nine temperature treatments and one control. The nine temperature treatments included five different temperature treatments (43°C, 40°C, -14°C, -17°C and -20°C), and four temperature treatments (43°C, -14°C, -17°C and -20°C) followed separately by a recovery for 24 hours. After each treatment, moths were immediately frozen in liquid nitrogen so that we could compare the difference in gene expression between each of the temperature treatments and the control.”

I second reviewer 1 in their impression that the manuscript seems to be relatively narrow and quickly zeroes in on *PxCad* as candidate gene. There is not much discussion on alternative explanations for the results or potentially confounding factors.

We have provided the following explanations for zeroing in on *PxCad* in our previous response letter (responses to reviewer 1):

“We zeroed in on *PxCad* because: 1) it had the highest number of SNPs (9 SNPs, 7 of which are located in coding region), identified to be putatively associated with

temperature adaptation according to the prediction based on three models of Samβada v0.5.3, latent factor mixed models (LFMM), and Bayenv 2 (Table S7); and 2). *PxCad* is annotated as a cadherin-like protein (Guo et al., 2015; Park et al., 2015). Classical cadherins are a superfamily of transmembrane proteins involved in regulating cell-cell adhesion, signal transduction and tissue morphogenesis (Bulgakova, Klapholz, & Brown, 2012; Wu & Maniatis, 1999). For example, in mammals, epithelial cadherin (E-cadherin) is involved in morphogenesis (Battle et al., 2000; Behrens, Löwrick, Klein-Hitpass, & Birchmeier, 1991). Cadherin genes have also been reported as responders for cold stress in *Ruditapes philippinarum* (Menike et al., 2014). Pigott & Ellar (2007) have also demonstrated the roles of cadherin-like proteins in maintaining structural integrity of midgut epithelial organization. All of the above support the potential of *PxCad* as a temperature-sensitive responder to climate change.”

We have also revised the main text to make our statement clearer. In the Results, it now reads: “With the highest number of SNPs (4 out of 94) identified in the core dataset (94 putatively adaptive SNPs), *PxCad* was the most likely candidate to be involved in regulating DBM’s responses to climate change. Homologs of *PxCad* encode cadherin-like proteins, which are known to be involved in cell adhesion²⁸, sensory perception of light²⁹, vocal and locomotory behavior³⁰, has been evidenced to be responders for thermal stress³¹.” (Lines 146-150)

Finally, I would like to ask the authors to include a lot of the additional information and references from the response letter into the main manuscript.

We have included the following information from the response letter into the main manuscript:

In the Methods:

“In the present study, to investigate the genetic variation associated with climate, we excluded samples from the regions that are only seasonally suitable for growth of DBM with an Ecoclimatic Index (EI = 0)¹⁶. This was done because in these regions populations are unlikely to receive perennially unpunctuated selection by local environmental variables and genetic variation cannot be continuously passed on to future generations over years. Specifically, regions that are only seasonally suitable for DBM growth and development (i.e., with an ecoclimatic index, EI = 0) are too harsh to allow survival in low temperature conditions during the winter. Annual recolonization of those regions from regions where DBM can overwinter (with an ecoclimatic index, EI > 0) has been biologically and genetically confirmed^{14,56-59}. No genetic differentiation was found among different geographical populations spanning from overwintering regions to seasonally inhabiting regions^{15,58,59}. If migration from one habitat overwhelms the other, migration from the source introduces new genetic variation that may prevent local adaptation^{60,61}. Accordingly, samples from regions that are only seasonally suitable for DBM (with EI=0 and the populations being subject to seasonally punctuated local selection by climatic factors) were excluded from the analysis of correlation between genetic variation and climatic variables. The retained samples included 372 individuals from 78 sampling sites in the year-round

persistence regions of DBM across the world. These samples were collected from different continents, with 13 samples from Africa, 29 from Asia, 5 from Europe, 13 from North America including Hawaii, 12 from South America, and 6 from Oceania (Supplementary Fig. 1 and Supplementary Table 1).

The retained 372 individuals shared a subset of 34,969,375 SNPs, which accounted for 87.19% of the total SNPs (40,107,925) and represented most of the genomic variation among 532 individuals (Supplementary Table 2)¹⁵. Using VCFtools v.0.1.6, we excluded the SNPs with minor allele frequency (MAF) < 5% and missing rate > 10%. We then sampled data by examining single SNPs in small, 25 bp DNA window to focus on loci independent of linkage disequilibrium. After quality filtering, a total of 200,055 bi-allelic SNPs across 357 DBM individuals collected from 75 different sites worldwide was retained for further analysis (Supplementary Fig. 1, Supplementary Tables 1 and 2)” (Lines 252-279)

“To down-select the bio variables, we preferentially discarded the variables with multiple related variables (Supplementary Table 8). We then ran the GDM model using 3,648 SNPs and the variables with | Pearson r | less than 0.8 (bio02, bio03, bio07, bio08, bio09, bio12, bio15, bio18 and bio19), one of bio01 and bio11 (| r | = 0.92), one of bio5 and bio 10 (| r | = 0.94), as well as one of bio14 and bio17 (| r | = 1.0). The subset of 12 climate variables with highest value of explained deviance (28.17%) in the GDM model were retained, including bio01, bio02, bio03, bio07, bio08, bio09, bio10, bio12, bio14, bio15, bio18, bio19, for further analysis.” (Lines 321-327)

“The CLIMEX model, which has been shown to be effective for examination of species distribution under future climate scenarios⁷², was used to predict the habitat suitability for DBM in 2050 under RCP8.5 scenario, the only one that is available in CliMond (<https://www.climond.org/>). The CLIMEX model for DBM in Zalucki and Furlong¹⁶ was developed based on temperature, moisture and stress index. Temperature index includes limiting minimum (DV0) and maximum temperature (DV3), lower (DV1) and upper (DV2) optimal temperature. Moisture index includes minimum (SM0) and maximum (SM3) tolerable soil moisture, lower (SM1) and upper (SM2) optimal soil moisture. Stress index includes cold stress, heat stress, dry stress, wet stress and hot-wet stress. The setting of these parameters for prediction of habitat suitability in our study were from the CLIMEX model for DBM in Zalucki and Furlong¹⁶. We altered one of the parameters in CLIMEX¹⁶, changing the hot-wet stress temperature threshold from 30°C to 32°C based on a study on the relationship between temperature and developmental rate showing that DBM can survive and develop at temperatures < 32°C⁷³ (Supplementary Table 9).” (Lines 373-386)

Reference:

- Batlle, E. et al. The transcription factor snail is a repressor of E-cadherin gene expression in epithelial tumour cells. *Nat Cell Biol.* 2, 84-89 (2000).
- Bay, R. A. et al. Genomic signals of selection predict climate-driven population

- declines in a migratory bird. *Science* 359, 83-86. (2018)
- Behrens, J., Löwrick, O., Klein-Hitpass, L. & Birchmeier, W. The E-cadherin promoter: functional analysis of a GC-rich region and an epithelial cell-specific palindromic regulatory element. *Proc Natl Acad Sci U S A.* 88, 11495-11499 (1991).
- Bulgakova, N. A., Klapholz, B. & Brown, N. H. Cell adhesion in *Drosophila*: versatility of cadherin and integrin complexes during development. *Curr Opin Cell Biol.* 24, 702-712 (2012).
- Dormann et al. Collinearity: A review of methods to deal with it and a simulation study evaluating their performance. *Ecography*, 36(1): 27-46. (2013)
- Guo, Z. et al. The midgut cadherin-like gene is not associated with resistance to *Bacillus thuringiensis* toxin Cry1Ac in *Plutella xylostella* (L.). *J Invertebr Pathol.* 126, 21-30 (2015).
- Jia K et al. Landscape genomics predicts climate change-related genetic offset for the widespread *Platycladus orientalis* (Cupressaceae). *Evolutionary Applications*, 13(4):665-676. (2020)
- Nguyen C, Bahar MH, Baker G, Andrew NR. Thermal tolerance limits of diamondback moth in ramping and plunging assays. *PLoS One*, 9, e87535. (2014)
- Park, Y., Herrero, S. & Kim, Y. A single type of cadherin is involved in *Bacillus thuringiensis* toxicity in *Plutella xylostella*. *Insect Mol Biol.* 24, 624-633 (2015).
- Pigott, C. R. & Ellar, D. J. Role of receptors in *Bacillus thuringiensis* crystal toxin activity. *Microbiol. Mol. Biol. Rev.* 71, 255-281 (2007).
- Wu, Q. & Maniatis, T. A striking organization of a large family of human neural cadherin-like cell adhesion genes. *Cell* 97, 779-790 (1999).
- Zhang Q, Denlinger DL. Molecular characterization of heat shock protein 90, 70 and 70 cognate cDNAs and their expression patterns during thermal stress and pupal diapause in the corn earworm. *Journal of Insect Physiology*, 56(2): 138-150. (2010)

Reviewer comments, response- -

Reviewer #2 (Remarks to the Author):

The authors have adequately addressed my remaining concerns. The revisions have in my estimation significantly improved the quality of the manuscript.

Reviewer #3 (Remarks to the Author):

Thank you for responding in detail to my comments. I'm happy with most, but I'm afraid I keep on getting back to the approach where you combine GO and DEI. It's great that you're using a weighting scheme and are able to solve for the optimal value of alpha. However, I still fail to see how the two indices are not conveying the same information. I agree that they may only partly overlap -and this is I guess where your weighting scheme may be helpful- but potential non-overlapping information seems to be extremely generalized. I hope you can enlighten me..

The issue being that in EI, habitat suitability is highly generalized, whereas in GO you capture site-specific genetic responses to local habitat conditions. What is "stress" for one population, may be the norm for another that is genetically adapted to those conditions. Hence, your landscape genomics approach should capture those details in much more detail. What can EI add to that? The answer may lie in epigenetic responses, variation in plasticity etc. However, I'm not quite convinced yet that the broad physiological tolerances you used to model EI provide sufficient additional information to generalize across the globe. As far as I understand, these tolerance levels are known for some individuals -presumably from a particular location- that were subjected to stress tests in the lab. But these tolerance levels are very likely to vary considerably among populations - and to be associated to specific genes. Thus, we're back to the GO.

I'm wondering whether it wouldn't make (more) sense to focus on the genotype-environment associations under current and future conditions, and identify areas that become suitable for given genotypes in the future. I think you can be fairly confident about the predictions of such an approach. If you actually (would) have population-specific data on tolerance levels (ideally the 75 you obtained genomic data for), it would make your argument much stronger. In that case your genetic data plus some term that includes non-genetic responses should equal the observed physiological response..

In whatever way you decide to proceed, I think that at the very least a thorough discussion related to the above should be included - where can or can't you be confident, and where are you generalizing? How are DEI and GO complementary and to what extent are they overlapping?

Minor comments

1. Why only focus on regions with decreasing EI?
2. Caption Fig 2 - "bright colors" > I assume you mean warm/red colors
3. You mention that after filtering you go down from 78 to 75 sites. I would suggest to change that number in the abstract.
4. Line 30-33: I'm not sure that this is something that you are showing here; rather a general hypothesis.
5. In the results section you describe genetic offset as if it were a new term that you coin in this manuscript; however it's been used numerous times, so please rephrase.

Reviewer #5 (Remarks to the Author):

Chen et al. present a study of adaptive genomic variation along environmental gradients and genetic offset under future climate in diamondback moth. This is the first version of the MS and response to reviewers I get to review. Overall, I consider the manuscript novel, interesting and highly relevant both from a scientific perspective as well as for the management of this agricultural pest. My comments center mostly on the way the genetic offsets and the eco-genetic index are calculated as well as ambiguous formulations concerning the interpretation of genetic offsets.

Title: I am not sure how to interpret the term "climate adaptive variability". In case this term refers to "variability in adaptive potential", it is in my opinion incorrect, since genomic offsets do not represent adaptive potential (see also comments below). Maybe better use "climate-associated adaptive genetic variation"?

L. 26-38: I was surprised to find that the novel eco-genetic index was not at all mentioned in the abstract

L. 90-91: The term "genetic offset" was previously defined by Fitzpatrick and Keller (Ecology Letters 2015). Not only should they be cited here, but their authorship of the term must be clearly acknowledged. I would therefore suggest the following wording:
"To investigate which DBM populations might be more vulnerable to future climate change, we applied a metric of genetic vulnerability called "genetic offset", originally developed Fitzpatrick and Keller (Ecology Letters 2015)."

L. 99-104: This section seems to confuse various lines of argumentation. Further up in L. 94-95 you state that "populations with higher genetic offset are more vulnerable to climate change and require greater adaptive potential (or genetic variation necessary for adaptation) to the changing environment." I agree with this argumentation. However, in L. 99 to 102 you suggest that low levels of genetic offset represent high adaptive potential, which is not true, they just indicate low vulnerability. I would suggest to reformulate L. 99 to L. 102 as follows: "Most DBM populations seem to experience low disruption of gene-climate associations under future climate. Taken together with our previous findings that high levels of genetic polymorphism and diversity among different populations enable DBM to adapt readily to different environments worldwide, we thus assume that they will likely remain damaging pests across most of their distribution range." Similarly, in L. 102 to 104, high levels of genetic offset do not indicate low adaptive potential, but higher vulnerability (i.e. high disruption of existing gene-environment relationships, which will require adaptation or gene flow to counteract), please reformulate accordingly. Also, in L. 101, it does not become entirely clear to me whether you refer to high genetic among population differentiation or high within population genetic diversity (in various populations across the range). Please formulate more precisely.

L. 105-106: Here, it is not clear to me how to interpretate the statement that local adaptation in the species involves "physiological and genetic responses". Usually, local adaptation refers to genetic differentiation among populations that leads to a fitness advantage of native over foreign genotypes in a given environment (e.g. Kawecki & Ebert, Ecology Letters 2004), whereas physiological response rather points to phenotypic plasticity. Or do you refer to local adaptation (i.e. genetically based among population differentiation) IN the physiological response? Please rephrase accordingly.

L. 119-122: Again, note that genetic offset does not represent adaptive potential! This interpretation of your results is thus not correct.

L. 121: Please replace "could be moderated by their genetically adaptive potential" with "could be moderate since they are predicted to experience only minor interruptions of gene-climate associations". Please note that GO does not tell anything about the adaptive potential of populations, but rather about the necessity to adapt.

L. 125: This interpretation seems far-fetched since the relationship between most of the genomic markers used for the offset calculation and physiological responses is unknown. Experiments would be required to clarify as to how far populations' physiological stress response and fitness depend on genetic offset.

L. 150: Please replace "has been evidenced" with "have been evidenced"

L. 193: I would suggest reformulating as "tied to variation in the physiological response to climate".

L.198: "majority" -> "the majority"

L. 366: Did you really calculate the genetic offsets as Euclidean distances between transformed current and future climates in a given grid cell? This approach is usually used for gradient forest based genomic offsets whereas the `gdm.predict` function of the GDM package in R predicts FSTs between the projected genomic compositions of populations under current and future climate based on the fitted GDM (i.e. the obtained values should be bounded between 0 and 1). The latter is preferable, because FST has a straightforward biological interpretation, whereas Euclidean distances between transformed climates have no clear biological meaning, they increase automatically with an increasing number of climatic variables involved (curse of dimensionality) and a statement about whether an offset is large or small (compared to other studies) is therefore not possible. Please clarify. I strongly advice re-calculating the offsets using the `gdm.predict` function if really Euclidean distances were used.

L. 395 ff: I share the concerns of Reviewer 3 about calculating the eco-genetic index based on a combination of genomic offset and difference in ecoclimatic index, whereby both the genomic offset and change in habitat suitability depend on climate. It does not become clear to me how the stress induced by climate change itself can be disentangled from stress of a lack of genomic adaptation to this change, since both will be correlated. At least a better, verbally formulated, explanation of what the index represents mechanistically, would be necessary.

RESPONSES TO REVIEWERS' COMMENTS

REVIEWER COMMENTS

Reviewer #3 (Remarks to the Author):

Thank you for responding in detail to my comments. I'm happy with most, but I'm afraid I keep on getting back to the approach where you combine GO and DEI. It's great that you're using a weighting scheme and are able to solve for the optimal value of alpha. However, I still fail to see how the two indices are not conveying the same information. I agree that they may only partly overlap -and this is I guess where your weighting scheme may be helpful- but potential non-overlapping information seems to be extremely generalized. I hope you can enlighten me.

The issue being that in EI, habitat suitability is highly generalized, whereas in GO you capture site-specific genetic responses to local habitat conditions. What is "stress" for one population, may be the norm for another that is genetically adapted to those conditions. Hence, your landscape genomics approach should capture those details in much more detail. What can EI add to that? The answer may lie in epigenetic responses, variation in plasticity etc. However, I'm not quite convinced yet that the broad physiological tolerances you used to model EI provide sufficient additional information to generalize across the globe. As far as I understand, these tolerance levels are known for some individuals -presumably from a particular location- that were subjected to stress tests in the lab. But these tolerance levels are very likely to vary considerably among populations – and to be associated to specific genes. Thus, we're back to the GO.

I'm wondering whether it wouldn't make (more) sense to focus on the genotype-environment associations under current and future conditions, and identify areas that become suitable for given genotypes in the future. I think you can be fairly confident about the predictions of such an approach. If you actually (would) have population-specific data on tolerance levels (ideally the 75 you obtained genomic data for), it would make your argument much stronger. In that case your genetic data plus some term that includes non-genetic responses should equal the observed physiological response..

In whatever way you decide to proceed, I think that at the very least a thorough discussion related to the above should be included – where can or can't you be confident, and where are you generalizing? How are DEI and GO complementary and to what extent are they overlapping?

Thank you for your insightful comments and the suggestion on a thorough discussion related to issue on combination of GO and DEI.

Despite the increasing pest status of DBM and affirmations that it is the most extensively distributed Lepidoptera species (Talekar and Shelton, 1993; Furlong et al., 2013), our knowledge of its relative population abundance is surprisingly limited. Zalucki and Furlong (2011) developed a CLIMEX-based algorithm that combined climate data with eco-physiological traits. The resulting ecoclimatic index (EI) values were used to predict the habitat suitability of DBM, showing its core distribution ($EI > 0$) where it persists year-round and the regions ($EI = 0$) where it can be a seasonal pest. We agree very well with you that the EI-based prediction of habitat suitability is highly generalized, only taking the physiological tolerance of DBM across the globe into account without reflecting evolutionary adaptation and variation of the tolerance levels among different populations that are likely associated with specific genes. We assume that the lack of consideration of evolutionary adaptation in EI model might be due to the absence of relevant genetic information on fitness traits, which is required for predicting species responses to climate change (Huey et al., 2012). When we generated the new dataset of genome-wide single nucleotide polymorphisms (SNPs) sequenced from a worldwide sample of different locations (sites) across a diverse range of biogeographical regions (You et al., 2020), we were interested in seeking a novel approach that allows us to incorporate our population-specific genomic data into EI for better predicting the habitat suitability of DBM subject to climate change. We were happy to find that the genetic offset (GO), which describes the genotype-environment relationships under current and future conditions (Fitzpatrick and Keller, 2015), should be a useful tool for us to directly estimate the genetic variation necessary for climate adaptation of DBM. With a mechanistic understanding of the assertion that “we are now reaching the stage at which specific genetic factors with known physiological effects can be tied directly and quantitatively to variation in phenology” (Wilczek et al., 2010), thus we combined the genetic offset and EI-based prediction of climatic habitat suitability, and developed the new metric, the “eco-genetic index” (EGI).

We appreciate your comments on the part overlap of EI and GO. To answer your question “to what extent they are overlapping?”, we calculated the Pearson Correlation Coefficient (Gillham, 2001) between DEI and GO to get a result of $R = 0.5317$ ($P < 0.001$) (Supplementary Fig. 5), which provides a numerical estimation of the overlapping. To solve the overlapping problem, as we replied to your previous comments, we used a linear transfer function of normalization to make values of dei_i and go_i being dimensionless, and then used the weighted geometric averaging (WGA) operator and the artificial bee colony (ABC) algorithm to optimize the value of alpha and improve the estimation of EGI. Although this new metric, EGI, is imperfect, it does allow us to link population genetic variation with physiological responses for predicting the adaptive vulnerability of DBM, particularly when the population-specific data on physiological tolerance levels are not available. Our results also indicate that DEI and GO are functionally complementary. Using the new metric (EGI), for example, we have identified that some DBM populations in regions of decreasing EI (such as in the central Africa and southern China) under future

climate can overcome the challenge of habitat suitability decline given lower levels of genetic vulnerability, suggesting their capacity to cope with climate change *in situ* (Fig. 2).

We are fairly sure, as you suggested, that it would make sense “to focus on the genotype-environment associations under current and future conditions, and identify areas that become suitable for given genotypes in the future”. However, as we have replied to your previous comments, in this paper, we attempted to directly and quantitatively estimate population-specific physiological and genetic variations necessary for climate adaptation and predict the habitat suitability of DBM under future conditions. We believe that the ecoclimatic index (EI) values based on the CLIMEX simulation are useful for predicting the habitat suitability of DBM, which enables us to determine the range of its distribution, the regions where it can persist year-round, and the regions where it is a seasonal pest. In this project, such knowledge helps us focus on the regions where DBM persists year-round with a positive ecoclimatic index ($EI > 0$) (Zalucki and Furlong, 2011), where populations are subject to seasonally uninterrupted local selection by climatic factors. Indeed, as you indicated, the broad tolerance levels we used to model EI cannot provide sufficient information to generalize the physiological limits across the globe. We understand if we had population-specific data on tolerance levels (to match our population-specific genomic data), and the data on phenotypic plasticity and epigenetic responses, we would be able to make our arguments much stronger or even develop a novel metric that incorporates more relevant information on the adaptive capacity of DBM into a model.

Given all this, we have revised several paragraphs as set out below.

(1) Results:

Original wording: “To understand how genetic variation interacts with ecophysiological effects to reflect population-level responses of species to future climate, we developed a new metric, the “eco-genetic index” (EGI), which combined the genetic vulnerability and EI-based prediction of climatic habitat suitability.”

Revised wording: “Local adaptation, resulting from environment-driven intraspecific genetic differentiation, is an important feature of species that inhabit spatially heterogeneous habitats²³. To better reflect the role of population-level genetic variation in DBM’s responses to climate change, we developed a new metric, the “eco-genetic index” (EGI), which combines the genetic offset⁴ and EI-based prediction of climatic habitat suitability.” (Lines 113 - 117)

2) Discussion:

New paragraph: “To understand the potential distribution and pest status of DBM, Zalucki and Furlong¹⁶ developed a CLIMEX-based algorithm to predict the ecoclimatic index (EI) for the habitat suitability of DBM using climate data and

eco-physiological traits. However, the EI-based prediction of habitat suitability is highly generalized without reflecting population-specific tolerance levels and genetic variations. In this paper, based on our recently-generated genomic data of a worldwide sample¹⁵, we developed a new metric, the eco-genetic index (EGI), which combines the genetic offset (GO)⁴ and DEI (difference in ecoclimatic index (DEI) between current and projected future climate scenarios). Although the EGI metric is imperfect, it illustrates how population genetic variation can be tied with physiological effects⁵¹ for predicting the adaptive capacity of DBM, particularly when the population-specific data on physiological tolerance levels are not available. Our results indicate that DEI and GO are functionally complementary. With the EI prediction, we are able to determine the regional distribution of DBM throughout the world, which helps us focus this project on the regions in which DBM persists year-round with a positive ecoclimatic index ($EI > 0$)¹⁶. Using the new metric (EGI), we have identified that some DBM populations in regions of decreasing EI (such as in the central Africa and southern China) under future climate can overcome the challenge of habitat suitability decline given lower levels of genetic vulnerability, suggesting their capacity of adaptive evolution *in situ* to track climate change (Fig. 2). Looking ahead, if population-specific data on tolerance levels (to match our population-specific genomic data), and data on phenotypic plasticity and epigenetic responses were available, we would be able to make more robust conclusions, potentially supported by the development of a still more sophisticated metric that incorporates relevant information on additional aspects of adaptive capacity.”(Lines 223 - 244)

3) Methods:

Rewritten *Prediction of eco-genetic adaptation* section: “The ecoclimatic index (EI) values based on CLIMEX simulations are highly generalized and do not reflect evolutionary adaptation and variation of tolerance levels among different populations, because of the absence of relevant information on fitness traits. Based on our recently-generated dataset of genome-wide single nucleotide polymorphisms (SNPs) sequenced from a worldwide DBM sample of different locations (sites) across a diverse range of biogeographical regions¹⁵, we developed a new metric, the “eco-genetic index” (EGI), which combines the predictions based on genetic offset (GO) with the difference in ecoclimatic index between current and projected future climate scenarios (DEI). EGI allows us to incorporate our population-specific genomic data into EI for better predicting the habitat suitability of DBM subject to climate in 2050 under RCP8.5 scenario based on NorESM1-M GCM. Because DEI and GO, are correlated (Pearson’s⁷⁶ $R = 0.53$, $P < 0.001$; Supplementary Fig. 5), we used a linear normalization transfer function to make values of dei_i and go_i dimensionless, and then used the weighted geometric averaging (WGA) operator and the artificial bee colony (ABC) algorithm to optimize the value of alpha and improve the algorithm of EGI. Here, only regions of decreasing EI ($DEI < 0$) were considered because in these regions, DBM populations will be challenged by habitat suitability decline under the RCP8.5 scenario for 2050. Each gridded point in ArcGIS was

described as a vector: $P_i = \{dei_i, go_i\}$, where $i = 1, 2, \dots, n$, and n is the number of the gridded points. Let the EGI and GO of each point be egi_i and go_i , and they are calculated as follows:

Step 1: Calculate the absolute values of dei_i and go_i and normalize^{72,73} them into [0.1, 0.9].

Step 2: (Please see the details in the text of our revised manuscript)
(Lines 413 - 468)

Minor comments

1. Why only focus on regions with decreasing EI?

As previously mentioned, EI is an index showing the habitat suitability of DBM. We assume that regions with increasing EI will remain hospitable to DBM while regions with decreasing EI will be challenged by habitat suitability decline under the RCP8.5 scenario for 2050. Thus, our analysis focused on regions with decreasing EI where DBM populations would harness their adaptive potential (genetic variation necessary for adaptation, Fitzpatrick and Edelsparre (2018) to cope with future climate change.

In the text of our manuscript, the sentence in Results “Our analysis focused on regions of decreasing EI where adaptive potential would be required for habitats to remain hospitable to DBM.” has been changed to “We assume that regions with increasing EI will remain hospitable to DBM while regions with decreasing EI will be challenged by habitat suitability decline under the RCP8.5 scenario for 2050. Thus, our analysis focused on regions with decreasing EI.” (Lines 117 - 120)

The sentence in Methods “Here, regions of decreasing EI ($DEI < 0$) were considered exclusively.” has been changed to “Here, only regions of decreasing EI ($DEI < 0$) were considered because in these regions, DBM populations will be challenged by habitat suitability decline under the RCP8.5 scenario for 2050.” (Lines 426 - 428)

2. Caption Fig 2 – “bright colors” > I assume you mean warm/red colors

As suggested, “bright colors” in caption of Fig. 2 have been changed to “warm colors”. (Line 826)

3. You mention that after filtering you go down from 78 to 75 sites. I would suggest to change that number in the abstract.

As suggested, the number in the abstract has been changed to 75. (Line 31)

4. Line 30-33: I’m not sure that this is something that you are showing here; rather a general hypothesis.

Thanks for your comments. In the abstract, we have changed the sentence of “Populations with higher genetic offset are more vulnerable to changing environments and require greater variability to adapt, highlighting the fundamental role of genomic variation in climate change responses.” to “By defining a new eco-genetic index (EGI

that combines genetic variation and physiological responses, we predict that most DBM populations have high tolerance to projected future climates.” (Lines 32 - 34)

5. In the results section you describe genetic offset as if it were a new term that you coin in this manuscript; however it’s been used numerous times, so please rephrase.

As suggested by Review #5, the statement of “To investigate which DBM populations might be more vulnerable to future climate change, we defined a metric of genetic vulnerability that we called ‘genetic offset’. It represents the mismatch between current and expected future genetic variation based on genotype-environment relationships modelled by GDM analysis²¹ across contemporary populations.” has been changed to “We applied a metric of genetic vulnerability called “genetic offset”, originally developed by Fitzpatrick and Keller⁴, to investigate which DBM populations might be more vulnerable to future climate change.” (Lines 90 - 92)

Reviewer #5 (Remarks to the Author):

Chen et al. present a study of adaptive genomic variation along environmental gradients and genetic offset under future climate in diamondback moth. This is the first version of the MS and response to reviewers I get to review. Overall, I consider the manuscript novel, interesting and highly relevant both from a scientific perspective as well as for the management of this agricultural pest. My comments center mostly on the way the genetic offsets and the eco-genetic index are calculated as well as ambiguous formulations concerning the interpretation of genetic offsets.

Title: I am not sure how to interpret the term “climate adaptive variability”. In case this term refers to “variability in adaptive potential”, it is in my opinion incorrect, since genomic offsets do not represent adaptive potential (see also comments below). Maybe better use “climate-associated adaptive genetic variation”?

Thanks for your comment on the title of our manuscript. We used the term “climate adaptive variability” to cover three components (or subtitles) of the Results, and characterize the dynamics of variation, rather than the points of variation, involved in the evolutionary adaptation of DBM to climate change.

L. 26-38: I was surprised to find that the novel eco-genetic index was not at all mentioned in the abstract

In the abstract, we have now provided the information of eco-genetic index with a sentence of “By defining a new eco-genetic index (EGI) that combines genetic variation and physiological responses, we predict that most DBM populations have high tolerance to projected future climates.” (Lines 32 - 34). This replaces the sentence “Populations with higher genetic offset are more vulnerable to changing environments and require greater variability to adapt, highlighting the fundamental role of genomic variation in climate change responses.”

L. 90-91: The term "genetic offset" was previously defined by Fitzpatrick and Keller (Ecology Letters 2015). Not only should they be cited here, but their authorship of the term must be clearly acknowledged. I would therefore suggest the following wording: "To investigate which DBM populations might be more vulnerable to future climate change, we applied a metric of genetic vulnerability called "genetic offset", originally developed Fitzpatrick and Keller (Ecology Letters 2015)."

As suggested, the sentence has been changed to "We applied a metric of genetic vulnerability called "genetic offset", originally developed by Fitzpatrick and Keller⁴, to investigate which DBM populations might be more vulnerable to future climate change." (Lines 90 - 92). The authorship of the term, genetic offset, has been acknowledged in Methods as well.

L. 99-104: This section seems to confuse various lines of argumentation. Further up in L. 94-95 you state that "populations with higher genetic offset are more vulnerable to climate change and require greater adaptive potential (or genetic variation necessary for adaptation) to the changing environment." I agree with this argumentation. However, in L. 99 to 102 you suggest that low levels of genetic offset represent high adaptive potential, which is not true, they just indicate low vulnerability. I would suggest to reformulate L. 99 to L. 102 as follows: "Most DBM populations seem to experience low disruption of gene-climate associations under future climate. Taken together with our previous findings that high levels of genetic polymorphism and diversity among different populations enable DBM to adapt readily to different environments worldwide, we thus assume that they will likely remain damaging pests across most of their distribution range."

Similarly, in L. 102 to 104, high levels of genetic offset do not indicate low adaptive potential, but higher vulnerability (i.e. high disruption of existing gene-environment relationships, which will require adaptation or gene flow to counteract), please reformulate accordingly. Also, in L. 101, it does not become entirely clear to me whether you refer to high genetic among population differentiation or high within population genetic diversity (in various populations across the range). Please formulate more precisely.

Thank you for your comments. As suggested, the first paragraph under the subtitle "Genetic vulnerability and adaptive potential" in the text of our manuscript has been reformulated as follows: "We applied a metric of genetic vulnerability called "genetic offset", originally developed by Fitzpatrick and Keller⁴, to investigate which DBM populations might be more vulnerable to future climate change. Populations with higher genetic offset are more vulnerable to climate change and require greater adaptive potential (or genetic variation necessary for adaptation) to the changing environment⁴. Under greenhouse gas emission scenarios RCP8.5 for 2050, the genetic offset was low for most populations (Fig. 2a). The comparison of genetic offset under different scenarios (RCP2.6, RCP4.5, RCP6.0 and RCP8.5) showed an increasing trend with rising greenhouse gas emissions (Supplementary Fig. 2). Most DBM

populations appeared to experience low disruption of gene-climate associations under future climate. Taken together with our previous findings that high levels of both intrapopulation genetic polymorphism and interpopulation genetic differentiation enable DBM to adapt readily to different environments worldwide^{15,21}, we thus assume that DBM will likely remain a damaging pest across most of its range. In contrast, high levels of genetic offset, indicating potentially lower capacity to adapt to future climate change, were observed in scattered populations of Asia (Fig. 2a).” (Lines 90 - 103)

L. 105-106: Here, it is not clear to me how to interpretate the statement that local adaptation in the species involves “physiological and genetic responses”. Usually, local adaptation refers to genetic differentiation among populations that leads to a fitness advantage of native over foreign genotypes in a given environment (e.g. Kawecki & Ebert, Ecology Letters 2004), whereas physiological response rather points to phenotypic plasticity. Or do you refer to local adaptation (i.e. genetically based among population differentiation) IN the physiological response? Please rephrase accordingly.

We have rephrased the sentence and moved it to the beginning of next paragraph: “Local adaptation, resulting from environment-driven intraspecific genetic differentiation, is an important feature of species that inhabit spatially heterogeneous habitats²³.” (Lines 113 - 115)

L. 119-122: Again, note that genetic offset does not represent adaptive potential! This interpretation of your results is thus not correct.

L. 121: Please replace “could be moderated by their genetically adaptive potential” with “could be moderate since they are predicted to experience only minor interruptions of gene-climate associations”. Please note that GO does not tell anything about the adaptive potential of populations, but rather about the necessity to adapt.

Thank you for the correction. As suggested, the sentence has been reformulated as “Under climate-change scenario RCP8.5, the challenges to most populations (Fig. 2b) can be moderate since they are predicted to experience only minor interruptions of gene-climate associations, except for some populations in South America and Southeast Asia (Fig. 2a).” (Lines 120 - 123)

L. 125: This interpretation seems far-fetched since the relationship between most of the genomic markers used for the offset calculation and physiological responses is unknown. Experiments would be required to clarify as to how far populations’ physiological stress response and fitness depend on genetic offset.

We have deleted the sentence. We agree that experiments would be required to clarify how far populations’ physiological stress response and fitness depend on genetic offset.

L. 150: Please replace “has been evidenced” with “have been evidenced”

As suggested, the sentence has been revised accordingly. (Line 149)

L. 193: I would suggest reformulating as “tied to variation in the physiological response to climate”.

As suggested, the sentence has been revised accordingly. (Line 192)

L.198: “majority” -> “the majority”

As suggested, the sentence has been revised accordingly. (Line 197)

L. 366: Did you really calculate the genetic offsets as Euclidean distances between transformed current and future climates in a given grid cell? This approach is usually used for gradient forest based genomic offsets whereas the `gdm.predict` function of the GDM package in R predicts FSTs between the projected genomic compositions of populations under current and future climate based on the fitted GDM (i.e. the obtained values should be bounded between 0 and 1). The latter is preferable, because FST has a straightforward biological interpretation, whereas Euclidean distances between transformed climates have no clear biological meaning, they increase automatically with an increasing number of climatic variables involved (curse of dimensionality) and a statement about whether an offset is large or small (compared to other studies) is therefore not possible. Please clarify. I strongly advice re-calculating the offsets using the `gdm.predict` function if really Euclidean distances were used.

As suggested, we have re-calculated the genetic offsets (GO) using the `gdm.predict` function of the GDM package in R, and update Fig. 2 and Supplementary Fig. 2 accordingly.

L. 395 ff: I share the concerns of Reviewer 3 about calculating the eco-genetic index based on a combination of genomic offset and difference in ecoclimatic index, whereby both the genomic offset and change in habitat suitability depend on climate. It does not become clear to me how the stress induced by climate change itself can be disentangled from stress of a lack of genomic adaptation to this change, since both will be correlated. At least a better, verbally formulated, explanation of what the index represents mechanistically, would be necessary.

We have added/reformulated a few paragraphs in our manuscript to make our arguments stronger.

(1) Results:

Original wording: “To understand how genetic variation interacts with ecophysiological effects to reflect population-level responses of species to future climate, we developed a new metric, the “eco-genetic index” (EGI), which combined the genetic vulnerability and EI-based prediction of climatic habitat suitability.”

Revised wording: “Local adaptation, resulting from environment-driven intraspecific genetic differentiation, is an important feature of species that inhabit spatially heterogeneous habitats²³. To better reflect the role of population-level genetic

variation in DBM's responses to climate change, we developed a new metric, the "eco-genetic index" (EGI), which combines the genetic offset⁴ and EI-based prediction of climatic habitat suitability." (Lines 113 - 117)

2) Discussion:

New paragraph: "To understand the potential distribution and pest status of DBM, Zalucki and Furlong¹⁶ developed a CLIMEX-based algorithm to predict the ecoclimatic index (EI) for the habitat suitability of DBM using climate data and eco-physiological traits. However, the EI-based prediction of habitat suitability is highly generalized without reflecting population-specific tolerance levels and genetic variations. In this paper, based on our recently-generated genomic data of a worldwide sample¹⁵, we developed a new metric, the eco-genetic index (EGI), which combines the genetic offset (GO)⁴ and DEI (difference in ecoclimatic index (DEI) between current and projected future climate scenarios). Although the EGI metric is imperfect, it illustrates how population genetic variation can be tied with physiological effects⁵¹ for predicting the adaptive capacity of DBM, particularly when the population-specific data on physiological tolerance levels are not available. Our results indicate that DEI and GO are functionally complementary. With the EI prediction, we are able to determine the regional distribution of DBM throughout the world, which helps us focus this project on the regions in which DBM persists year-round with a positive ecoclimatic index ($EI > 0$)¹⁶. Using the new metric (EGI), we have identified that some DBM populations in regions of decreasing EI (such as in the central Africa and southern China) under future climate can overcome the challenge of habitat given lower levels of genetic vulnerability, suggesting their capacity of adaptive evolution *in situ* to track climate change (Fig. 2). Looking ahead, if population-specific data on tolerance levels (to match our population-specific genomic data), and data on phenotypic plasticity and epigenetic responses were available, we would be able to make more robust conclusions, potentially supported by the development of a still more sophisticated metric that incorporates relevant information on additional aspects of adaptive capacity." (Lines 223 - 244)

3) Methods:

Rewritten *Prediction of eco-genetic adaptation* section: "The ecoclimatic index (EI) values based on CLIMEX simulations are highly generalized and do not reflect evolutionary adaptation and variation of tolerance levels among different populations, because of the absence of relevant information on fitness traits. Based on our recently-generated dataset of genome-wide single nucleotide polymorphisms (SNPs) sequenced from a worldwide DBM sample of different locations (sites) across a diverse range of biogeographical regions¹⁵, we developed a new metric, the "eco-genetic index" (EGI), which combines the predictions based on genetic offset (GO) with the difference in ecoclimatic index between current and projected future climate scenarios (DEI). EGI allows us to incorporate our population-specific genomic data into EI for better predicting the habitat suitability of DBM subject to climate in 2050 under RCP8.5 scenario based on NorESM1-M GCM. Because DEI

and GO, are correlated (Pearson's⁷⁶ $R = 0.53$, $P < 0.001$; Supplementary Fig. 5), we used a linear normalization transfer function to make values of dei_i and go_i dimensionless, and then used the weighted geometric averaging (WGA) operator and the artificial bee colony (ABC) algorithm to optimize the value of alpha and improve the algorithm of EGI. Here, only regions of decreasing EI ($DEI < 0$) were considered because in these regions, DBM populations will be challenged by habitat suitability decline under the RCP8.5 scenario for 2050. Each gridded point in ArcGIS was described as a vector: $P_i = \{dei_i, go_i\}$, where $i = 1, 2, \dots, n$, and n is the number of the gridded points. Let the EGI and GO of each point be egi_i and go_i , and they are calculated as follows:

Step 1: Calculate the absolute values of dei_i and go_i and normalize^{72,73} them into $[0.1, 0.9]$.

Step 2: (Please see the details in the text of our revised manuscript)
(Lines 413 - 468)

References

- Fitzpatrick, M. C. & Keller, S. R. Ecological genomics meets community-level modelling of biodiversity: mapping the genomic landscape of current and future environmental adaptation. *Ecol. Lett.* 18, 1-16 (2015).
- Fitzpatrick, M. J. & Edelsparre, A. H. The genomics of climate change. *Science* 359, 29-30 (2018).
- Furlong, M. J., Wright, D. J., Dossall, L. M. Diamondback moth ecology and management: problems, progress, and prospects. *Annu Rev Entomol.* 58: 517-541 (2013)
- Gillham, E. M. A Life of Sir Francis Galton. Oxford: Oxford University Press (2001).
- Huey, R. B., Kearney, M. R., Krockenberger, A., Holtum, J. A. M., Jess, M. & Williams, S.E. Predicting organismal vulnerability to climate warming: roles of behaviour, physiology and adaptation. *Philos. Trans. R. Soc. Lond. B Biol. Sci.* 367, 1665–1679 (2010).
- Talekar, N. S., Shelton, A. M. Biology, ecology, and management of the diamondback moth. *Annu Rev Entomol.* 38, 275-301 (1993)
- Wilczek, A. M. et al. Genetic and physiological bases for phenological responses to current and predicted climates. *Phil. Trans. R. Soc. B.* 365, 3129–3147 (2010).
- You, M. S., Ke, F. S., You, S. J. et al. Variation among 532 genomes unveils the origin and evolutionary history of a global insect herbivore. *Nat. Commun.* 11, 1-8 (2020).
- Zalucki M. & Furlong M. in International Workshop on Management of the Diamondback Moth and Other Crucifer Insect Pests (6th) Predicting outbreaks of a migratory pest: an analysis of DBM distribution and abundance 8-14 (AVRDC: The World Vegetable Centre (2011).

Reviewer comments, response- -

Reviewer #3 (Remarks to the Author):

Thank you for your detailed response to my comments. I appreciate that you can show that DEI and GO apparently do not convey the exact same information. I do think that there are still some uncertainties regarding the wording and interpretation of your results. For instance, in the discussion you write: "Although the EGI metric is imperfect, it illustrates how population genetic variation can be tied with physiological effects for predicting the adaptive capacity of DBM, particularly when the population-specific data on physiological tolerance levels are not available." First, I think "population genetic variation" is a bit confusing in this context. GDM is a distance-based approach, and only indirectly takes into account genetic variation within populations (i.e. sampling sites). Variation within populations can be extremely low but when F_{st} values are correlated to differences in environmental conditions, this pattern may indicate strong selection and local adaptation. Second, aren't the "physiological effects" based on underlying genetic data? Even phenotypic plasticity may have a genetic basis. Indeed, there may also be epigenetic effects on physiology and physiological responses, but this is not clear from the discussion. I don't think you are tying genetic variation to physiological effects, as they are not independent from one another. Genetic responses result in physiological responses, even if genetics only (at least the genotype-environment associations you found) cannot explain all of the observed physiological responses; there is unexplained genetic variation not related to any of the used environmental variation, and there may be non-/epigenetic effects. To this end, I would like to reiterate Reviewer 5's suggestion to discuss the mechanistic representation of EGI.

Reviewer #5 (Remarks to the Author):

I am largely content with how the authors addressed the reviewer comments and have only some minor suggestions for formulation changes, which again focus on eliminating the framing that (low) genetic offsets represent (high) adaptive capacity.

L. 88: Please replace "variation" with "composition"

L. 102: Please replace "indicating potentially lower capacity to adapt" with "indicating the need for more comprehensive adaptive change"

L. 183: "allows" -> "allowed"

L. 193: I would prefer the formulation "will experience little interruption of existing gene-environment associations under projected future climate".

L. 239-240: Please replace "suggesting their capacity of adaptive evolution" with "suggesting that only minor adaptive evolution will be required"

L. 823: Please replace "Genetic vulnerability and adaptive potential of" with "Vulnerability of DBM to climate change under ..". Please also use the full species name in all figure and table captions - these need to be interpretable for the reader without reference to the main text.

RESPONSES TO REVIEWERS' COMMENTS

REVIEWERS' COMMENTS

Reviewer #3 (Remarks to the Author):

Thank you for your detailed response to my comments. I appreciate that you can show that DEI and GO apparently do not convey the exact same information. I do think that there are still some uncertainties regarding the wording and interpretation of your results. For instance, in the discussion you write: “Although the EGI metric is imperfect, it illustrates how population genetic variation can be tied with physiological effects for predicting the adaptive capacity of DBM, particularly when the population-specific data on physiological tolerance levels are not available.” First, I think “population genetic variation” is a bit confusing in this context. GDM is a distance-based approach, and only indirectly takes into account genetic variation within populations (i.e. sampling sites). Variation within populations can be extremely low but when F_{st} values are correlated to differences in environmental conditions, this pattern may indicate strong selection and local adaptation. Second, aren't the “physiological effects” based on underlying genetic data? Even phenotypic plasticity may have a genetic basis. Indeed, there may also be epigenetic effects on physiology and physiological responses, but this is not clear from the discussion. I don't think you are tying genetic variation to physiological effects, as they are not independent from one another. Genetic responses result in physiological responses, even if genetics only (at least the genotype-environment associations you found) cannot explain all of the observed physiological responses; there is unexplained genetic variation not related to any of the used environmental variation, and there may be non-/epigenetic effects. To this end, I would like to reiterate Reviewer 5's suggestion to discuss the mechanistic representation of EGI.

Many thanks for your kind comments and helpful suggestion on how to clarify our argument and avoid any possible confusion. The key change we have made is to drop mention of “population genetic variation and physiological effects”. The relevant sentence now reads as follows. “ In this study, we found that most DBM populations might experience little interruption of existing gene-environment associations under projected future climates.” (Lines 195 - 197). Further, to discuss the mechanistic representation of EGI as you suggested, we have clarified in the discussion the explanation of what this index represents. The relevant section has been changed to read as follows. “In this paper, based on our recently-generated genomic data from a worldwide sample¹⁵, we have developed a new metric, the eco-genetic index (EGI), which combines the genetic offset (GO)⁴ and difference in ecoclimatic index (DEI) between current and projected future climate scenarios. This index mechanistically represents the genetic change required for adaptation to a changing environment, and reflects how population-specific genomic data can be incorporated into EI to better predict the habitat suitability of DBM when subjected to climate change, particularly

when the population-specific data on physiological tolerance levels are not available. ”
(Lines 231 - 238).

To address your comment on the epigenetic effects on physiology and physiological responses, we have revised the last paragraph in the discussion to make this point clearer. The revised statements are as follows. “It is increasingly recognized that acclimatization through non-genetic inheritance (e.g. epigenetic processes) may buffer populations against environmental changes, allowing rapid adaptive responses to climate change⁵¹⁻⁵³. Because the mutation rate of epigenetic sites is significantly higher than that of DNA sequences, epigenetic modification provides a complementary mode for species to respond to a changing environment in a rapid and finely regulated process^{54,55}. In addition to directly regulating the expression of temperature responsive genes, epigenetic effects can also regulate other traits to indirectly affect the response of insects to temperature fluctuation⁵⁶. However, epigenetics is a recently emerged field in insect studies⁵⁷, so further work is needed to understand the role of non-genetic effects in adaptation to future climates including how they interact with genetic adaptive capacity.” (Lines 250 - 259)

Reviewer #5 (Remarks to the Author):

I am largely content with how the authors addressed the reviewer comments and have only some minor suggestions for formulation changes, which again focus on eliminating the framing that (low) genetic offsets represent (high) adaptive capacity.

L. 88: Please replace “variation” with “composition”

As suggested, “variation” has been replaced with “composition”. (Line 91)

L. 102: Please replace “indicating potentially lower capacity to adapt” with “indicating the need for more comprehensive adaptive change”

As suggested, “indicating potentially lower capacity to adapt” has been replaced with “indicating the need for more comprehensive adaptive change”. (Line 105)

L. 183: “allows” -> “allowed”

As suggested, “allows” has been changed to “allowed”. (Line 186)

L 193: I would prefer the formulation “will experience little interruption of existing gene-environment associations under projected future climate”.

As suggested, the sentence has been reformulated as: “In this study, we found that most DBM populations will experience little interruption of existing gene-environment associations under projected future climate” (Lines 195-197)

L. 239-240: Please replace “suggesting their capacity of adaptive evolution” with “suggesting that only minor adaptive evolution will be required”

As suggested, “suggesting their capacity of adaptive evolution” has been replaced

with “suggesting that only minor adaptive evolution will be required”. (Line 244)

L. 823: Please replace “Genetic vulnerability and adaptive potential of“ with “Vulnerability of DBM to climate change under ..”. Please also use the full species name in all figure and table captions – these need to be interpretable for the reader without reference to the main text.

As suggested, the figure caption has been changed to: “Vulnerability of diamondback moth to climate change under greenhouse gas emission scenario RCP8.5 in 2050.”(Line 835)

Also, we have used full species name in all figure and table captions in our revised version.